# ACCELERATED PARALLEL TEMPERING VIA NEURAL TRANSPORTS

**Leo Zhang**[1*] **Peter Potaptchik**[1*] **Jiajun He**[2*] **Yuanqi Du**[3]
**Arnaud Doucet**[1] **Francisco Vargas**[4] **Hai-Dang Dau**[5] **Saifuddin Syed**[6]

[1]University of Oxford, [2]University of Cambridge, [3]Cornell University,
[4]Xaira Therapeutics, [5]National University of Singapore, [6]University of British Columbia

## ABSTRACT

Markov Chain Monte Carlo (MCMC) algorithms are essential tools in computational statistics for sampling from unnormalised probability distributions, but can be fragile when targeting high-dimensional, multimodal, or complex target distributions. Parallel Tempering (PT) enhances MCMC's sample efficiency through annealing and parallel computation, propagating samples from tractable reference distributions to intractable targets via state swapping across interpolating distributions. The effectiveness of PT is limited by the often minimal overlap between adjacent distributions in challenging problems, which requires increasing computational resources to compensate. We introduce a framework that accelerates PT by leveraging neural samplers—including normalising flows, diffusion models, and controlled diffusions—to reduce the required overlap. Our approach utilises neural samplers in parallel, circumventing the computational burden of neural samplers while preserving the asymptotic consistency of classical PT. We demonstrate theoretically and empirically on a variety of multimodal sampling problems that our method improves sample quality, reduces the computational cost compared to classical PT, and enables efficient free energy/normalising constant estimation.

## 1 INTRODUCTION

Sampling from a probability distribution $\pi(x) = \exp(-U(x))/Z$ defined over a state-space $\mathcal{X}$ with a tractable un-normalised density $\tilde{\pi} : \mathcal{X} \to \mathbb{R}$ and intractable normalising constant $Z = \int_{\mathcal{X}} \tilde{\pi}(x)\mathrm{d}x$ is a fundamental task in machine learning and natural sciences. Markov Chain Monte Carlo (MCMC) methods are usually employed for such purposes, constructing an ergodic Markov chain $(X_t)_{t\in\mathbb{N}}$ using local moves leaving $\pi$ invariant. While MCMC algorithms are guaranteed to converge asymptotically, in practice, they struggle when the target is complex with multiple well-separated modes (Papamarkou et al., 2022; Hénin et al., 2022). To handle such cases, *Parallel Tempering* (PT) (Swendsen & Wang, 1986; Geyer, 1991; Hukushima & Nemoto, 1996) is a popular class of MCMC methods designed to improve the global mixing of locally efficient MCMC algorithms.

PT works by considering an *annealing path* $\pi^0, \pi^1, \ldots, \pi^N$ of distributions over $\mathcal{X}$ interpolating between a simple reference distribution $\pi^0 = \eta$ (e.g., a Gaussian) and the target $\pi^N = \pi$. PT algorithms construct a Markov chain $\mathbf{X}_t = (X_t^0, \ldots, X_t^N)$ on the extended state-space $\mathcal{X}^{N+1}$, targeting the joint distribution $\boldsymbol{\pi} = \pi^0 \otimes \cdots \otimes \pi^N$. The PT chain $\mathbf{X}_t$ is constructed by alternating between (1) a *local exploration phase* where the $n$-th chain [1] of $\mathbf{X}_t$ is updated according to a $\pi^n$-invariant MCMC algorithm; and (2) a *communication phase* which proposes a sequence of swaps between neighbouring states accepted according to a Metropolis–Hastings correction ensuring invariance (see Figure 1 (a)). Crucially, PT offsets the additional computation burden of simulating the extended $N$ chains through parallel computation, allowing for a similar effective computational cost as a single chain when implemented in a maximally parallelised manner.

Typically, the chains $X_t^n$ mix faster when closer to the reference and struggle closer towards the target. Therefore, communication between the reference and target, facilitated through swaps, can

---

*First Authors. Corresponding to <leo.zhang@stx.ox.ac.uk>
[1]Following the PT literature, we also refer to components of $\mathbf{X}_t$ as chains.

induce rapid mixing between modes of the target component (Woodard et al., 2009; Surjanovic et al., 2024). While the importance of the swapping mechanism for PT has led to a literature dedicated to optimising communication between the reference and target (Syed et al., 2021; 2022; Surjanovic et al., 2022), such works still rely on the original swapping mechanism (Geyer, 1991).

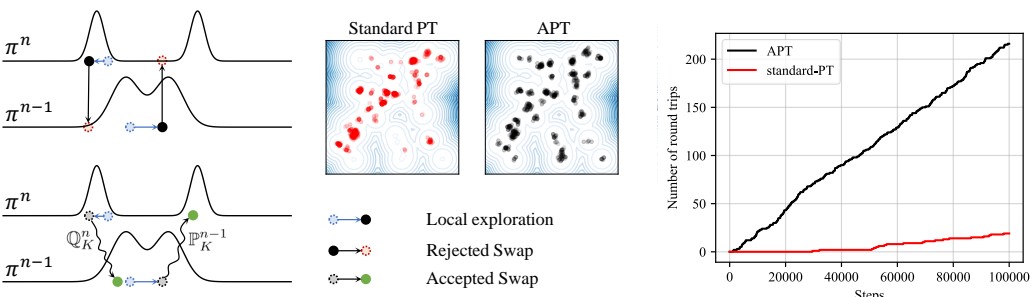

Figure 1: (Left) An illustration of the local exploration and communication step for PT vs APT. (Middle) 1,000 samples of a Gaussian mixture model target obtained using PT vs APT with a standard Gaussian reference. See Appendix 6.1 for more details. (Right) Round trips for PT and APT with $N = 6$ chains over $T = 100,000$ iterations of Algorithm 1.

As an alternative to PT, recent work has explored both continuous (Zhang & Chen, 2022; Vargas et al., 2023; Berner et al., 2024; Akhound-Sadegh et al., 2024; Vargas et al., 2024; Máté & Fleuret, 2023; Albergo & Vanden-Eijnden, 2024; Erives et al., 2025) and discrete (Noé et al., 2019; Papamakarios et al., 2021; Midgley et al., 2023; Gabrié et al., 2022) flows for sampling, under the umbrella of *neural samplers*. However, these methods usually incur a bias, foregoing theoretical guarantees of MCMC, and can be expensive to implement and train. Due to these shortcomings, standard PT provides a strong baseline that neural samplers struggle to match (He et al., 2025b).

Recent work (Arbel et al., 2021; Albergo & Vanden-Eijnden, 2024; Phillips et al., 2024; Chen et al., 2025) has explored approaches to debias neural samplers using Sequential Monte Carlo (SMC)-based ideas (Del Moral et al., 2006). Despite their consistency guarantees, these approaches do not address the mode-collapsing nature of many modern neural samplers (He et al., 2025b). PT offers a comparable but computationally dual framework to SMC, where the roles of parallelism and time are reversed (Syed et al., 2024). In SMC, particles are generated in parallel, and approximate annealing distributions are constructed sequentially. In contrast, in PT, particles are generated sequentially and annealing distributions are built in parallel. This raises the question: just as neural samplers have been integrated with SMC, can we integrate neural samplers with PT, combining the consistency of PT with the flexibility of neural samplers?

We answer positively this question by formalising and exploiting the framework introduced by Ballard & Jarzynski (2009; 2012) in physics for designing more flexible swap mechanisms, which we call *Accelerated Parallel Tempering* (APT). APT preserves PT's asymptotic consistency and allows us to easily integrate normalising flows, diffusion models and stochastic control into existing PT implementations. Moreover, APT uses these neural samplers in a *parallelised* manner, mitigating their high computational burden. Empirically, APT outperforms PT by accelerating the communication between the reference and target states, even when controlling for the additional computation incurred by neural samplers.

A relevant prior work integrating PT with normalising flows is Invernizzi et al. (2022), which trains a flow to directly map configurations from the highest-temperature reference to the lowest-temperature target, effectively bypassing the intermediate annealing distributions. Abbott et al. (2024) also combines PT with normalising flows to accelerate sampling in lattice quantum chromodynamics; their approach is equivalent to the NF-APT scheme of Section 5.1. Our framework is strictly more general, accommodating any learned or approximate transport map between neighbouring distributions.

## 2 PARALLEL TEMPERING

Let $\pi^0, \pi^1, \ldots, \pi^N$ be an *annealing path* of probability distributions on $\mathcal{X}$ where $\pi^0 = \eta$ is the reference and $\pi^N = \pi$ is the target. We assume the $n$-th *annealing distribution* admits density $\pi^n(x) := \tilde{\pi}^n(x)/Z_n$ with respect to a base measure $\mathrm{d}x$, where $\tilde{\pi}^n : \mathcal{X} \to \mathbb{R}$ is the un-normalised

density which we can evaluate, with normalising constant $Z_n := \int_{\mathcal{X}} \tilde{\pi}^n(x)\mathrm{d}x$. Our goal is to estimate $\pi[f] := \int_{\mathcal{X}} f(x)\pi(\mathrm{d}x)$, the expectation of $f : \mathcal{X} \to \mathbb{R}$ with respect to $\pi = \pi^N$, and the normalising constant $Z = Z_N$.

There is considerable flexibility in choosing annealing distributions provided $\pi^0 = \eta$ with $Z_0 = 1$ and $\pi^N = \pi$ with $Z_N = Z$. Without loss of generality, we can assume $\pi^n = \pi^{\beta_n}$ where for $\beta \in [0, 1]$, $\pi^\beta$ continuously interpolates between the reference and target as $\beta$ increases from 0 to 1, according to some *annealing schedule* $0 = \beta_0 < \cdots < \beta_N = 1$. A common choice is the *geometric path*, $\pi^\beta(x) \propto \eta(x)^{1-\beta}\pi(x)^\beta$, which linearly interpolates between reference and target in log-space. See Masrani et al. (2021); Syed et al. (2021); Máté & Fleuret (2023); York (2023) for alternative non-geometric annealing paths.

## 2.1 Non-Reversible Parallel Tempering

The PT algorithm constructs a Markov chain $\mathbf{X}_t = (X_t^0, \ldots, X_t^N)$ on the extended state-space $\mathcal{X}^{N+1}$ invariant to the joint distribution $\boldsymbol{\pi} := \pi^0 \otimes \cdots \otimes \pi^N$. We construct $\mathbf{X}_t$ from $\mathbf{X}_{t-1}$ by doing (1) a *local exploration* step followed by (2) a *communication* step seen in the top of Figure 1 (a). For $n = 0, \ldots, N$, the $n$-th local exploration move (1) updates the $n$-th component of $\mathbf{X}_{t-1}$ using a $\pi^n$-invariant Markov kernel $K^n(x, \mathrm{d}x')$ on $\mathcal{X}$ corresponding to an MCMC move targeting $\pi^n$,

$$X_t^n \sim K^n(X_{t-1}^n, \mathrm{d}x^n),$$

We additionally assume that $K^0(x, \mathrm{d}x') = \eta(\mathrm{d}x')$ corresponds to an independent sample from the reference. The communication step (2) applies a sequence of swap moves between adjacent components of $\mathbf{X}_t$, where the $n$-th swap move illustrated in Figure 1 exchanges components $X_t^{n-1}$ and $X_t^n$ in $\mathbf{X}_t = (X_t^0, \ldots, X_t^N)$ with probability $\alpha^n(X_t^{n-1}, X_t^n)$, where for $x, x' \in \mathcal{X}$,

$$\alpha^n(x, x') := \min\left\{1, \frac{w^n(x')}{w^n(x)}\right\}. \tag{1}$$

Where $w^n : \mathcal{X} \to \mathbb{R}$ is the *incremental weight* equal to the un-normalised Radon-Nikodym derivative:

$$w^n(x) := \frac{Z_n}{Z_{n-1}}\frac{\mathrm{d}\pi^n}{\mathrm{d}\pi^{n-1}}(x) = \frac{\tilde{\pi}^n(x)}{\tilde{\pi}^{n-1}(x)}. \tag{2}$$

In practice, it is advantageous to use a *non-reversible* communication (Okabe et al., 2001; Syed et al., 2022), where the $n$-th swap move is proposed only at iterations with matching parity: $n \equiv t \mod 2$. Both local exploration and communication steps can be done in parallel, allowing distributed implementations to leverage parallel computation to accelerate sampling (Surjanovic et al., 2023).

**Round trips.** While the effective sample size (ESS) of samples generated by a Markov chain is the gold standard for evaluating the performance of MCMC algorithms, in our setting, we are mainly interested in improving the swap kernel within PT. As ESS measures the intertwined performance of the local exploration and swap kernels, we instead evaluate the performance of PT and APT in terms of the communication between reference and target, in order to disentangle the influence of the swap kernels from the local exploration. This is empirically measured by counting the total number of *round trips* $R_T$ which tracks the number of independent reference samples transported to the target after $T$ iterations of PT (Katzgraber et al., 2006; Lingenheil et al., 2009); see Appendix A for a formal definition. In particular, the mixing time of PT is related to the time it takes for a round trip to occur and the total number of round trips is a measure of the particle diversity generated by PT and strongly correlates with the ESS (Surjanovic et al., 2024).

## 3 Accelerated Parallel Tempering

A limitation of PT is its inflexible swap move, which only proposes directly exchangeable samples between distributions, considering just the relative change in likelihood $\pi^{n-1}$ and $\pi^n$. Low acceptance probability occurs when these distributions have minimal overlap. Addressing this requires increasing the number of parallel chains $N$ which may not always be possible. We propose *Accelerated Parallel Tempering* (APT), expanding the framework developed in Ballard & Jarzynski (2009; 2012), to improve distributional overlap and accelerate communication for a fixed number of chains.

---

**Algorithm 1** Accelerated Parallel Tempering

1: Initialise $\mathbf{X}_0 = (X_0^0, \ldots, X_0^N)$;
2: **for** $t = 1, \ldots, T$ **do**
3:     $\mathbf{X}_t = (X_t^0, \ldots, X_t^N), \quad X_t^n \sim K^n(X_{t-1}^n, \mathrm{d}x)$             $\triangleright$ Local exploration move
4:     **for** $n \equiv t \mod 2$ **do**                   $\triangleright$ Non-reversible communication
5:         $\vec{X}_{t,0}^{n-1}, \overleftarrow{X}_{t,K}^n \leftarrow X_t^{n-1}, X_t^n$         $\triangleright$ Initialise forward/backward paths
6:         **for** $k = 1, \ldots, K$ **do**
7:             $\vec{X}_{t,k}^{n-1} \sim P_k^{n-1}(\vec{X}_{t,k-1}^{n-1}, \mathrm{d}x)$               $\triangleright$ Accelerate forward
8:             $\overleftarrow{X}_{t,K-k}^n \sim Q_{K-k}^n(\overleftarrow{X}_{t,K-k+1}^n, \mathrm{d}x)$         $\triangleright$ Accelerate backward
9:         **end for**
10:         $\vec{w}_{K,t}^n, \overleftarrow{w}_{K,t}^n \leftarrow w_K^n(\vec{X}_{t,0:K}^{n-1}), w_K^n(\overleftarrow{X}_{t,0:K}^n)$   $\triangleright$ Work of forward/backward paths
11:         $U \sim \mathrm{Uniform}([0,1])$
12:         **if** $\log U < \log \vec{w}_{K,t}^n - \log \overleftarrow{w}_{K,t}^n$ **then**       $\triangleright$ Accelerated swap move
13:             $X_t^{n-1}, X_t^n \leftarrow \overleftarrow{X}_{t,0}^n, \vec{X}_{t,K}^{n-1}$
14:         **end if**
15:     **end for**
16: **end for**
**Output:** Return: $\mathbf{X}_1, \ldots, \mathbf{X}_T$

---

## 3.1 FORWARD AND BACKWARD ACCELERATORS

For $n = 1, \ldots, N$, let $P_k^{n-1}(x_{k-1}, \mathrm{d}x_k)$ and $Q_{k-1}^n(x_k, \mathrm{d}x_{k-1})$, $k = 1, \ldots, K$, be two families of $K$ transition kernels which we call *forward accelerators* and *backward accelerators*. They induce $\mathbb{P}_K^{n-1}$ and $\mathbb{Q}_K^n$, two time-inhomogeneous Markov processes obtained by propagating $\pi^{n-1}$ forward in time and $\pi^n$ backward in time respectively using the forward and the backward accelerators:

$$\mathbb{P}_K^{n-1}(\mathrm{d}x_{0:K}) := \pi^{n-1}(\mathrm{d}x_0)\prod_{k=1}^K P_k^{n-1}(x_{k-1}, \mathrm{d}x_k), \mathbb{Q}_K^n(\mathrm{d}x_{0:K}) := \pi^n(\mathrm{d}x_K)\prod_{k=1}^K Q_{k-1}^n(x_k, \mathrm{d}x_{k-1}).$$

We assume $\mathbb{P}_K^{n-1}$ and $\mathbb{Q}_K^n$ are mutually absolutely continuous and we can evaluate $w_K^n : \mathcal{X}^{K+1} \to \mathbb{R}$, the incremental weights between the forward and the backward paths, extending $w^n$ in Equation (2),

$$w_K^n(x_{0:K}) := \frac{Z_n}{Z_{n-1}}\frac{\mathrm{d}\mathbb{Q}_K^n}{\mathrm{d}\mathbb{P}_K^{n-1}}(x_{0:K}). \tag{3}$$

## 3.2 NON-REVERSIBLE ACCELERATED PARALLEL TEMPERING

We construct the Markov chain $\mathbf{X}_t = (X_t^0, \ldots, X_t^N)$ for $t = 1, \ldots, T$ using the same local exploration and non-reversible communication as classical PT, but we use the accelerated PT swap move as described below and summarized in Algorithm 1. Given PT state $\mathbf{X}_t = (X_t^0, \ldots, X_t^N) \in \mathcal{X}^{N+1}$ after the local exploration move, we define the $n$-th accelerated swap move as follows: generate paths $\vec{X}_{t,0:K}^{n-1}$, and $\overleftarrow{X}_{t,0:K}^n$ obtained by propagating $X_t^{n-1}$ and $X_t^n$ forward and backward in time using the forward and backward transitions respectively,

$$\vec{X}_{t,0}^{n-1} = X_t^{n-1}, \quad \vec{X}_{t,k}^{n-1} \sim P_k^{n-1}(\vec{X}_{t,k-1}^{n-1}, \mathrm{d}x_k),$$
$$\overleftarrow{X}_{t,K}^n = X_t^n, \quad \overleftarrow{X}_{t,k-1}^n \sim Q_{k-1}^n(\overleftarrow{X}_{t,k}^n, \mathrm{d}x_{k-1}).$$

The $n$-th accelerated swap move illustrated in Figure 1 replaces components $X_t^{n-1}$ and $X_t^n$ in $\mathbf{X}_t$ with $\overleftarrow{X}_{t,0}^n$ and $\vec{X}_{t,K}^{n-1}$ respectively with probability $\alpha_K^n(\vec{X}_{t,0:K}^{n-1}, \overleftarrow{X}_{t,0:K}^n)$, where for $x_{0:K}, x'_{0:K} \in \mathcal{X}^{K+1}$,

$$\alpha_K^n(x_{0:K}, x'_{0:K}) = \min\left\{1, \frac{w_K^n(x'_{0:K})}{w_K^n(x_{0:K})}\right\}.$$

When $K = 0$, the accelerated swap coincides with the traditional PT swap in Equation (1). However, an accelerated swap can obtain an acceptance of 1 even if $\pi^{n-1} \neq \pi^n$ provided $\mathbb{P}_K^{n-1} = \mathbb{Q}_K^n$.

Therefore, we aim to choose the forward and backward accelerators to make the laws of simulated forward and backward paths as close to each other as possible. Theorem 1 shows we can quantify this discrepancy through the *(theoretical) rejection rate*, $r(\mathbb{P}_K^{n-1}, \mathbb{Q}_K^n) := \|\mathbb{P}_K^{n-1} \otimes \mathbb{Q}_K^n - \mathbb{Q}_K^n \otimes \mathbb{P}_K^{n-1}\|_{\mathrm{TV}}$ which by Pinsker inequality is controlled by the symmetric KL divergence between $\mathbb{P}_K^{n-1}$ and $\mathbb{Q}_K^n$ (for the proof, see Appendix B.1):

$$r(\mathbb{P}_K^{n-1}, \mathbb{Q}_K^n)^2 \leq \frac{1}{2}\mathbb{P}_K^{n-1}[-\log w_K^n] + \frac{1}{2}\mathbb{Q}_K^n[\log w_K^n] =: \mathrm{SKL}(\mathbb{P}_K^{n-1}, \mathbb{Q}_K^n).^2 \tag{4}$$

**Theorem 1.** *The APT Markov chain $X_t$ generated by Algorithm 1 is ergodic and $\pi$-invariant. Moreover, the probability the $n$-th accelerated swap is rejected at stationarity equals $r(\mathbb{P}_K^{n-1}, \mathbb{Q}_K^n)$.*

**Expectation and free energy estimators.** As a by-product of Algorithm 1, we obtain a consistent estimator $\hat{\pi}_T^n[f] = \frac{1}{T}\sum_{t=1}^T f(X_t^n)$ for the expectation of $f : \mathcal{X} \to \mathbb{R}$ with respect to $\pi^n$ by taking the Monte Carlo average over the $n$-th chain $X_t^n$. To also obtain free energy estimates, we can average over $\vec{w}_{K,t}^n := w_K^n(\vec{X}_{t,0:K}^{n-1})$ and $\overleftarrow{w}_{K,t}^n := w_K^n(\overleftarrow{X}_{t,0:K}^n)$, the weights of the forward and backward accelerated paths generated during the $n$-th accelerated swap at time $t$ during the communication phase of Algorithm 1, to obtain consistent estimators $\vec{Z}_T$ and $\overleftarrow{Z}_T$ for $Z$ respectively as $T \to \infty$,

$$\vec{Z}_T := \prod_{n=1}^N \frac{2}{T}\sum_{n \equiv t \bmod 2} \vec{w}_{K,t}^n, \quad \frac{1}{\overleftarrow{Z}_T} := \prod_{n=1}^N \frac{2}{T}\sum_{n \equiv t \bmod 2} \frac{1}{\overleftarrow{w}_{K,t}^n}.$$

We then obtain a consistent estimator $\hat{Z}_T = (\vec{Z}_T \overleftarrow{Z}_T)^{1/2}$ for $Z$ by taking the geometric mean of the forward and backward estimators. Proposition 1 provides the formal statement of these results; for the proof, see Appendix B.1.3.

**Proposition 1.** *The estimators $\hat{\pi}_T^n[f]$ and $\hat{Z}_T$ a.s. converge to $\pi^n[f]$ and $Z$ respectively as $T \to \infty$. Moreover, if $\mathbb{P}_K^{n-1} = \mathbb{Q}_K^n$ for all $n$, then $\hat{Z}_T \stackrel{a.s.}{=} Z$.*

When we consider the classical PT with $K = 0$, these estimators degrade back to free energy perturbation (FEP) (Zwanzig, 1954); when the path is deterministic (e.g., implemented through normalising flow), these estimators recover the target FEP (Jarzynski, 2002); and when the path is stochastic, the estimators can be viewed as a form of Jarzynski equality (Jarzynski, 1997) or escorted Jarzynski equality (Vaikuntanathan & Jarzynski, 2008). Additionally, following He et al. (2025a, Proposition 3.2), we can also apply the Bennett acceptance ratio using the weight for the paths (Bennett, 1976; Shirts et al., 2003; Hahn & Then, 2009; Minh & Chodera, 2009; Vaikuntanathan & Jarzynski, 2011) to achieve reduced variance without additional functional calls.

## 4 ANALYSIS OF ACCELERATED PT

We analyse how the performance, measured by the round trips $R_T$ observed after $T$ iterations, scales with increasing parallel chains $N$ and acceleration time $K$. This is equivalent to analysing the *round trip rate* $\tau := \lim_{T\to\infty} \mathbb{E}[R_T]/T$ defined as the expected percentage of PT iterations where a round trip occurs (Katzgraber et al., 2006; Lingenheil et al., 2009). We use the slope in Figure 1 (Right) to illustrate this quantity. To derive theoretical insights independent of the problem-specific local exploration moves, we employ an *efficient local exploration* assumption analogous to Syed et al. (2021; 2022); Surjanovic et al. (2024), which assumes the weight of forward and backward accelerated paths is independent across chains and PT iterations.

**Assumption 1** (Efficient local exploration). *For all $n = 1, \ldots, N$, $(\vec{w}_{K,t}^n, \overleftarrow{w}_{K,t}^n)$ are iid in $t$ and equal in distribution to $(w_K^n(\vec{X}_{0:K}^{n-1}), w_K^n(\overleftarrow{X}_{0:K}^n))$ where $(\vec{X}_{0:K}^{n-1}, \overleftarrow{X}_{0:K}^n) \sim \mathbb{P}_K^{n-1} \otimes \mathbb{Q}_K^n$.*

Proposition 2 relates the round trip rate to the rejections, extending Syed et al. (2021, Corollary 2); for the proof, see Appendix B.2.

**Proposition 2.** *If Assumption 1 holds, then $\tau = \tau(\mathbb{P}_K^{0:N-1}, \mathbb{Q}_K^{1:N})$ where,*

$$\tau(\mathbb{P}_K^{0:N-1}, \mathbb{Q}_K^{1:N}) := \left(2 + 2\sum_{n=1}^N \frac{r(\mathbb{P}_K^{n-1}, \mathbb{Q}_K^n)}{1 - r(\mathbb{P}_K^{n-1}, \mathbb{Q}_K^n)}\right)^{-1}.$$

---

²We note that $[\cdot]$ represents expectation with respect to the measure on the left.

**Asymptotic scaling with acceleration time.** We now fix $N$ and use Proposition 2 to study how the round trip rate scales with $K$ in Proposition 3; for the proof, see Appendix B.3. We focus on a special case where $\mathbb{P}_K^{n-1}$ and $\mathbb{Q}_K^n$ arise as $K$-step discretisations of an underlying stochastic differential equation (SDE) which bridge $\pi^{n-1}$ and $\pi^n$ with path measure $\mathbb{P}_\infty^{n-1}$ and $\mathbb{Q}_\infty^n$ as $K \to \infty$.

**Proposition 3.** *Under appropriate conditions on the drifts of the SDE (Appendix B.3), as $K \to \infty$, $\tau(\mathbb{P}_K^{0:N-1}, \mathbb{Q}_K^{1:N})$ converges to $\tau(\mathbb{P}_\infty^{0:N-1}, \mathbb{Q}_\infty^{1:N})$ and $r(\mathbb{P}_K^{n-1}, \mathbb{Q}_K^n) \leq r(\mathbb{P}_\infty^{n-1}, \mathbb{Q}_\infty^n) + \mathcal{O}(1/\sqrt{K})$.*

The main appeal of APT comes from the fact that for well-designed $\mathbb{P}_K^{0:N-1}, \mathbb{Q}_K^{1:N}$, that aim to induce approximately the same measure, we have

$$r(\mathbb{P}_\infty^{n-1}, \mathbb{Q}_\infty^n) \approx 0 \ll r(\pi^{n-1}, \pi^n).$$

Thus, increasing $K$ can dramatically reduce the rejection rate and increase the round trip rate in challenging scenarios. In practice, we find that for moderate values of $K$, this benefit outweights the increased computational cost as seen in Section 6.1.

**Asymptotic scaling with increased parallelism.** Following the analysis of PT in Syed et al. (2022), we consider how the performance of APT scales as $N \to \infty$. In particular, Theorem 2 shows that as $N \to \infty$, the round trip rate is controlled by the *global barrier* $\Lambda_K$ for APT and this can be estimated by $\sum_{n=1}^N r(\mathbb{P}_K^{n-1}, \mathbb{Q}_K^n)$ (this can be empirically estimated by approximating $r(\mathbb{P}_K^{n-1}, \mathbb{Q}_K^n)$ by the empirical rejection rates observed from Algorithm 1); for the proof, see Appendix B.4.2. This provides a direct generalisation of the analysis of the global barrier for PT from Syed et al. (2022).

**Theorem 2.** *Suppose $\mathbb{P}_K^{\beta,\beta'}$ and $\mathbb{Q}_K^{\beta,\beta'}$ are sufficiently regular and satisfy Assumptions 2–4 in Appendix B.4. If $\max_{n \leq N} |\beta_n - \beta_{n-1}| = O(N^{-1})$ as $N \to \infty$, then $\sum_{n=1}^N r(\mathbb{P}_K^{n-1}, \mathbb{Q}_K^n)$ converges to $\Lambda_K$ and $\tau(\mathbb{P}_K^{0:N-1}, \mathbb{Q}_K^{1:N})$ converges to $\bar{\tau}_K = (2 + 2\Lambda_K)^{-1}$, where $\Lambda_K$ equals,*

$$\Lambda_K := \int_0^1 \frac{1}{2} \mathbb{E}[|\dot{w}_K^\beta(\overleftarrow{X}_{0:K}^\beta) - \dot{w}_K^\beta(\vec{X}_{0:K}^\beta)|]d\beta, \quad (\vec{X}_{0:K}^\beta \overleftarrow{X}_{0:K}^\beta) \sim \mathbb{P}_K^{\beta,\beta} \otimes \mathbb{Q}_K^{\beta,\beta},$$

*and $\dot{w}_K^\beta : \mathcal{X}^{K+1} \to \mathbb{R}$ is the partial derivative with respect to $\beta'$ of $\log w_K^{\beta,\beta'}$ at $\beta' = \beta$.*

Notably, APT observes a sharp deterioration in round trip when $N \ll \Lambda_K$, begins to stabilise when $N \approx \Lambda_K$ and observes a marginal improvement when $N \gg \Lambda_K$. Intuitively, we can consider $\Lambda_K$ to be a statistical invariant of APT which quantifies the difficulty of the sampling problem. As discussed in Proposition 3, for a reasonable choice of accelerators, we expect $\Lambda_K$ to decrease with $K$. This suggests APT can substantially improve compared to classical PT ($K = 0$) for challenging problems by having $\Lambda_K < \Lambda_0$.

**Schedule tuning.** Another important consequence of Theorem 2 is that this allows us to import the schedule tuning algorithm from (Syed et al., 2022, Section 5) by directly following the same reasoning. This provides a simple and robust method for optimising the annealing schedule $\{\beta_n\}_{n=0}^N$ which is one of the most important hyper-parameters impacting the performance of APT. We provide further details in Appendix C.

**Additional analysis.** Finally, we provide a theoretical equivalence of how APT can be viewed as an instance of PT and explore the trade-off between increasing parallelism and increasing acceleration time in Appendix B.5.

## 5 DESIGN SPACE FOR ACCELERATED PT

In this section, we explore the design space of APT where $\mathcal{X} = \mathbb{R}^d$. We provide several variants of APT algorithms which utilise different neural transports in order to improve the round trip rate of APT.

### 5.1 NORMALISING FLOW ACCELERATED PT

Normalising flows (Tabak & Vanden-Eijnden, 2010; Rezende & Mohamed, 2015) provide a flexible framework for approximating complex probability distributions by transforming a simple base distribution (e.g., a Gaussian) through a sequence of invertible, differentiable mappings.

Let $T^n : \mathcal{X} \to \mathcal{X}$ be such a diffeomorphism. We define NF-APT as APT with forward and backward accelerators defined by the deterministic transport given by the normalising flow $T^n$ and its inverse respectively,

$$P_1^{n-1}(x_0, \mathrm{d}x_1) = \delta_{T^n(x_0)}(\mathrm{d}x_1), \quad Q_0^n(x_1, \mathrm{d}x_0) = \delta_{(T^n)^{-1}(x_1)}(\mathrm{d}x_0).$$

Then the incremental weight equals

$$w_1^n(x_0, x_1) = \frac{\tilde{\pi}^n(x_1)}{\tilde{\pi}^{n-1}(x_0)} |\det J_{T^n}(x_0)|, \quad x_1 = T^n(x_0).$$

We can parametrise the normalising flow $T^n$ by a neural network which is trained to minimise $\mathcal{L}(T) = \sum_{n=1}^N \mathrm{SKL}(\mathbb{P}_1^{n-1}, \mathbb{Q}_1^n)$ following Equation (4); see Appendix D.1 for further details. We note that NF-APT is partly motivated by prior work (Arbel et al., 2021; Matthews et al., 2022) which utilise normalising flows to map between annealing distributions in the context of SMC samplers (Del Moral et al., 2006). Interestingly, NF-APT is able to mitigate mode dropping issues associated with these works, as we are able to train with the symmetric KL divergence, since NF-APT provides access to samples from both $\pi^{n-1}$ and $\pi^n$.

## 5.2 Controlled Monte Carlo Diffusions Accelerated PT

Another common transport for sampling is based on the escorted Jarzynski equality (Vaikuntanathan & Jarzynski, 2008). One way to realise this in practice with neural networks is via Controlled Monte Carlo Diffusions (CMCD) (Vargas et al., 2024).

Suppose for $s \in [0, 1]$, we have $b_s^n \in \mathbb{R}^d$, $\sigma_s^n \in \mathbb{R}_+$, and $U_s^n = (1 - \phi_s^n) \log \tilde{\pi}^{n-1} + \phi_s^n \log \tilde{\pi}^n$, where $\phi_s^n \in [0, 1]$ is monotonically increasing in $s$ and $\phi_0^n = 0, \phi_1^n = 1$. For a fixed $K$, let $s_k = k/K$ and $\Delta s_k = 1/K$ be the uniform discretisation of the unit interval. We define CMCD-APT as APT with $k$-th forward accelerator,

$$P_k^{n-1}(x_{k-1}, \mathrm{d}x_k) = \mathcal{N}(x_{k-1} + \Delta s_k (\sigma_{s_{k-1}}^n)^2 \nabla U_{s_{k-1}}^n(x_{k-1}) + \Delta s_k b_{s_{k-1}}^n(x_{k-1}), 2\Delta s_k (\sigma_{s_{k-1}}^n)^2 \mathbf{I}),$$

and the backward accelerator,

$$Q_{k-1}^n(x_k, \mathrm{d}x_{k-1}) = \mathcal{N}(x_k - \Delta s_k (\sigma_{s_k}^n)^2 \nabla U_{s_k}^n(x_k) + \Delta s_k b_{s_k}^n(x_k), 2\Delta s_k (\sigma_{s_k}^n)^2 \mathbf{I}).$$

The corresponding incremental weights for CMCD-APT equals

$$w_K^n(x_{0:K}) = \frac{\tilde{\pi}^n(x_K) \prod_{k=1}^K Q_{k-1}^n(x_k, x_{k-1})}{\tilde{\pi}^{n-1}(x_0) \prod_{k=1}^K P_k^{n-1}(x_{k-1}, x_k)} \tag{5}$$

where $P_k^{n-1}(x_{k-1}, x_k)$ and $Q_{k-1}^n(x_k, x_{k-1})$ are the Gaussian densities of the forward and backwards transition kernels with respect to the Lebesgue measure.

We parametrise and learn the neural transport $b_s^n$, the time schedule $\phi_s^n$ and the diffusion coefficient $\sigma_s^n$ via the objective: $\mathcal{L}(b_s, \phi_s, \sigma_s) = \sum_{n=1}^N \mathrm{SKL}(\mathbb{P}_K^{n-1}, \mathbb{Q}_K^n)$, where we also learn $\phi_s^n, \sigma_s^n$ following Geffner & Domke (2023)[3]. While in theory, once the vectorised field $b_s^n$ is perfectly learned, any $\sigma_s^n$ can be used, we found that learning $\sigma_s^n$ can significantly stabilise the training and enhance performance. We note that alternative divergences and losses can also be employed—see Máté & Fleuret (2023); Richter & Berner (2023); Albergo & Vanden-Eijnden (2024). We discuss the form of the transition kernels, the incremental weights and objective in the limit as $K \to \infty$ in Appendix D.2.

## 5.3 Diffusion Accelerated PT

The Variance-Preserving (VP) diffusion model (Ho et al., 2020; Song et al., 2020) is defined by the (forward) SDE: $dY_s = -\gamma_s Y_s \mathrm{d}s + \sqrt{2\gamma_s} \mathrm{d}W_s$ with $s \in [0, 1]$, $Y_0 \sim \pi$, and a rate function $\gamma_s : [0, 1] \to \mathbb{R}^+$. The time-reversal SDE $(X_s)_{s \in [0,1]} = (Y_{1-s})_{s \in [0,1]}$ has the form $dX_s = [\gamma_{1-s} X_s + 2\gamma_{1-s} \nabla \log \pi_s^{\mathrm{VP}}(X_s)]\mathrm{d}s + \sqrt{2\gamma_{1-s}} \mathrm{d}W_s$ where $\pi_s^{\mathrm{VP}}$ is the density of $Y_{1-s}$ and $X_0 \sim \pi_0^{\mathrm{VP}}$ is close to a standard Gaussian in the common scenario that $\int_0^1 \gamma_s \mathrm{d}s \gg 1$. Samples from $\pi$ can then

---

[3] We note that in standard CMCD, we cannot use the symmetric KL divergence in the objective as standard CMCD only has access to data from one side.

Table 1: PT versus APT with different acceleration methods, targeting a 40-mode Gaussian Mixture model (GMM-10) target in 10 dimensions and standard Gaussian reference using $N = 6, 10, 30$ parallel chains for $T = 100,000$ iterations. For each method, we report the round trips (R), round trips per target evaluation, denoted as compute-normalised round trips (CN-R), the number of neural network evaluations per parallel chain every iteration (Neural Calls), and $\Lambda_K$ estimated using $N = 30$ chains $(\hat{\Lambda}_K)$.

| # Chain | | | $N = 6$ | | $N = 10$ | | $N = 30$ | |
|---|---|---|---|---|---|---|---|---|
| Method | Neural Calls ($\downarrow$) | $\hat{\Lambda}_K$ ($\downarrow$) | R ($\uparrow$) | CN-R ($\uparrow$) | R ($\uparrow$) | CN-R ($\uparrow$) | R ($\uparrow$) | CN-R ($\uparrow$) |
| NF-APT | 1 | 7.198 | 194 | 97.0 | 1655 | 827.5 | 2441 | 1220.5 |
| CMCD-APT ($K = 1$) | 2 | 6.911 | 234 | 117.0 | 2126 | 1063.0 | 3264 | **1632.0** |
| CMCD-APT ($K = 2$) | 3 | 5.932 | 526 | 175.3 | 3287 | **1092.7** | 4767 | 1589.0 |
| CMCD-APT ($K = 5$) | 6 | **4.822** | 1743 | **290.5** | **5525** | 920.8 | **6231** | 1038.5 |
| Diff-APT ($K = 1$) | 2 | 9.025 | 375 | 187.5 | 1551 | 775.5 | 2820 | 1410.0 |
| Diff-APT ($K = 2$) | 3 | 7.298 | 748 | 249.3 | 2064 | 688.0 | 3480 | 1160.0 |
| Diff-APT ($K = 5$) | 6 | 5.795 | 1565 | 260.8 | 3080 | 513.3 | 4334 | 722.3 |
| Diff-PT ($K = 0$) | 2 | 8.932 | 204 | 102.0 | 734 | 367.0 | 1586 | 793.0 |
| PT | **0** | 8.346 | 17 | 8.5 | 681 | 340.5 | 1888 | 944.0 |

be generated by simulating the time-reversal SDE initialised with samples from $\mathcal{N}(0, \mathbf{I})$; the score $\nabla \log \pi_s^{\text{VP}}$ is unknown but can be learned by score matching objectives.

To employ diffusion models for APT, we consider parametrising an energy-based model (Salimans & Ho, 2021) $\pi_s^\theta$ which satisfies the boundary conditions: $\pi_0^\theta = \mathcal{N}(0, \mathbf{I})$ and $\pi_1^\theta = \pi$, and which we train to approximate $\pi_s^{\text{VP}}$. We use $\pi_s^\theta$ to define our annealing path $\pi^n$ by $\pi^n = \pi_{s_n}^\theta$ for some choice of schedule $0 = s_0 < \ldots < s_N = 1$. Finally, for a fixed $K$, we define Diff-APT as APT with forward and backward accelerator respectively given by some $K$-step integrator of the time-reversal SDE (where we substitute $\pi_s^\theta$ for $\pi_s^{\text{VP}}$) and the forward SDE between the times $s_{n-1}$ and $s_n$; for further details, see Appendix D.3.

To provide a concrete example of $P_k^{n-1}$ and $Q_{k-1}^n$, consider the Euler-Maruyama discretisation with step-size $\delta_n = (s_n - s_{n-1})/K$ and interpolating times $s_n^k = s_{n-1} + k\delta_n$. The forward and backward accelerators are given by:

$$P_k^{n-1}(x_{k-1}, \mathrm{d}x_k) = \mathcal{N}(x_{k-1} + \delta_n \gamma_{1-s_n^{k-1}}[x_{k-1} + 2\nabla \log \pi_{s_n^{k-1}}^\theta(x_{k-1})], 2\delta_n \gamma_{1-s_n^{k-1}} \mathbf{I})$$

$$Q_{k-1}^n(x_k, \mathrm{d}x_{k-1}) = \mathcal{N}(x_k - \delta_n \gamma_{1-s_{n,k}} x_k, 2\delta_n \gamma_{1-s_{n,k}} \mathbf{I}),$$

and the resulting formula for the incremental weights $w_K^n$ is the same as Equation (5).

Regarding training, as we initially do not have access to samples from $\pi$, we iteratively train, using the score matching objective, on approximate samples from $\pi$ obtained by sampling from Diff-APT. Furthermore, we stress that a parallelised implementation of Diff-APT allows for generating a sample from $\pi$ with a similar effective cost as $K$ discretisation steps solving the time-reversal SDE.

## 6 EXPERIMENTS

In this section, we evaluate our proposed variants of APT on a variety of targets. We defer further experimental details to Appendix E.

### 6.1 COMPARISON OF ACCELERATION METHODS

We evaluate the three variants of APT introduced in Section 5: NF-APT, CMCD-APT and Diff-APT, against the baseline of classical PT using the geometric path with reference $\eta = \mathcal{N}(0, \mathbf{I})$, on a 40-mode Gaussian mixture model in 10 dimensions (GMM-10) (Midgley et al., 2023). Table 1 compares the performance of these methods in terms of round trips achieved and, in order to control for the additional computational cost required by APT, their *compute-normalised round trips*; for further details, see Appendix E.3. We choose to evaluate the different methods for $N = 6, 10, 30$ chains to

cover[4]: (1) the smallest $N$ where PT is able to obtain round trips (2) the regime $N \approx \Lambda$ where PT is stable ($\Lambda$ is the global barrier for PT), and (3) the regime where $N \gg \Lambda$ and PT is close to optimal.

We see all three APT variants substantially improve over PT with increased round trips as $N$ and $K$ increase, and even when controlling for the additional computation required by APT. For instance, in the small $N$ regime, (1) resulted in a 10x to 100x improvement and highlights the potential of APT for challenging problems where PT struggles and parallel chains are constrained. By contrast, when $N \gg \Lambda$ and the improvements over standard PT are less dramatic, they might not offset the cost of additional neural evaluations. Notably, CMCD with $K = 5$ and $N = 30$ outperforms the theoretical limit $T/(2 + 2\Lambda) \approx 5,349$ of round trips for classical PT (when $N \to \infty$) predicted by Theorem 2.

## 6.2 SCALING WITH DIMENSIONS

To understand the theoretical performance of $K$-step APT for a fixed number of chains when we scale the dimension $d$, we use the fact that the path of distributions $(\pi_s^{\text{VP}})_{s \in [0,1]}$ induced by a VP diffusion process initialised at GMM-$d$ is analytically tractable, in order to run $K$-step Diff-APT with the true diffusion path. We compare this against the same PT setup as previously. In Figure 2, we plot the resulting round trip rate and compute-normalised round trip rate of the different algorithms when we fix $N = 30$; for further details, see Appendix E.4. For all values of $d$, the round trip rate

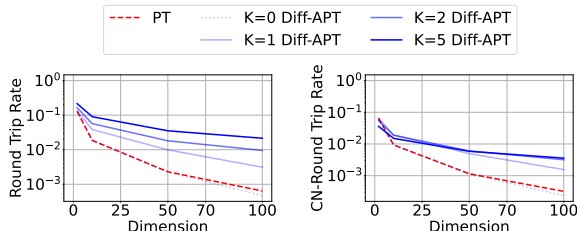

Figure 2: Round trip metrics for $K$-step Diff-APT ($K = 1, 2, 5$) and Diff-PT using the true diffusion path, and PT targeting GMM-$d$ for $d = 2, 10, 50, 100$ when using 30 chains. (Left) Round trip rate against $d$. (Right) Compute-normalised round trip rate against $d$.

of Diff-APT monotonically increases over PT as we increase $K$, with greater gains for larger values of $d$, demonstrating that the accelerated swap improves the communication of states over the standard PT swap. We see a substantial improvement in round trip rate (right) with acceleration compared to PT ($K = 0$) and a minor difference between algorithms as $K$ increases when normalised for compute (Right), suggesting the extra computation for APT is justified.

## 6.3 LOG-NORMALISING CONSTANT (FREE-ENERGY) ESTIMATION

As we described in Section 3.2, a by-product of Algorithm 1 is the estimation of change in free energy $\Delta F = -\log Z$. We compare free energy estimates from CMCD-APT and Diff-APT against PT on two targets: DoubleWell-4 (DW-4), a particle system in Cartesian coordinates; and ManyWell-32 (MW-32), a highly multi-modal density introduced by Midgley et al. (2023). Figure 3 presents box-plots of 30 free energy estimates for DW-4 and MW-32 using 1,000 samples each; for further details, see Appendix E.5.

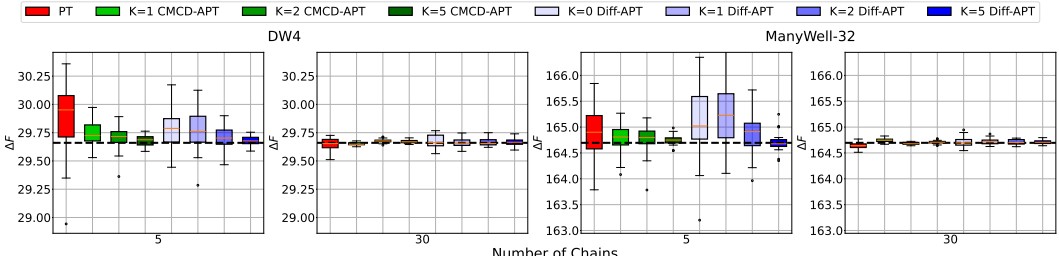

Figure 3: Estimates of $\Delta F$ for DW4 and ManyWell-32 by PT, CMCD-APT ($K = 1, 2, 5$) and Diff-APT ($K = 0, 1, 2, 5$) using 1,000 samples. Each box consists of 30 estimates. The black dashed lines denote the reference constant $\Delta F \approx 29.660$ estimated with PT using 60 chains and 100,000 samples and $\Delta F \approx 164.696$ from Midgley et al. (2023) for ManyWell-32.

---

[4]We refer to the discussion in Section 5 and 6.3 of Syed et al. (2024) for this classification of regimes.

Several key observations emerge: (1) Both CMCD-APT and Diff-APT exhibit markedly lower variance and bias than PT across both targets. (2) For CMCD/Diff-APT, the variance and bias reduce steadily as $K$ increases. (3) For both PT and CMCD/Diff-APT, the variance and bias markedly decrease as the number of chains $N$ increases, thereby empirically validating Proposition 1.

### 6.4 COMPARING APT WITH NEURAL SAMPLERS

A significant advantage of APT is its asymptotic consistency—unlike most neural samplers, the Metropolis correction ensures APT does not incur a bias if the neural transport is poorly trained. To demonstrate this advantage, for CMCD-APT and Diff-APT on DW-4, we take the trained forward transition kernels $P_k^{n-1}$ and map samples directly from the reference distribution to the target; we denote these respective new samplers by *CMCD* and *Diffusion* respectively and we compare the performance of these neural samplers to CMCD-APT and Diff-APT in Figure 4; for further details, see Appendix E.6. To aid with visualisation, we plot the *interatomic distances* of samples in DW-4.

As we can see, directly using the learnt neural samplers dramatically drops performance, especially for small $K$. Out of the two, diffusion performs better than CMCD but still misallocates probability mass across two modes. On the contrary, all variants of CMCD-APT and Diff-APT recover the correct mode weights and align closely with the ground truth.

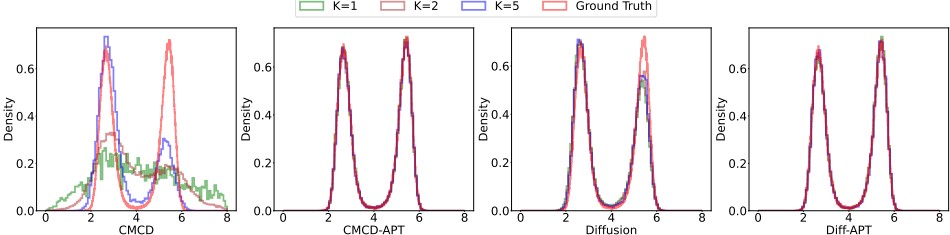

Figure 4: Interatomic distance $d_{ij}$ of 5,000 samples by CMCD, CMCD-APT, Diffusion, Diff-APT with 30 chains, $K = 1, 2, 5$ on DW4. We take 100,000 samples by PT with 60 chains as ground truth.

### 6.5 ALANINE DIPEPTIDE

Finally, to provide a realistic target on which to evaluate APT, we take the Boltzmann distribution of Alanine Dipeptide in Cartesian coordinates at 300K, and we compare PT against our best performing APT variant: CMCD-APT with $N = 4$. This is a small molecule with 22 atoms (66 dimensions in total) which is highly challenging to sample from due to many prohibitive energy barriers and high multimodality. Following the conventions of molecular dynamics, we take the reference

Table 2: PT versus CMCD-APT targeting Alanine Dipeptide. We use the same metrics as defined in Table 1.

| Method | $\hat{\Lambda}_K$ ($\downarrow$) | R ($\uparrow$) | CN-R ($\uparrow$) |
|---|---|---|---|
| CMCD-APT ($K = 1$) | 3.23 | 465 | **232.5** |
| CMCD-APT ($K = 2$) | 3.15 | 597 | 199 |
| CMCD-APT ($K = 5$) | **3.09** | **627** | 104.5 |
| PT | 3.38 | 199 | 99.5 |

distribution to be a tempered version of the target corresponding to 1200K. We run each algorithm for $T = 50,000$ iterations and report the results in Table 2; for further details, see Appendix E.7. We can see that CMCD-APT is able to provide significant acceleration even in this difficult setting with increased round trips, as well as computed-normalised round trips, over PT.

## 7 CONCLUSION

In this work, we formalise the framework of Accelerated Parallel Tempering which integrates neural transports to improve the efficiency of PT. However, several limitations, common to neural samplers, remain, including the burden of well optimising a neural network (Grenioux & Noble, 2026) which has a large impact on performance—i.e. a poorly trained neural transport may result in underperformance. Further, while our approach accelerates PT by increasing the round trip rate, it incurs additional neural network evaluations. Future work should therefore develop principled criteria for deciding when to rely on PT or APT for robustness. On a different note, an interesting avenue for future work is developing APT for inference-time control of generative models (He et al., 2025c), following the similar application of SMC for these purposes.

ACKNOWLEDGEMENTS

The authors are grateful to George Deligiannidis for helpful discussions. LZ and PP are supported by the EPSRC CDT in Modern Statistics and Statistical Machine Learning (EP/S023151/1). JH acknowledges support from the University of Cambridge Harding Distinguished Postgraduate Scholars Programme. SS acknowledges support from the EPSRC CoSInES grant (EP/R034710/1) and the NSERC Postdoctoral Fellowship.

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

## A    FURTHER DETAILS ON ROUND TRIPS

Consider the APT Markov chain $\mathbf{X}_t = (X_t^0, \ldots, X_t^N)$ constructed by Algorithm 1. Assume there are $m = 0, \ldots, N$ parallel threads, each storing a component of $\mathbf{X}_t$ at time $t$. During the communication step, it is equivalent to swap indices rather than states. This has computational advantages in a distributed implementation of PT since swapping states across machines incurs a cost of $O(d)$ compared to the $O(1)$ cost of swapping indices. By tracking how the indices are shuffled, we can recover the PT state and also track the communication between reference and target through the dynamics of the permuted indices. See (Syed et al., 2022; Surjanovic et al., 2023, Algorithm 5) for details on implementing PT with a distributed implementation.

Summarising Syed et al. (2022), we define the *index process* $(\mathbf{I}_t, \boldsymbol{\epsilon}_t)$ where $\mathbf{I}_t = (I_t^0, \ldots, I_t^N)$ is a sequence of permutations of $[N] = \{0, \ldots, N\}$ that tracks the underlying communication of states in $\mathbf{X}_t$, and $\boldsymbol{\epsilon}_t = (\epsilon_t^0, \ldots, \epsilon_t^N)$ with $\epsilon_t^m \in \{-1, 1\}$ tracks the direction of the swap proposal on machine $m$.

We initialize $I_0^m = m$ and $\epsilon_0^m = -1$ if $m$ is even and $\epsilon_0^m = 1$ if $m$ is odd. The subsequent values of $\mathbf{I}_t$ are then determined by the swap moves. At iteration $t$ of PT, we apply the same swaps to the components of $\mathbf{I}_t$ that are proposed and accepted during the communication phase. As discussed in Syed et al. (2022), for non-reversible communication, $I_t^m$ and $\epsilon_t^m$ satisfy the recursion:

$$I_t^m = \begin{cases} I_{t-1}^m + \epsilon_{t-1}^m, & \text{if } S_t^{I_{t-1}^m \vee I_{t-1}^m + \epsilon_{t-1}^m} = 1 \\ I_{t-1}^m, & \text{if } S_t^{I_{t-1}^m \vee I_{t-1}^m + \epsilon_{t-1}^m} = 0 \end{cases},$$

$$\epsilon_t^m = \begin{cases} \epsilon_{t-1}^m, & \text{if } I_t^m = I_{t-1}^m + \epsilon_{t-1}^m \\ -\epsilon_{t-1}^m, & \text{if } I_t^m = I_{t-1}^m \end{cases}.$$

Here $S_t^n = 1$ if the $n$-th swap at time $t$ was accepted and $S_t^n = 0$ otherwise.

Hence, for a realization of $\mathbf{X}_t$ with $T$ steps, we define the *round trips for index* $n \in [N]$ as the number of times the index $n$ in $\mathbf{I}_t$ completes the journey from the $0$-th component to the $N$-th component and then back to the $0$-th component—i.e., completes the round trip from the reference to the target and back again.

For $m \in [N]$, let $T_{\downarrow,0}^m := \inf\{t : (I_t^m, \epsilon_t^m) = (0, -1)\}$ and for $j \geq 1$ define $T_{\uparrow,j}^m$ and $T_{\downarrow,j}^m$ recursively as follows:

$$T_{\uparrow,j}^m = \inf\{t > T_{\downarrow,j-1}^m : (I_t^m, \epsilon_t^m) = (N, 1)\},$$
$$T_{\downarrow,j}^m = \inf\{t > T_{\uparrow,j}^m : (I_t^m, \epsilon_t^m) = (0, -1)\}.$$

Notably, $T_{\downarrow,j}^m$ represents the $j$-th time the index for machine $m$ has traversed from $0$ to $N$ and back to $0$, thus defining a *round trip*. A round trip indicates that a sample on machine $m$ from the reference propagated a reference sample to the target independent of previously visited target states on the same machine. Let $R_T^m = \max\{j : T_{\downarrow,j}^m \leq T\}$ denote the number of round trips that occur on machine $m$ by iteration $T$. The overall *round trips* count is then defined as the sum of round trips over all index values: $R_T = \sum_{m=0}^N R_T^m$. Finally, the empirical round trip rate $\tau_T = R_T/T$ represents the fraction of PT iterations during which a round trip occurred, and our objective is to maximise the expected round trip rate $\tau$ as $T \to \infty$:

$$\tau := \lim_{T \to \infty} \mathbb{E}[\tau_T] = \lim_{T \to \infty} \frac{\mathbb{E}[R_T]}{T}.$$

## B    THEORETICAL ANALYSIS OF ACCELERATED PT

### B.1    PROOF OF THEOREM 1

#### B.1.1    PROOF OF ERGODICITY (THEOREM 1, PART 1)

*Proof.* At stationary, we have $(X_t^{n-1}, X_t^n) \sim \pi^{n-1} \otimes \pi^n$. We next simulate the two paths $\vec{X}_{t,0:K}^{n-1} = (\vec{X}_{t,0}^{n-1}, \ldots, \vec{X}_{t,K}^{n-1})$ and $\overleftarrow{X}_{t,0:K}^n = (\overleftarrow{X}_{t,0}^n, \ldots, \overleftarrow{X}_{t,K}^n)$ as described in Section 3 and calculate the acceptance probability

$$\alpha = \alpha_K^n(\vec{X}_{t,0:K}^{n-1}, \overleftarrow{X}_{t,0:K}^n)$$

given in Section 3.2. The new states $\hat{X}_t^{n-1}, \hat{X}_t^n$ are determined by

$$(\hat{X}_t^{n-1}, \hat{X}_t^n) = \begin{cases} (\overleftarrow{X}_{t,0}^n, \overrightarrow{X}_{t,K}^{n-1}) & \text{with probability } \alpha, \\ (\overrightarrow{X}_{t,0}^{n-1}, \overleftarrow{X}_{t,K}^n) & \text{with probability } 1-\alpha. \end{cases}$$

We would like to prove that the swap operation keeps the target distribution invariant, that is, for any real-valued test function $\varphi$ we have

$$\mathbb{E}\left[\varphi(\hat{X}_t^{n-1}, \hat{X}_t^n)\right] = \mathbb{E}\left[\varphi(X_t^{n-1}, X_t^n)\right]. \tag{6}$$

One way to do this is to consider the extended target distribution

$$\overrightarrow{X}_{t,0:K}^{n-1} \otimes \overleftarrow{X}_{t,0:K}^n \sim \mathbb{P}_K^{n-1}(\mathrm{d}x_{0:K}^{n-1}) \otimes \mathbb{Q}_K^n(\mathrm{d}x_{0:K}^n)$$

and the involution

$$\mathfrak{T}(x_{0:K}, y_{0:K}) = (y_{0:K}, x_{0:K})$$

and to apply an instance of MCMC algorithms with a deterministic proposal (Tierney, 1998), here given by $\mathfrak{T}$. For completeness we give a self-contained proof. Write

$$\mathbb{E}\left[\varphi(\hat{X}_t^{n-1}, \hat{X}_t^n)\right] = \mathbb{E}\left[\varphi(\overleftarrow{X}_{t,0}^n, \overrightarrow{X}_{t,K}^{n-1})\alpha + \varphi(\overrightarrow{X}_{t,0}^{n-1}, \overleftarrow{X}_{t,K}^n)(1-\alpha)\right]$$

$$= \mathbb{E}\left[\varphi(X_t^{n-1}, X_t^n)\right] - \mathbb{E}\left[\varphi(\overrightarrow{X}_{t,0}^{n-1}, \overleftarrow{X}_{t,K}^n)\alpha\right] +$$

$$+ \mathbb{E}\left[\varphi(\overleftarrow{X}_{t,0}^n, \overrightarrow{X}_{t,K}^{n-1})\alpha\right].$$

To arrive at Equation (6) we need to show that

$$\mathbb{E}_{\mathbb{P}\otimes\mathbb{Q}}\left[\varphi \circ \psi(\overrightarrow{X}_{t,0:K}^{n-1}, \overleftarrow{X}_{t,0:K}^n)\alpha\right] = \mathbb{E}_{\mathbb{P}\otimes\mathbb{Q}}\left[\varphi \circ \psi(\overleftarrow{X}_{t,0:K}^n, \overrightarrow{X}_{t,0:K}^{n-1})\alpha\right] \tag{7}$$

where $\psi(x_{0:K}, y_{0:K}) := (x_0, y_K)$ and the notation $\mathbb{E}_{\mathbb{P}\otimes\mathbb{Q}}$ means that the expectation is taken with respect to $\mathbb{P}_K^{n-1}(\mathrm{d}x_{0:K}^{n-1}) \otimes \mathbb{Q}_K^n(\mathrm{d}x_{0:K}^n)$. Noting that

$$\alpha = \alpha_K^n(\overrightarrow{X}_{t,0:K}^{n-1}, \overleftarrow{X}_{t,0:K}^n) = 1 \wedge \frac{\mathrm{d}\mathbb{Q}/\mathrm{d}\mathbb{P}(\overrightarrow{X}_{t,0:K}^{n-1})}{\mathrm{d}\mathbb{Q}/\mathrm{d}\mathbb{P}(\overleftarrow{X}_{t,0:K}^n)}$$

we write

$$\mathbb{E}_{\mathbb{P}\otimes\mathbb{Q}}\left[\varphi \circ \psi(\overrightarrow{X}_{t,0:K}^{n-1}, \overleftarrow{X}_{t,0:K}^n)\alpha\right]$$

$$= \mathbb{E}_{\mathbb{Q}\otimes\mathbb{P}}\left[\varphi \circ \psi(\overrightarrow{X}_{t,0:K}^{n-1}, \overleftarrow{X}_{t,0:K}^n)\alpha \frac{\mathrm{d}\mathbb{P}\otimes\mathbb{Q}}{\mathrm{d}\mathbb{Q}\otimes\mathbb{P}}(\overrightarrow{X}_{t,0:K}^{n-1}, \overleftarrow{X}_{t,0:K}^n)\right]$$

$$= \mathbb{E}_{\mathbb{Q}\otimes\mathbb{P}}\left[\varphi \circ \psi(\overrightarrow{X}_{t,0:K}^{n-1}, \overleftarrow{X}_{t,0:K}^n)\alpha \frac{\mathrm{d}\mathbb{Q}/\mathrm{d}\mathbb{P}(\overleftarrow{X}_{t,0:K}^n)}{\mathrm{d}\mathbb{Q}/\mathrm{d}\mathbb{P}(\overrightarrow{X}_{t,0:K}^{n-1})}\right]$$

$$= \mathbb{E}_{\mathbb{Q}\otimes\mathbb{P}}\left[\varphi \circ \psi(\overrightarrow{X}_{t,0:K}^{n-1}, \overleftarrow{X}_{t,0:K}^n)\left\{\frac{\mathrm{d}\mathbb{Q}/\mathrm{d}\mathbb{P}(\overleftarrow{X}_{t,0:K}^n)}{\mathrm{d}\mathbb{Q}/\mathrm{d}\mathbb{P}(\overrightarrow{X}_{t,0:K}^{n-1})} \wedge 1\right\}\right]$$

$$= \mathbb{E}_{\mathbb{P}\otimes\mathbb{Q}}\left[\varphi \circ \psi(\overleftarrow{X}_{t,0:K}^n, \overrightarrow{X}_{t,0:K}^{n-1})\left\{\frac{\mathrm{d}\mathbb{Q}/\mathrm{d}\mathbb{P}(\overrightarrow{X}_{t,0:K}^{n-1})}{\mathrm{d}\mathbb{Q}/\mathrm{d}\mathbb{P}(\overleftarrow{X}_{t,0:K}^n)} \wedge 1\right\}\right]$$

$$= \mathbb{E}_{\mathbb{P}\otimes\mathbb{Q}}\left[\varphi \circ \psi(\overleftarrow{X}_{t,0:K}^n, \overrightarrow{X}_{t,0:K}^{n-1})\alpha\right].$$

Thus Equation (7) is established.

Finally, the accelerated-PT Markov chain is aperiodic because the swaps can be rejected. The chain is clearly irreducible if each exploration kernel is irreducible. (In fact it can be proved, using more complicated arguments, that the chain is irreducible if the exploration kernel for the reference is irreducible and, for all $n$, the two distributions $\mathbb{P}_K^{n-1}$ and $\mathbb{Q}_K^n$ are mutually absolutely continuous.) By Roberts & Rosenthal (2004, Theorem 4, Fact 5), the Markov chain is ergodic and in particular the law of large numbers holds. $\qquad\square$

B.1.2   PROOF OF REJECTION RATE (THEOREM 1, PART 2)

*Proof.* The rejection rate at stationary is defined as

$$r(\mathbb{P}_K^{n-1}, \mathbb{Q}_K^n) := \mathbb{E}_{\mathbb{P}\otimes\mathbb{Q}}\left[1 - \alpha_K^n(\vec{X}_{t,0:K}^{n-1}, \overleftarrow{X}_{t,0:K}^n)\right] \qquad (8)$$

where $\mathbb{E}_{\mathbb{P}\otimes\mathbb{Q}}$ is a shorthand for the expectation under $\mathbb{P}_K^{n-1} \otimes \mathbb{Q}_K^n$ and the acceptance ratio $\alpha_K^n$ is given by

$$\alpha_K^n(\vec{X}_{t,0:K}^{n-1}, \overleftarrow{X}_{t,0:K}^n) := 1 \wedge \frac{d\mathbb{Q}_K^n/d\mathbb{P}_K^{n-1}(\vec{X}_{t,0:K}^{n-1})}{d\mathbb{Q}_K^n/d\mathbb{P}_K^{n-1}(\overleftarrow{X}_{t,0:K}^n)} = 1 \wedge \frac{d(\mathbb{Q}_K^n \otimes \mathbb{P}_K^{n-1})}{d(\mathbb{P}_K^{n-1} \otimes \mathbb{Q}_K^n)}(\vec{X}_{t,0:K}^{n-1}, \overleftarrow{X}_{t,0:K}^n). \qquad (9)$$

From Equation (8), Equation (9), and Lemma 1 below we have

$$r(\mathbb{P}_K^{n-1}, \mathbb{Q}_K^n) = \|\mathbb{P}_K^{n-1} \otimes \mathbb{Q}_K^n - \mathbb{Q}_K^n \otimes \mathbb{P}_K^{n-1}\|_{\text{TV}}.$$

$\square$

**Lemma 1.** *Let $\mu_1$ and $\mu_2$ be two mutually absolutely continuous probability measures. Then*

$$\|\mu_1 - \mu_2\|_{\text{TV}} = \mathbb{E}_{\mu_1}\left[1 - \min\left(1, \frac{d\mu_2}{d\mu_1}\right)\right].$$

*Proof.* Recall the definition of the TV distance

$$\|\mu_1 - \mu_2\|_{\text{TV}} := \sup_{h:\mathcal{X}\to[0,1]} |\mathbb{E}_{\mu_1}[h] - \mathbb{E}_{\mu_2}[h]|.$$

First, we remark that the absolute value in the definition can be omitted since

$$\|\mu_1 - \mu_2\|_{\text{TV}} = \sup_{h:\mathcal{X}\to[0,1]} \max\left(\mathbb{E}_{\mu_1}[h] - \mathbb{E}_{\mu_2}[h], \mathbb{E}_{\mu_2}[h] - \mathbb{E}_{\mu_1}[h]\right)$$

$$= \sup_{h:\mathcal{X}\to[0,1]} \max\left(\mathbb{E}_{\mu_1}[h] - \mathbb{E}_{\mu_2}[h], \mathbb{E}_{\mu_1}[1-h] - \mathbb{E}_{\mu_2}[1-h]\right)$$

$$= \sup_{h:\mathcal{X}\to[0,1]} \mathbb{E}_{\mu_1}[h] - \mathbb{E}_{\mu_2}[h].$$

Therefore

$$\|\mu_1 - \mu_2\|_{\text{TV}} = \sup_{h:\mathcal{X}\to[0,1]} \mathbb{E}_{\mu_1}\left[h \cdot \left(1 - \frac{d\mu_2}{d\mu_1}\right)\right]$$

$$\leq \sup_{h:\mathcal{X}\to[0,1]} \mathbb{E}_{\mu_1}\left[h \cdot \left(1 - \min\left(1, \frac{d\mu_2}{d\mu_1}\right)\right)\right] \qquad (10)$$

$$\leq \mathbb{E}_{\mu_1}\left[1 - \min\left(1, \frac{d\mu_2}{d\mu_1}\right)\right].$$

On the other hand, let $B := \{x \in X \text{ such that } d\mu_2/d\mu_1(x) \leq 1\}$ and put $h^*(x) := \mathbb{1}_B(x)$. Using

$$h^*(x) = 1 - \min\left(1, \frac{d\mu_2}{d\mu_1}(x)\right) + \frac{d\mu_2}{d\mu_1}(x)\mathbb{1}_B(x)$$

write

$$\|\mu_1 - \mu_2\|_{\text{TV}} \geq \mathbb{E}_{\mu_1}[h^*] - \mathbb{E}_{\mu_2}[h^*]$$

$$= \mathbb{E}_{\mu_1}\left[1 - \min\left(1, \frac{d\mu_2}{d\mu_1}\right)\right] + \mathbb{E}_{\mu_1}\left[\frac{d\mu_2}{d\mu_1}\mathbb{1}_B\right] - \mathbb{E}_{\mu_2}[\mathbb{1}_B] \qquad (11)$$

$$= \mathbb{E}_{\mu_1}\left[1 - \min\left(1, \frac{d\mu_2}{d\mu_1}\right)\right].$$

Combining Equation (10) and Equation (11) we get the desired result. $\square$

### B.1.3 PROOF OF PROPOSITION 1

*Proof.* We will show the consistency of the expectation and free energy estimator separately. By Theorem 1, the PT chain $\mathbf{X}_t = (X_t^0, \ldots, X_t^N)$ is ergodic with stationary distribution $\pi^0 \otimes \cdots \otimes \pi^N$. By the ergodic theorem, we have

$$\hat{\pi}_T^n[f] = \frac{1}{T} \sum_{t=1}^T f(X_t^n) \xrightarrow[T \to \infty]{a.s.} \pi^n[f].$$

We will now show the consistency of the free energy estimator. By definition of $W_K^n$ in Equation (3), we can write $\Delta F_n$ in terms of expectations of $W_K^n$ and the Radon-Nikodym derivative between $\mathbb{P}_K^{n-1}$ and $\mathbb{Q}_K^n$ respectively:

$$\frac{Z_n}{Z_{n-1}} \frac{\mathrm{d}\mathbb{Q}_K^n}{\mathrm{d}\mathbb{P}_K^{n-1}} = w_K^n, \quad \frac{Z_{n-1}}{Z_n} \frac{\mathrm{d}\mathbb{P}_K^{n-1}}{\mathrm{d}\mathbb{Q}_K^n} = \frac{1}{w_K^n}.$$

By taking expectations of the left and right expressions with respect to $\mathbb{P}_K^{n-1}$ and $\mathbb{Q}_K^n$ respectively, and taking the product over $n = 1, \ldots, N$, we obtain the following expressions for $Z$ and $Z^{-1}$ respectively:

$$Z = \prod_{n=1}^N \mathbb{P}_K^{n-1}[w_K^n], \quad Z^{-1} = \prod_{n=1}^N \mathbb{Q}_K^n\left[(w_K^n)^{-1}\right]. \tag{12}$$

Since $\mathbf{X}_t$ is ergodic, $\vec{Z}_T$ and $\overleftarrow{Z}_T$ are consistent estimators for $Z$ respectively. Therefore $\hat{Z}_T = (\vec{Z}_T \overleftarrow{Z}_T)^{1/2}$ is consistent as well. Finally, note that if $\mathbb{P}_K^{n-1} = \mathbb{Q}_K^n$, then $w_K^n(x_{0:K}) \stackrel{a.s.}{=} Z_n/Z_{n-1}$, and hence $\vec{Z}_T = \overleftarrow{Z}_T \stackrel{a.s.}{=} Z$. □

### B.2 PROOF OF PROPOSITION 2

Let $S_t^n$ be the indicator random variable with $S_t^n = 1$ if the $n$-th swap is proposed and accepted at iteration $t$, and $S_t^n = 0$ otherwise. By Assumption 1, the random variables $S_t^n$ are i.i.d. in $t$ with $S_1^n, S_2^n, \ldots \stackrel{d}{=} \text{Bernoulli}(s_n)$, where $s_n := 1 - r(\mathbb{P}^{n-1}, \mathbb{Q}^n)$. This implies that the index processes $(I_t^m, \epsilon_t^m)$ form Markov chains on $\{0, \ldots, N\} \times \{-1, 1\}$ with initial conditions $I_0^m = m$ and $\epsilon_0^m = -1$ when $m$ is even, $\epsilon_0^m = 1$ when $m$ is odd. For $t \geq 1$, each process satisfies the following recursion:

$$I_t^m \sim \begin{cases} I_{t-1}^m + \epsilon_{t-1}^m, & \text{with probability } s_{I_{t-1}^m \vee (I_{t-1}^m + \epsilon_{t-1}^m)} \\ I_{t-1}^m, & \text{with probability } 1 - s_{I_{t-1}^m \vee (I_{t-1}^m + \epsilon_{t-1}^m)} \end{cases}, \tag{13}$$

$$\epsilon_t^m = \begin{cases} \epsilon_{t-1}^m, & \text{if } I_t^m = I_{t-1}^m + \epsilon_{t-1}^m \\ -\epsilon_{t-1}^m, & \text{if } I_t^m = I_{t-1}^m \end{cases}. \tag{14}$$

The remainder of the proof follows identically to (Syed et al., 2022, Corollary 2).

### B.3 PROOF OF PROPOSITION 3

We first spell out the full statement of Proposition 3.

**Proposition.** Consider two SDEs

$$\vec{X}^{n-1}(0) \sim \pi^{n-1}, \quad d\vec{X}^{n-1}(u) = f(u, \vec{X}^{n-1}(u))du + \sigma dB_u \text{ for } u \in [0,1] \tag{15}$$

$$\overleftarrow{X}^n(1) \sim \pi^n, \quad d\overleftarrow{X}^n(u) = b(u, \overleftarrow{X}^n(u))dt + \sigma dB'_u \text{ for } u \in [0,1]; \tag{16}$$

where the second equation is integrated backwards in time. Let $\mathbb{P}_\infty^{n-1}(\mathrm{d}x_\infty)$ and $\mathbb{P}_K^{n-1}(\mathrm{d}x_K)$ be respectively the full and the Euler-discretised path measures of Equation (15), in particular $x_\infty = (x_\infty(u))_{u \in [0,1]} \in \mathcal{C}[0,1]$ and $x_K = [x_K(0), x_K(1/K), \ldots, x_K(1)] \in \mathbb{R}^{K+1}$. Similarly let $\mathbb{Q}_\infty^n(\mathrm{d}x_\infty)$ and $\mathbb{Q}_K^n(\mathrm{d}x_K)$ be respectively the full and the Euler-discretised path measures of Equation (16). Assume that $f(u, \cdot)$ and $f(\cdot, x)$ are Lipschitz for all $u$ and $x$ with a global constant; and that the same holds for $b$. Moreover, suppose that there exists a $G \geq 0$ such that $||f(u,x)|| + ||b(u,x)|| \leq G(1 + ||x||)$. Then

$$\lim_{K \to \infty} r(\mathbb{P}_K^{n-1}, \mathbb{Q}_K^n) = r(\mathbb{P}_\infty^{n-1}, \mathbb{Q}_\infty^n) \tag{17}$$

and

$$r(\mathbb{P}_K^{n-1}, \mathbb{Q}_K^n) \le r(\mathbb{P}_\infty^{n-1}, \mathbb{Q}_\infty^n) + \mathcal{O}(\frac{1}{\sqrt{K}}) \tag{18}$$

where $r(p,q) = \|p \times q - q \times p\|_{\mathrm{TV}}$.

*Proof.* Let $\mathbb{P}_{\infty|K}^{n-1}$ and $\mathbb{Q}_{\infty|K}^n$ be the restrictions of the path measures $\mathbb{P}_\infty^{n-1}$ and $\mathbb{Q}_\infty^n$ to the time discretisation points. (As such, $\mathbb{P}_{\infty|K}^{n-1}$ and $\mathbb{Q}_{\infty|K}^n$ are defined on the same space as $\mathbb{P}_K^{n-1}$ and $\mathbb{Q}_K^n$.) We have

$$|r(\mathbb{P}_K^{n-1}, \mathbb{Q}_K^n) - r(\mathbb{P}_{\infty|K}^{n-1}, \mathbb{Q}_{\infty|K}^n)| \le 2\left(\|\mathbb{P}_K^{n-1} - \mathbb{P}_{\infty|K}^{n-1}\|_{\mathrm{TV}} + \|\mathbb{Q}_K^n - \mathbb{Q}_{\infty|K}^n\|_{\mathrm{TV}}\right)$$
$$\le \mathcal{O}(1/\sqrt{K}) \tag{19}$$

where the first inequality is elementary and the second follows from Proposition 4 below. By data processing inequality

$$r(\mathbb{P}_{\infty|K}^{n-1}, \mathbb{Q}_{\infty|K}^n) \le r(\mathbb{P}_\infty^{n-1}, \mathbb{Q}_\infty^n). \tag{20}$$

Equation (19) and Equation (20) establish Equation (18). On the other hand, putting

$$R_K(x_k, y_k) := \frac{d(\mathbb{Q}_K^n \times \mathbb{P}_K^{n-1})}{d(\mathbb{P}_K^{n-1} \times \mathbb{Q}_K^n)}(x_k, y_k);$$

$$R_\infty(x_\infty, y_\infty) := \frac{d(\mathbb{Q}_\infty^n \times \mathbb{P}_\infty^{n-1})}{d(\mathbb{P}_\infty^{n-1} \times \mathbb{Q}_\infty^n)}(x_\infty, y_\infty);$$

$$\varphi(\alpha) := 1 - \min(1, \alpha) \tag{21}$$

we have by Lemma 1

$$r(\mathbb{P}_K^{n-1}, \mathbb{Q}_K^n) = \mathbb{E}_{\mathbb{P}_K^{n-1} \times \mathbb{Q}_K^n}[\varphi \circ R_K]. \tag{22}$$

Since $\varphi(\alpha) \in [0,1]$ the definition of the total variation distance implies

$$\left|\mathbb{E}_{\mathbb{P}_K^{n-1} \times \mathbb{Q}_K^n}[\varphi \circ R_K] - \mathbb{E}_{\mathbb{P}_{\infty|K}^{n-1} \times \mathbb{Q}_{\infty|K}^n}[\varphi \circ R_K]\right| \le \mathcal{O}(1/\sqrt{K}). \tag{23}$$

In addition define the discretisation operator

$$D_K : \mathcal{C}[0,1] \to (\mathbb{R}^d)^{K+1};$$
$$x_\infty \mapsto x_{\infty|K} = [x_\infty(0), x_\infty(1/K), \ldots, x_\infty(1)]$$

which takes a continuous path $x_\infty$ and extract its values at $K+1$ discretisation points. Let $\tilde{D}_K(x_\infty, y_\infty) = (D_K(x_\infty), D_K(y_\infty))$. Then

$$\mathbb{E}_{\mathbb{P}_{\infty|K}^{n-1} \times \mathbb{Q}_{\infty|K}^n}[\varphi \circ R_K] = \mathbb{E}_{\mathbb{P}_\infty^{n-1} \times \mathbb{Q}_\infty^n}[\varphi \circ R_K \circ \tilde{D}_K]$$
$$\overset{K \to \infty}{\longrightarrow} \mathbb{E}_{\mathbb{P}_\infty^{n-1} \times \mathbb{Q}_\infty^n}[\varphi \circ R_\infty] = r(\mathbb{P}_\infty^{n-1}, \mathbb{Q}_\infty^n) \tag{24}$$

by the dominated convergence theorem, taking into account the pathwise convergence of the Euler-Maruyama schemes (Kloeden & Neuenkirch (2007); Gyöngy (1998), see also Berner et al. (2025, Lemma B.7.a)). More precisely, the function $\varphi$ defined in Equation (21) is continuous and bounded between 0 and 1, and

$$R_K \circ \tilde{D}_K(x_\infty, y_\infty) = \frac{d\mathbb{Q}_K^n}{d\mathbb{P}_K^{n-1}}(D_K x_\infty)\frac{d\mathbb{P}_K^{n-1}}{d\mathbb{Q}_K^n}(D_K y_\infty) =$$

$$\left[\frac{\pi^n(x_\infty(1))\prod_{k=1}^K \mathcal{N}(x_\infty(\frac{k-1}{K})|x_\infty(\frac{k}{K})+\frac{1}{K}b(\frac{k}{K},x(\frac{k}{K}));\frac{\sigma^2}{K})}{\pi^{n-1}(x_\infty(0))\prod_{k=0}^{K-1} \mathcal{N}(x_\infty(\frac{k+1}{K})|x_\infty(\frac{k}{K})+\frac{1}{K}f(\frac{k}{K},x(\frac{k}{K}));\frac{\sigma^2}{K})}\right] \times$$

$$\times \left[\frac{\pi^{n-1}(y_\infty(0))\prod_{k=0}^{K-1} \mathcal{N}(y_\infty(\frac{k+1}{K})|y_\infty(\frac{k}{K})+\frac{1}{K}f(\frac{k}{K},y(\frac{k}{K}));\frac{\sigma^2}{K})}{\pi^n(y_\infty(1))\prod_{k=1}^K \mathcal{N}(y_\infty(\frac{k-1}{K})|y_\infty(\frac{k}{K})+\frac{1}{K}b(\frac{k}{K},x(\frac{k}{K}));\frac{\sigma^2}{K})}\right].$$

When $K \to \infty$, the term in the first big square bracket converges to $d\mathbb{Q}_K^n / d\mathbb{P}_K^{n-1}(x_\infty)$ and that in the second converges to $d\mathbb{P}_K^{n-1} / d\mathbb{Q}_K^n(y_\infty)$, establishing Equation (24).

Finally, Equation (22), Equation (23), and Equation (24) entail Equation (17), finishing the proof. □

**Proposition 4.** *Let $(X_t)_{t \in [0,1]}$ be the solution of an SDE of the form*

$$dX_t = b(t, X_t)dt + \sigma dB_t, \tag{25}$$

*where we assume that $b(t, \cdot)$ is L-Lipschitz continuous for all $t$ and that $b(\cdot, x)$ is B-Lipschitz continuous for all $x$. We also assume a linear growth condition of the form $\|b(t, x)\| \leq G(1 + \|x\|)$. Consider the K-step Euler-Maruyama discretization $\hat{X}_{t_0}, \hat{X}_{t_1}, \ldots, \hat{X}_{t_K}$ where $t_k = k/K$. Let $\mathbb{P}_\infty$ be the path-measure of the process $(X_t)_{t \in [0,1]}$, $\mathbb{P}_K^*$ be the law of $X_{t_0}, \ldots, X_{t_K}$, and $\mathbb{P}_K$ be the law of $\hat{X}_{t_0}, \ldots, \hat{X}_{t_K}$. Then*

$$\|\mathbb{P}_K^* - \mathbb{P}_K\|_{\mathrm{TV}} \in \mathcal{O}\left(\frac{1}{\sqrt{K}}\right).$$

*Proof.* Consider the continuous-time extension $(\hat{X}_t)_{t \in [0,1]}$ of the $K$-step Euler-Maruyama discretization of the SDE in Equation (25) given by

$$d\hat{X}_t = b(t_k, \hat{X}_{t_k})dt + \sigma dB_t, \quad t \in [t_k, t_{k+1}).$$

We use $\hat{\mathbb{P}}_\infty = \hat{\mathbb{P}}_\infty^{(K)}$ to denote its path-measure. By Pinsker's inequality

$$\|\mathbb{P}_K^* - \mathbb{P}_K\|_{\mathrm{TV}} \leq \sqrt{\frac{1}{2}\mathrm{KL}(\mathbb{P}_K^* \| \mathbb{P}_K)}.$$

By the data-processing inequality and Girsanov's theorem, we have that

$$\mathrm{KL}(\mathbb{P}_K^* \| \mathbb{P}_K) \leq \mathrm{KL}(\mathbb{P}_\infty \| \hat{\mathbb{P}}_\infty) = \frac{1}{2\sigma^2} \sum_{k=0}^{K-1} \int_{t_k}^{t_{k+1}} \mathbb{E}\|b(t, X_t) - b(t_k, X_{t_k})\|^2 dt.$$

By the Lipschitz continutity of $b$, we have the following bound

$$\mathbb{E}\|b(t, X_t) - b(t_k, X_{t_k})\|^2 \leq 2\mathbb{E}\|b(t, X_t) - b(t, X_{t_k})\|^2 + 2\mathbb{E}\|b(t, X_{t_k}) - b(t_k, X_{t_k})\|^2$$
$$\leq 2L^2 \mathbb{E}\|X_t - X_{t_k}\|^2 + 2B^2(t - t_k)^2.$$

By the Cauchy-Schwarz inequality, the fact that $B_t - B_{t_k} \sim \mathcal{N}(0, (t - t_k)I_d)$ and the linear growth assumption, we obtain

$$\mathbb{E}\|X_t - X_{t_k}\|^2 = \mathbb{E}\left\|\int_{t_k}^t b(s, X_s)ds + \sigma(B_t - B_{t_k})\right\|^2$$

$$\leq 2(t - t_k) \int_{t_k}^t \mathbb{E}\|b(s, X_s)\|^2 ds + 2d\sigma^2(t - t_k)$$

$$\leq 2G^2(t - t_k) \int_{t_k}^t \mathbb{E}(1 + \|X_s\|)^2 ds + 2d\sigma^2(t - t_k)$$

$$\leq 4G^2(t - t_k) \int_{t_k}^t (1 + \mathbb{E}\|X_s\|^2)ds + 2d\sigma^2(t - t_k)$$

It is well known that under the assumptions in the proposition, there exists a constant $M$ (independent of $K$) such that $\mathbb{E}\|X_t\|^2 \leq M$ for all $t$. Putting all of this together, we have shown that

$$\mathbb{E}\|b(t, X_t) - b(t_k, X_{t_k})\|^2 \leq 8G^2 L^2(1 + M)(t - t_k)^2 + 4d\sigma^2 L^2(t - t_k) + 2B^2(t - t_k)^2.$$

Summing over $k$ and integrating, this yields

$$\frac{1}{2\sigma^2} \sum_{k=0}^{K-1} \int_{t_k}^{t_{k+1}} \mathbb{E}\|b(t, X_t) - b(t_k, X_{t_k})\|^2 dt$$

$$\leq \frac{1}{2\sigma^2} \sum_{k=0}^{K-1} \frac{8}{3}G^2 L^2(1 + M)(t_{k+1} - t_k)^3 + 2d\sigma^2 L^2(t_{k+1} - t_k)^2 + \frac{2}{3}B^2(t_{k+1} - t_k)^3$$

$$\leq \frac{4}{3\sigma^2}G^2 L^2(1 + M)\frac{1}{K^2} + dL^2\frac{1}{K} + \frac{1}{3\sigma^2}B^2\frac{1}{K^2}.$$

Therefore

$$\|\mathbb{P}_K^* - \mathbb{P}_K\|_{\mathrm{TV}} \in \mathcal{O}(1/\sqrt{K}).$$

$\square$

We verify this result in practice by plotting the TV distance between $\mathbb{P}_K$ and $\mathbb{P}_\infty$ for the Ornstein–Uhlenbeck process in 1 dimension.

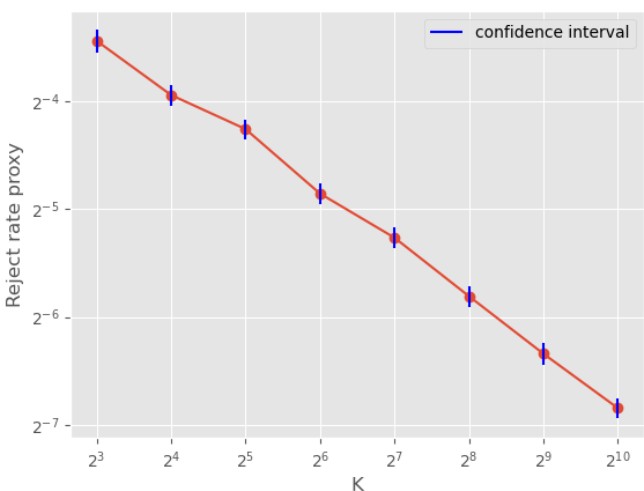

### B.4 FURTHER DETAILS ON SCALING WITH PARALLEL CHAINS

#### B.4.1 REGULARITY ASSUMPTIONS

Suppose $\pi^\beta(x) = \exp(-U^\beta(x))/Z_\beta$, where $Z_\beta = \int_{\mathcal{X}} \exp(-U^\beta(x))\mathrm{d}x$. For $\boldsymbol{\beta} = (\beta, \beta')$, suppose $\mathbb{P}_K^{\boldsymbol{\beta}}(\mathrm{d}x_{0:K})$ and $\mathbb{Q}_K^{\boldsymbol{\beta}}(\mathrm{d}x_{0:K})$ are mutually absolutely continuous path measures on $\mathcal{X}^{K+1}$ with marginals $\mathbb{P}^{\boldsymbol{\beta}}(x_0) = \pi^\beta(\mathrm{d}x_0)$, and $\mathbb{Q}^{\boldsymbol{\beta}}(\mathrm{d}x_K) = \pi^{\beta'}(\mathrm{d}x_K)$ with weight functional $w^{\boldsymbol{\beta}} : \mathcal{X}^{K+1} \mapsto \mathbb{R}$,

$$w^{\boldsymbol{\beta}}(x_{0:K}) = \frac{Z_{\beta'}}{Z_\beta} \frac{\mathrm{d}\mathbb{Q}^{\boldsymbol{\beta}}}{\mathrm{d}\mathbb{P}^{\boldsymbol{\beta}}}(x_{0:K}),$$

We will make the following regularity assumptions on $\mathbb{P}^{\boldsymbol{\beta}}$ and $\mathbb{Q}^{\boldsymbol{\beta}}$ analogous to Assumptions 5–7 in Syed et al. (2024). Suppose there exists $\mathcal{F} = (\mathcal{F}_i)_{i \in \mathbb{N}}$ of a nested sequence $\mathcal{F}_0 \subset \mathcal{F}_1 \subset \cdots$ of vector spaces of measurable function $f : \mathcal{X}^{K+1} \to \mathbb{R}$, such that (1) $\mathcal{F}_i$ contains the constant functions, and (2) $\mathcal{F}_i$ is closed under domination, i.e., if $g$ is a measurable function such that $|g| \le |f|$ for some $f \in \mathcal{F}_i$, then $g \in \mathcal{F}_i$.

**Assumption 2.** *For $i = 0, 1, 2$, and $\boldsymbol{\beta} \mapsto \mathbb{P}^{\boldsymbol{\beta}}[f]$ and $\boldsymbol{\beta} \mapsto \mathbb{Q}^{\boldsymbol{\beta}}[f]$ are $i$-times continuously differentiable for all $f \in \mathcal{F}_i$, there exists signed-measures $\partial_{\boldsymbol{\beta}}^i \mathbb{P}^{\boldsymbol{\beta}}$ and $\partial_{\boldsymbol{\beta}}^i \mathbb{Q}^{\boldsymbol{\beta}}$ integrable over $\mathcal{F}_i$ such that for all $f \in \mathcal{F}_i$,*

$$\partial_{\boldsymbol{\beta}}^i \mathbb{P}^{\boldsymbol{\beta}}[f] = \partial_{\boldsymbol{\beta}}^i(\mathbb{P}^{\boldsymbol{\beta}}[f]), \quad \partial_{\boldsymbol{\beta}}^i \mathbb{Q}^{\boldsymbol{\beta}}[f] = \partial_{\boldsymbol{\beta}}^i(\mathbb{Q}^{\boldsymbol{\beta}}[f]).$$

**Assumption 3.** *$\boldsymbol{\beta} \mapsto W^{\boldsymbol{\beta}}$ is 2-times continuously differentiable, and for $i = 0, 1, 2$ and $i_1, \ldots, i_j \ge 1$ and $i_1 + \cdots + i_j = i$, we have $\bar{W}_{i_1} \cdots \bar{W}_{i_j} \in \mathcal{F}_{3-i}$, where $\bar{W}_i = \sup_{\boldsymbol{\beta}} \left| \partial_{\boldsymbol{\beta}}^i W^{\boldsymbol{\beta}} \right|$.*

**Assumption 4.** *For all $\beta$ we have $\mathbb{P}^\beta := \mathbb{P}^{\beta,\beta} = \mathbb{Q}^{\beta,\beta} =: \mathbb{Q}^\beta$.*

**Lemma 2.** *Let $g^{\boldsymbol{\beta}} : \mathcal{X}^{K+1} \times \mathcal{X}^{K+1} \to \mathbb{R}$,*

- *$\boldsymbol{\beta} \mapsto g^{\boldsymbol{\beta}}(x_{0:K}, x'_{0:K})$ is continuously differentiable,*

- *$x_{0:K} \mapsto g^{\boldsymbol{\beta}}(x_{0:K}, x'_{0:K})$ and $x'_{0:K} \mapsto g^{\boldsymbol{\beta}}(x_{0:K}, x'_{0:K})$ are in $\mathcal{F}_1$,*

- $x_{0:K} \mapsto \partial_{\boldsymbol{\beta}} g^{\boldsymbol{\beta}}(x_{0:K}, x'_{0:K})$ and $x'_{0:K} \mapsto \partial_{\boldsymbol{\beta}} g^{\boldsymbol{\beta}}(x_{0:K}, x'_{0:K})$ are in $\mathcal{F}_0$.

*Then $\boldsymbol{\beta} \mapsto \mathbb{P}^{\boldsymbol{\beta}} \otimes \mathbb{Q}^{\boldsymbol{\beta}}[g^{\boldsymbol{\beta}}]$ is continuously differentiable with derivative,*

$$\partial_{\boldsymbol{\beta}}(\mathbb{P}^{\boldsymbol{\beta}} \otimes \mathbb{Q}^{\boldsymbol{\beta}}[g^{\boldsymbol{\beta}}]) = \partial_{\boldsymbol{\beta}}\mathbb{P}^{\boldsymbol{\beta}} \otimes \mathbb{Q}^{\boldsymbol{\beta}}[g] + \mathbb{P}^{\boldsymbol{\beta}} \otimes \partial_{\boldsymbol{\beta}}\mathbb{Q}^{\boldsymbol{\beta}}[g] + \mathbb{P}^{\boldsymbol{\beta}} \otimes \mathbb{Q}^{\boldsymbol{\beta}}[\partial_{\boldsymbol{\beta}}g^{\boldsymbol{\beta}}].$$

*Proof.* For notational convenience we denote $x := x_{0:K}$ and $x' := x_{0:K}$. Given $x \in \mathcal{X}^{K+1}$ let $g_x^{\boldsymbol{\beta}} : \mathcal{X}^{K+1} \to \mathbb{R}$ be the marginal $g_x^{\boldsymbol{\beta}}(x') = g(x, x')$ and let $q^{\boldsymbol{\beta}} : \mathcal{X}^{K+1} \to \mathbb{R}$ denote the expectation of $g_x^{\boldsymbol{\beta}}$ with respect to $\mathbb{Q}^{\boldsymbol{\beta}}$, i.e. $q^{\boldsymbol{\beta}}(x) = \mathbb{Q}^{\boldsymbol{\beta}}[g_x^{\boldsymbol{\beta}}]$. By product rule for measure-valued derivatives, we have for all $x, \boldsymbol{\beta} \mapsto q^{\boldsymbol{\beta}}(x)$ is continuously differentiable with derivative,

$$\begin{aligned}
\partial_{\boldsymbol{\beta}}q^{\boldsymbol{\beta}}(x) &= \partial_{\boldsymbol{\beta}}(\mathbb{Q}^{\boldsymbol{\beta}}[g_x^{\boldsymbol{\beta}}]) \\
&= \partial_{\boldsymbol{\beta}}\mathbb{Q}^{\boldsymbol{\beta}}[g_x^{\boldsymbol{\beta}}] + \mathbb{Q}^{\boldsymbol{\beta}}[\partial_{\boldsymbol{\beta}}g_x^{\boldsymbol{\beta}}].
\end{aligned}$$

by taking expectation of both sides with respect to $\mathbb{P}^{\boldsymbol{\beta}}$ and using Fubini's theorem we have

$$\mathbb{P}^{\boldsymbol{\beta}}[\partial_{\boldsymbol{\beta}}q^{\boldsymbol{\beta}}] = \mathbb{P}^{\boldsymbol{\beta}} \otimes \partial_{\boldsymbol{\beta}}\mathbb{Q}^{\boldsymbol{\beta}}[g^{\boldsymbol{\beta}}] + \mathbb{P}^{\boldsymbol{\beta}} \otimes \mathbb{Q}^{\boldsymbol{\beta}}[\partial_{\boldsymbol{\beta}}g^{\boldsymbol{\beta}}].$$

Again, using the product rule for measure-valued derivatives,

$$\begin{aligned}
\partial_{\boldsymbol{\beta}}(\mathbb{P}^{\boldsymbol{\beta}} \otimes \mathbb{Q}^{\boldsymbol{\beta}}[g^{\boldsymbol{\beta}}]) &= \partial_{\boldsymbol{\beta}}(\mathbb{P}^{\boldsymbol{\beta}}[q^{\boldsymbol{\beta}}]) \\
&= \partial_{\boldsymbol{\beta}}\mathbb{P}^{\boldsymbol{\beta}}[q^{\boldsymbol{\beta}}] + \mathbb{P}^{\boldsymbol{\beta}}[\partial_{\boldsymbol{\beta}}q^{\boldsymbol{\beta}}].
\end{aligned}$$

The result follows by noting that $\partial_{\boldsymbol{\beta}}\mathbb{P}^{\boldsymbol{\beta}}[q^{\boldsymbol{\beta}}] = \partial_{\boldsymbol{\beta}}\mathbb{P}^{\boldsymbol{\beta}} \otimes \mathbb{Q}^{\boldsymbol{\beta}}[g^{\boldsymbol{\beta}}]$ by Fubini's theorem. $\square$

### B.4.2 PROOF OF THEOREM 2

For $\boldsymbol{\beta} = (\beta, \beta')$ recall the rejection rate can be expressed as $r(\mathbb{P}^{\boldsymbol{\beta}}, \mathbb{Q}^{\boldsymbol{\beta}}) = 1 - \mathbb{P}^{\boldsymbol{\beta}} \otimes \mathbb{Q}^{\boldsymbol{\beta}}[\alpha^{\boldsymbol{\beta}}]$, where $\mathbb{P}^{\boldsymbol{\beta}} \otimes \mathbb{Q}^{\boldsymbol{\beta}}$ is a measure over $\mathcal{X}^{K+1} \times \mathcal{X}^{K+1}$ is the product between the forward and backwards paths,

$$\mathbb{P}^{\boldsymbol{\beta}} \otimes \mathbb{Q}^{\boldsymbol{\beta}}(\mathrm{d}x_{0:K}, \mathrm{d}x'_{0:K}) := \mathbb{P}^{\boldsymbol{\beta}}(\mathrm{d}x_{0:K})\mathbb{Q}^{\boldsymbol{\beta}}(\mathrm{d}x'_{0:K}),$$

$\alpha^{\boldsymbol{\beta}} : \mathcal{X}^{K+1} \times \mathcal{X}^{K+1} \to [0, 1]$ is the acceptance probability for swap proposed,

$$\alpha^{\boldsymbol{\beta}}(x_{0:K}, x'_{0:K}) = \min\left\{1, \frac{w_K^{\boldsymbol{\beta}}(x_{0:K})}{w_K^{\boldsymbol{\beta}}(x'_{0:K})}\right\} = \exp\left(\min\left\{0, \Delta W_K^{\boldsymbol{\beta}}(x_{0:K}, x'_{0:K})\right\}\right)$$

where $\Delta W^{\boldsymbol{\beta}} : \mathcal{X}^{K+1} \times \mathcal{X}^{K+1} \to \mathbb{R}$ is the change in log-weight $W^{\boldsymbol{\beta}}(x_{0:K}) := \log w^{\boldsymbol{\beta}}(x_{0:K})$,

$$\Delta W^{\boldsymbol{\beta}}(x_{0:K}, x'_{0:K}) := W^{\boldsymbol{\beta}}(x_{0:K}) - W^{\boldsymbol{\beta}}(x'_{0:K})$$

**Lemma 3.** *Suppose Assumptions 2–4 hold. For all $\boldsymbol{\beta} = (\beta, \beta')$ with $\Delta\beta = \beta' - \beta > 0$, exists a constant $C > 0$ independent of $\boldsymbol{\beta}$ such that,*

$$\left| r(\mathbb{P}^{\boldsymbol{\beta}}, \mathbb{Q}^{\boldsymbol{\beta}}) - \int_\beta^{\beta'} \lambda_b \mathrm{d}b \right| \leq C\Delta\beta^2.$$

*where $\lambda_\beta := \frac{1}{2}\mathbb{P}^{\boldsymbol{\beta}} \otimes \mathbb{Q}^{\boldsymbol{\beta}}[|\Delta\dot{W}^{\boldsymbol{\beta}}|]$ and $\Delta\dot{W}^{\boldsymbol{\beta}} := \lim_{\beta' \to \beta} \partial_{\beta'}\Delta W^{\boldsymbol{\beta}}$. Moreover, $\lambda_\beta$ is continuously differentiable in $\beta$.*

Given an annealing schedule $0 = \beta_0 < \cdots < \beta_N = 1$, let $\boldsymbol{\beta}_n = (\beta_{n-1}, \beta_n)$ and $\Delta\beta_n = \beta_n - \beta_{n-1}$. It follows from Lemma 3 that the sum of the rejection satisfies,

$$\left| \sum_{n=1}^N r(\mathbb{P}^{\boldsymbol{\beta}_n}, \mathbb{Q}^{\boldsymbol{\beta}_n}) - \Lambda \right| \leq C \sum_{n=1}^N \Delta\beta_n^2 \leq C \max_{n \leq N} |\Delta\beta_n|.$$

Therefore as $N \to \infty$ and $\max_{n \leq N} |\Delta\beta_n| \to 0$ we have the sum of the rejection rates converge to $\Lambda$.

For the same annealing schedule define $\tau^N(\beta_{0:N})$ as

$$\tau^N(\beta_{0:N}) := \left(2 + 2\sum_{n=1}^N \frac{r(\mathbb{P}^{\boldsymbol{\beta}_n}, \mathbb{Q}^{\boldsymbol{\beta}_n})}{1 - r(\mathbb{P}^{\boldsymbol{\beta}_n}, \mathbb{Q}^{\boldsymbol{\beta}_n})}\right)^{-1}$$

By Lemma 3 we have,

$$r(\mathbb{P}^{\boldsymbol{\beta}_n}, \mathbb{Q}^{\boldsymbol{\beta}_n}) \leq \frac{r(\mathbb{P}^{\boldsymbol{\beta}_n}, \mathbb{Q}^{\boldsymbol{\beta}_n})}{1 - r(\mathbb{P}^{\boldsymbol{\beta}_n}, \mathbb{Q}^{\boldsymbol{\beta}_n})} \leq \frac{r(\mathbb{P}^{\boldsymbol{\beta}_n}, \mathbb{Q}^{\boldsymbol{\beta}_n})}{1 - \max_{n \leq N} r(\mathbb{P}^{\boldsymbol{\beta}_n}, \mathbb{Q}^{\boldsymbol{\beta}_n})} \leq \frac{r(\mathbb{P}^{\boldsymbol{\beta}_n}, \mathbb{Q}^{\boldsymbol{\beta}_n})}{1 - \sup_{\beta} \lambda_{\beta} \max_{n \leq N} |\Delta \beta_n|}.$$

Therefore, by taking the sum over $n$ and using the squeeze theorem, we have in the limit $N \to \infty$ and $\max_{n \leq N} |\Delta \beta_n| \to 0$,

$$\lim_{N \to \infty} \sum_{n=1}^{N} \frac{r(\mathbb{P}^{\boldsymbol{\beta}_n}, \mathbb{Q}^{\boldsymbol{\beta}_n})}{1 - r(\mathbb{P}^{\boldsymbol{\beta}_n}, \mathbb{Q}^{\boldsymbol{\beta}_n})} = \Lambda$$

and hence $\lim_{N \to \infty} \tau^N(\beta_{0:N}) = (2 + 2\Lambda)^{-1}$, which completes the proof.

*Proof of Lemma 3.* Since $\boldsymbol{\beta} \mapsto W^{\boldsymbol{\beta}}$ is twice differentiable, we have that $\boldsymbol{\beta} \mapsto \alpha^{\boldsymbol{\beta}}$ is twice differentiable when $\Delta W^{\boldsymbol{\beta}} \neq 0$, with first-order partial derivatives with respect to $\beta'$:

$$\partial_{\beta'} \alpha^{\boldsymbol{\beta}} = \partial_{\beta'} \Delta W^{\boldsymbol{\beta}} \exp(\Delta W^{\boldsymbol{\beta}}) 1[\Delta W^{\boldsymbol{\beta}} < 0],$$

and the second-order partial derivative with respect to $\beta'$:

$$\partial_{\beta'}^2 \alpha^{\boldsymbol{\beta}} = [\partial_{\beta'}^2 \Delta W^{\boldsymbol{\beta}} + (\partial_{\beta'} \Delta W^{\boldsymbol{\beta}})^2] \exp(\Delta W^{\boldsymbol{\beta}}) 1[\Delta W^{\boldsymbol{\beta}} < 0].$$

By Taylor's theorem, for $\Delta \beta > 0$, we have

$$\alpha^{\boldsymbol{\beta}} = 1 + \dot{\alpha}^{\beta} \Delta \beta + \epsilon^{\boldsymbol{\beta}},$$

where $\dot{\alpha}^{\beta} = \lim_{\beta' \to \beta^+} \partial_{\beta'} \alpha^{\boldsymbol{\beta}}$ and $|\epsilon^{\boldsymbol{\beta}}| \leq \frac{1}{2} \sup_{\boldsymbol{\beta}} |\partial_{\beta'}^2 \alpha^{\boldsymbol{\beta}}| \Delta \beta^2$. Therefore, the rejection rate equals:

$$r(\mathbb{P}^{\boldsymbol{\beta}}, \mathbb{Q}^{\boldsymbol{\beta}}) = -\mathbb{P}^{\boldsymbol{\beta}} \otimes \mathbb{Q}^{\boldsymbol{\beta}}[\dot{\alpha}^{\beta}] \Delta \beta - \mathbb{P}^{\boldsymbol{\beta}} \otimes \mathbb{Q}^{\boldsymbol{\beta}}[\epsilon^{\boldsymbol{\beta}}]. \tag{26}$$

We will first approximate the first term in Equation (26). Note that since $\Delta W^{\beta} = 0$ and $\partial_{\beta'} \Delta W^{\boldsymbol{\beta}} = \lim_{\beta' \to \beta} \Delta W^{\boldsymbol{\beta}} / \Delta \beta =: \Delta \dot{W}^{\beta}$, we have that $\dot{\alpha}^{\beta}$ satisfies:

$$\begin{aligned}
\dot{\alpha}^{\beta} &= \lim_{\beta' \to \beta} \partial_{\beta'} \Delta W^{\boldsymbol{\beta}} \exp(\Delta W^{\boldsymbol{\beta}}) 1\left[\frac{\Delta W^{\boldsymbol{\beta}}}{\Delta \beta} < 0\right] \\
&= \Delta \dot{W}^{\beta} 1[\Delta \dot{W}^{\beta} < 0] \\
&= -|\Delta \dot{W}^{\beta}| 1[\Delta \dot{W}^{\beta} < 0].
\end{aligned}$$

We can bound $\dot{\alpha}^{\beta}$ uniformly in $\beta$ in terms of $\bar{W}_1$:

$$\begin{aligned}
|\dot{\alpha}^{\beta}(x_{0:K}, x'_{0:K})| &\leq |\dot{W}^{\beta}(x_{0:K}) - \dot{W}^{\beta}(x'_{0:K})| \\
&\leq \bar{W}_1(x_{0:K}) + \bar{W}_1(x'_{0:K}) \\
&:= \bar{W}_1 \oplus \bar{W}_1(x_{0:K}, x'_{0:K}).
\end{aligned}$$

It follows from Lemma 2 that $\beta' \mapsto \mathbb{P}^{\boldsymbol{\beta}} \otimes \mathbb{Q}^{\boldsymbol{\beta}}[\dot{\alpha}^{\beta}]$ is differentiable in $\beta'$. By Lemma 2 and the mean value theorem, there exists $\tilde{\boldsymbol{\beta}} = (\beta, \tilde{\beta}')$ with $\beta \leq \tilde{\beta}' \leq \beta'$ such that

$$\begin{aligned}
\frac{\mathbb{P}^{\boldsymbol{\beta}} \otimes \mathbb{Q}^{\boldsymbol{\beta}}[\dot{\alpha}^{\beta}] - \mathbb{P}^{\beta} \otimes \mathbb{Q}^{\beta}[\dot{\alpha}^{\beta}]}{\Delta \beta} &= \partial_{\beta'}(\mathbb{P}^{\boldsymbol{\beta}} \otimes \mathbb{Q}^{\boldsymbol{\beta}}[\dot{\alpha}^{\beta}])|_{\boldsymbol{\beta} = \tilde{\boldsymbol{\beta}}} \\
&= \partial_{\beta'} \mathbb{P}^{\boldsymbol{\beta}} \otimes \mathbb{Q}^{\boldsymbol{\beta}}[\dot{\alpha}^{\beta}]|_{\boldsymbol{\beta} = \tilde{\boldsymbol{\beta}}} + \mathbb{P}^{\boldsymbol{\beta}} \otimes \partial_{\beta'} \mathbb{Q}^{\boldsymbol{\beta}}[\dot{\alpha}^{\beta}]|_{\boldsymbol{\beta} = \tilde{\boldsymbol{\beta}}}.
\end{aligned}$$

Using the triangle inequality and $|\dot{\alpha}^{\beta}| \leq \bar{W}_1 \oplus \bar{W}_1$, we have:

$$\begin{aligned}
\left| \frac{\mathbb{P}^{\boldsymbol{\beta}} \otimes \mathbb{Q}^{\boldsymbol{\beta}}[\dot{\alpha}^{\beta}] - \mathbb{P}^{\beta} \otimes \mathbb{Q}^{\beta}[\dot{\alpha}^{\beta}]}{\Delta \beta} \right| &\leq \sup_{\boldsymbol{\beta}} |\partial_{\beta'} \mathbb{P}^{\boldsymbol{\beta}}| \otimes \mathbb{Q}^{\boldsymbol{\beta}}[\bar{W}_1 \oplus \bar{W}_1] \\
&\quad + \sup_{\boldsymbol{\beta}} \mathbb{P}^{\boldsymbol{\beta}} \otimes |\partial_{\beta'} \mathbb{Q}^{\boldsymbol{\beta}}|[\bar{W}_1 \oplus \bar{W}_1] \\
&= \sup_{\boldsymbol{\beta}} \left( |\partial_{\beta'} \mathbb{P}^{\boldsymbol{\beta}}|[1] \mathbb{Q}^{\boldsymbol{\beta}}[\bar{W}_1] + |\partial_{\beta'} \mathbb{P}^{\boldsymbol{\beta}}|[\bar{W}_1] \mathbb{Q}^{\boldsymbol{\beta}}[1] \right) \\
&\quad + \sup_{\boldsymbol{\beta}} \left( \mathbb{P}^{\boldsymbol{\beta}}[1] |\partial_{\beta'} \mathbb{Q}^{\boldsymbol{\beta}}|[\bar{W}_1] + \mathbb{P}^{\boldsymbol{\beta}}[\bar{W}_1] |\partial_{\beta'} \mathbb{Q}^{\boldsymbol{\beta}}|[1] \right).
\end{aligned}$$

Since 1 and $\bar{W}_1$ are in $\mathcal{F}_1$, we have that each of the terms on the right-hand side is continuous and hence by the extreme value theorem, there exists $C_1$ such that

$$|\mathbb{P}^{\boldsymbol{\beta}} \otimes \mathbb{Q}^{\boldsymbol{\beta}}[\dot{\alpha}^{\beta}] - \mathbb{P}^{\beta} \otimes \mathbb{Q}^{\beta}[\dot{\alpha}^{\beta}]| \leq C_1 \Delta\beta.$$

For the second term in Equation (26), we have

$$
\begin{aligned}
|\mathbb{P}^{\boldsymbol{\beta}} \otimes \mathbb{Q}^{\boldsymbol{\beta}}[\epsilon^{\boldsymbol{\beta}}]| &\leq \frac{\Delta\beta^2}{2} \mathbb{P}^{\boldsymbol{\beta}} \otimes \mathbb{Q}^{\boldsymbol{\beta}}[\sup_{\boldsymbol{\beta}} |\partial_{\beta'}^2 \alpha^{\boldsymbol{\beta}}|] \\
&\leq \frac{\Delta\beta^2}{2} \mathbb{P}^{\boldsymbol{\beta}} \otimes \mathbb{Q}^{\boldsymbol{\beta}}[\bar{W}_2 \oplus \bar{W}_2 + (\bar{W}_1 \oplus \bar{W}_1)^2] \\
&\leq \frac{\Delta\beta^2}{2} \sup_{\boldsymbol{\beta}} \mathbb{P}^{\boldsymbol{\beta}} \otimes \mathbb{Q}^{\boldsymbol{\beta}}[\bar{W}_2 \oplus \bar{W}_2 + (\bar{W}_1 \oplus \bar{W}_1)^2] \\
&:= C_2 \Delta\beta^2,
\end{aligned}
$$

where $\bar{W}_2 \oplus \bar{W}_2(x_{0:K}, x'_{0:K}) := \bar{W}_2(x_{0:K}) + \bar{W}_2(x'_{0:K})$. Assumption 2 guarantees that the expectation in the second-to-last line is continuous in $\boldsymbol{\beta}$, and hence $C_2$ is finite. Next, we note that since $|\Delta\dot{W}^\beta(x_{0:K}, x'_{0:K})| = |\Delta\dot{W}^\beta(x'_{0:K}, x_{0:K})|$ is symmetric and $\mathbb{P}^\beta = \mathbb{Q}^\beta$, we have:

$$
\begin{aligned}
\mathbb{P}^\beta \otimes \mathbb{Q}^\beta[\dot{\alpha}^\beta] &= \mathbb{P}^\beta \otimes \mathbb{Q}^\beta[-|\Delta\dot{W}^\beta|\mathbf{1}[\Delta\dot{W}^\beta < 0]] \\
&= -\frac{1}{2}\mathbb{P}^\beta \otimes \mathbb{Q}^\beta[|\Delta\dot{W}^\beta|] \\
&= -\lambda_\beta.
\end{aligned}
$$

By Lemma 2, $\beta \mapsto \lambda_\beta$ is continuously differentiable. Since $\lambda_\beta \Delta\beta$ is a right Riemann sum for the integral of $\lambda_\beta$ with error:

$$\left|\lambda_\beta \Delta\beta - \int_\beta^{\beta'} \lambda_b \, \mathrm{d}b\right| = \frac{1}{2}\sup_{\beta'}\left|\frac{\mathrm{d}\lambda_\beta}{\mathrm{d}\beta}\right|\Delta\beta^2 =: C_3 \Delta\beta^2.$$

Finally, by the triangle inequality:

$$\left|r(\mathbb{P}^{\boldsymbol{\beta}}, \mathbb{Q}^{\boldsymbol{\beta}}) - \int_\beta^{\beta'} \lambda_b \, \mathrm{d}b\right| \leq C\Delta\beta^2,$$

for $C := C_1 + C_2 + C_3$. $\qquad\square$

### B.5 ACCELERATED PT AS VANILLA PT ON EXTENDED SPACE

In this section we establish the theoretical relationship between accelerated PT and vanilla PT. We state an equivalence between accelerated PT and a particular vanilla PT problem which "linearises" it. More concretely, we define a sequence of distributions $\pi_{\mathrm{ex}}^0, \ldots, \pi_{\mathrm{ex}}^N$ supported on an extended space, such that if we run *vanilla* PT on it, we obtain the same round trip rate as if we ran *accelerated* PT on $\pi^0, \ldots, \pi^N$.

We consider the case $K = 1$ and simplify the notations of forward and backward kernels to $P^n(x^{n-1}, \mathrm{d}x^n)$ and $Q^{n-1}(x^n, \mathrm{d}x^{n-1})$. This does not incur any loss of generality since conceptually multiple Markov steps can be collapsed into one.

Given a sequence of distributions $\pi^0, \ldots, \pi^N$ each supported on $\mathcal{X}$, define the distributions $\pi_{\mathrm{ex}}^n$ as:

$$\pi_{\mathrm{ex}}^n(\mathrm{d}x^0, \ldots, \mathrm{d}x^N) := \pi^n(\mathrm{d}x^n) \prod_{i \geq n+1} P^i(x^{i-1}, \mathrm{d}x^i) \prod_{j \leq n-1} Q^j(x^{j+1}, \mathrm{d}x^j). \tag{27}$$

In particular we stress that the distributions $\pi_{\mathrm{ex}}^n$ are supported on $\mathcal{X}^{N+1}$ and not $\mathcal{X}$.

The following proposition establishes the isometry between accelerated PT on $\pi^1, \ldots, \pi^N$ and vanilla PT on $\pi_{\mathrm{ex}}^1, \ldots, \pi_{\mathrm{ex}}^N$. Recall that we use $r(\mu_1, \mu_2)$ to denote the rejection between two distributions $\mu_1$ and $\mu_2$.

**Proposition 5.** *For all $1 \le n \le N$, we have $r(\pi^{n-1} \times P^n, Q^{n-1} \times \pi^n) = r(\pi_{\text{ex}}^{n-1}, \pi_{\text{ex}}^n)$.*

*Proof.* Since the rejection rate only depends on the Radon-Nikodym derivative, it suffices to verify that

$$\frac{\mathrm{d}\pi_{\text{ex}}^{n-1}}{\mathrm{d}\pi_{\text{ex}}^n}(x^0, \ldots, x^N) = \frac{\mathrm{d}(\pi^{n-1} \times P^n)}{\mathrm{d}(Q^{n-1} \times \pi^n)}(x^{n-1}, x^n)$$

which is straightforward from Equation (27). $\qquad\square$

This proposition shows that Accelerated PT outperforms traditional PT in two ways:

- First, while traditional PT bridges $\pi^0$ and $\pi^N$, accelerated PT bridges $\pi_{\text{ex}}^0$ and $\pi_{\text{ex}}^N$ which can be much closer to each other if the forward and backward kernels are good;
- In addition, accelerated PT inserts $N-1$ distributions between $\pi_{\text{ex}}^0$ and $\pi_{\text{ex}}^N$.

### B.5.1 PARALLELISM VERSUS ACCELERATION TIME

Given two distributions $\pi^{n-1}$ and $\pi^n$, should we apply $K$-step forward and backward kernels; or insert $K-1$ intermediate distributions $(\pi^{n-1,k})_{k=1}^{K-1}$ and use only one-step forward and backward kernels instead?

By Proposition 2, the inverse of the local round trip rate between $\pi^{n-1}$ and $\pi^n$ for the first method (*time-accelerated*) is

$$\tau_{\text{TA}}^{-1} := 2 + 2 \frac{r(\mathbb{P}_K^{n-1}, \mathbb{Q}_K^n)}{1 - r(\mathbb{P}_K^{n-1}, \mathbb{Q}_K^n)}.$$

The inverse of the local round trip rate between $\pi^{n-1}$ and $\pi^n$ for the second method (*parallel-accelerated*) is

$$\tau_{\text{PA}}^{-1} := 2 + 2 \sum_{k=1}^K \frac{r(\pi^{n-1,k-1} \times P_k^{n-1}, Q_{k-1}^n \times \pi^{n-1,k})}{1 - r(\pi^{n-1,k-1} \times P_k^{n-1}, Q_{k-1}^n \times \pi^{n-1,k})}$$

where we make the convention $\pi^{n-1,0} \equiv \pi^{n-1}$ and $\pi^{n-1,K} \equiv \pi^n$.

The following proposition analyses these local rates for both methods. As in Appendix B.5, the main idea is to find a vanilla PT equivalent for both algorithms. We define

$$\tau_{\text{VA}}^{-1}(\mu_0, \ldots, \mu_K) := 2 + 2 \sum_{k=1}^K \frac{r(\mu_{k-1}, \mu_k)}{1 - r(\mu_{k-1}, \mu_k)}$$

as the inverse round trip rate of a vanilla PT algorithm on a sequence of distributions $\mu_0, \ldots, \mu_K$.

**Proposition 6.** *Define the sequence of distributions $(\mathbb{S}_k)_{k=0}^K$ as*

$$\mathbb{S}_k(\mathrm{d}x_0, \mathrm{d}x_1, \ldots, \mathrm{d}x_K) := \pi^{n-1,k}(\mathrm{d}x_k) \times \prod_{i \ge k+1} P_i^{n-1}(x_{i-1}, \mathrm{d}x_i) \prod_{j \le k-1} Q_j^n(x_{j+1}, \mathrm{d}x_j).$$

*Then the following equalities hold*

$$\tau_{\text{TA}} = \tau_{\text{VA}}(\mathbb{S}_0, \mathbb{S}_K) \tag{28}$$
$$\tau_{\text{PA}} = \tau_{\text{VA}}(\mathbb{S}_0, \mathbb{S}_1, \ldots, \mathbb{S}_K). \tag{29}$$

*Proof.* The first point is straightforward. To show the second point, we need to check that

$$r(\pi^{n-1,k-1} \times P_k^{n-1}, Q_{k-1}^n \times \pi^{n-1,k}) = r(\mathbb{S}_{k-1}, \mathbb{S}_k).$$

Note that the rejection rates only depend on the Radon-Nikodym derivatives, so it suffices to verify that

$$\frac{\mathrm{d}\mathbb{S}_k}{\mathrm{d}\mathbb{S}_{k-1}}(x_0, x_1, \ldots, x_K) = \frac{\mathrm{d}(Q_{k-1}^n \times \pi^{n-1,k})}{\mathrm{d}(\pi^{n-1,k-1} \times P_k^{n-1})}(x_{k-1}, x_k)$$

which is straightforward from the definition of $(\mathbb{S}_k)_{k=0}^K$. $\qquad\square$

This proposition shows that it is preferable to use the parallel-accelerated method, as the quantity in Equation (29) is generally greater than that of Equation (28) thanks to the effect of the bridge between $\mathbb{S}_0$ and $\mathbb{S}_K$. However, in practice the time-accelerated method consumes less memory and so might be more suitable in certain circumstances.

## C    SCHEDULE TUNING

Under Assumption 1 (efficient local exploration), by Proposition 2, we wish to optimise our schedule $0 = \beta_0 < \ldots < \beta_N = 1$ to minimise

$$\sum_{n=1}^{N} \frac{r(\mathbb{P}_K^{n-1}, \mathbb{Q}_K^n)}{1 - r(\mathbb{P}_K^{n-1}, \mathbb{Q}_K^n)},$$

in order to maximise the round trip rate of APT. Additionally, by Theorem 2, we have $\sum_{n=1}^{N} r(\mathbb{P}_K^{n-1}, \mathbb{Q}_K^n) \approx \Lambda_K$. Therefore, a reasonable proxy objective is to minimise $\sum_{n=1}^{N} r_n/(1 - r_n)$ under the constraint that $\sum_{n=1}^{N} r_n = \Lambda_K$ and $r_n > 0$. Indeed this is same reasoning employed in (Syed et al., 2022, Section 5) to derive their schedule tuning algorithm.

Thus, following Syed et al. (2022), the solution to the proxy objective is to ensure the $r_n$ are constant in $n$. This provides a natural objective to choose our schedule in such a way which results in the theoretical rejection rates $r(\mathbb{P}_K^{n-1}, \mathbb{Q}_K^n)$ being uniform for all $n$. Moreover, while we do not have access to $r(\mathbb{P}_K^{n-1}, \mathbb{Q}_K^n)$, we can estimate these quantities empirically through computing the average rejection rates $\hat{r}_K^n$ observed in running Algorithm 1. Specifically, during sampling, we keep track of the average $\hat{\alpha}_K^n$ of the computed acceptance probabilities $\alpha_K^n$ for $n = 1, \ldots, N$. The estimated rejection rates $\hat{r}_K^n \approx r(\mathbb{P}_K^{n-1}, \mathbb{Q}_K^n)$ are then given by $\hat{r}_K^n = 1 - \hat{\alpha}_K^n$.

Finally, to achieve our above goal, we can use the same schedule tuning algorithm from (Syed et al., 2022, Section 5). For completeness, we reproduce this in Algorithm 2.

---

**Algorithm 2** Schedule Tuning

---

**Input:** Initial schedule $0 = \beta_0 < \ldots < \beta_N = 1$, number of tuning steps $m$, number of sampling steps $T_{\text{tune}}$;
1: **for** $i = 1, \ldots, m$ **do**
2:     Run Algorithm 1 for $T_{\text{tune}}$ steps and compute the average rejection rates $\{\hat{r}_K^n\}_{n=1}^N$.
3:     Set $\hat{\Lambda}_K(n) = \sum_{j=1}^{n} \hat{r}_K^n$ for $n = 1, \ldots, N$ and $\hat{\Lambda}_K(0) = 0$.
4:     $S \leftarrow \{(\beta_n, \frac{\Lambda_K(n)}{\Lambda_K(N)})\}_{n=0}^{N}$
5:     Fit a monotonically increasing interpolation $\Lambda_K(\cdot)$ (e.g. spline) of the points in $S$ which satisfies the boundary conditions: $\Lambda_K(0) = 0$ and $\Lambda_K(1) = 1$.
6:     $\beta_n \leftarrow \Lambda_K^{-1}\left(\frac{n}{N}\right)$ where $\Lambda_K^{-1}(\cdot)$ denotes the inverse of $\Lambda_K(\cdot)$.
7: **end for**
**Output:** $0 = \beta_1 < \ldots < \beta_N = 1$

---

## D    FURTHER DETAILS ON DESIGN-SPACE FOR ACCELERATED PT

### D.1    FURTHER DETAILS ON FLOW APT

#### D.1.1    WORK FORMULA FOR DETERMINISTIC FLOWS

**Proposition 7.** *Let $\mathcal{X} = \mathbb{R}^d$ and suppose that $\pi^{n-1}$ and $\pi^n$ admit strictly positive densities $\tilde{\pi}^{n-1}$ and $\tilde{\pi}_n$ with respect to the Lebesgue measure. Let $T^n : \mathbb{R}^d \to \mathbb{R}^d$ be a diffeomorphism with Jacobian matrix $J_{T_n}(x)$. If we choose the one-step forward and backward kernels $P^{n-1}$ and $Q^n$ such that*

$$P^{n-1}(x^{n-1}, \mathrm{d}x_*^n) = \delta_{T^n(x^{n-1})}(\mathrm{d}x_*^n), \qquad Q^n(x^n, \mathrm{d}x_*^{n-1}) = \delta_{(T^n)^{-1}(x^n)}(\mathrm{d}x_*^{n-1}),$$

*then, for all $(z^{n-1}, z^n)$ such that $T^n(z^{n-1}) = z^n$, we have the following expression for the weight defined in Equation (3):*

$$w^n(z^{n-1}, z^n) = \frac{\tilde{\pi}^n(z^n)}{\tilde{\pi}^{n-1}(z^{n-1})}|\det J_{T^n}(z^{n-1})|$$

*and the acceptance rate $\alpha^n$ defined in Section 3.2 becomes*

$$\alpha^n(x^{n-1}, x^n; x_*^{n-1}, x_*^n) = 1 \wedge \left[ \frac{\tilde{\pi}^{n-1}(x_*^{n-1})\tilde{\pi}^n(x_*^n)}{\tilde{\pi}^{n-1}(x^{n-1})\tilde{\pi}^n(x^n)} \cdot \frac{|\det(J_{T^n}(x^{n-1}))|}{|\det(J_{T^n}(x_*^{n-1}))|} \right]. \tag{30}$$

*Proof.* Let $S := \{(z^{n-1}, z^n) \in \mathbb{R}^d \times \mathbb{R}^d \text{ such that } T^n(z^{n-1}) = z^n\}$. Recall the definition of the extended measures $\mathbb{P}^{n-1}$ and $\mathbb{Q}^n$:

$$\mathbb{P}^{n-1}(\mathrm{d}z^{n-1}, \mathrm{d}z^n) = \pi^{n-1}(\mathrm{d}z^{n-1})P^{n-1}(z^{n-1}, \mathrm{d}z^n),$$
$$\mathbb{Q}^n(\mathrm{d}z^{n-1}, \mathrm{d}z^n) = \pi^n(\mathrm{d}z^n)Q^n(z^n, \mathrm{d}z^{n-1}).$$

Moreover for $(z^{n-1}, z^n) \in S$,

$$\mathbb{P}^{n-1}(\mathrm{d}z^{n-1}, \mathrm{d}z^n) = (T^n \# \pi^{n-1})(\mathrm{d}z^n)Q^n(z^n, \mathrm{d}z^{n-1}).$$

Therefore

$$\frac{\mathrm{d}\mathbb{Q}^n}{\mathrm{d}\mathbb{P}^{n-1}}(z^{n-1}, z^n) = \frac{\pi^n(\mathrm{d}z^n)}{(T^n \# \pi^{n-1})(\mathrm{d}z^n)}$$
$$= \frac{\pi^n(z^n)}{\pi^{n-1}((T^n)^{-1}(z^n))|\det(J_{(T^n)^{-1}}(z^n))|} = \frac{\pi^n(z^n)|\det(J_{T^n}(z^{n-1}))|}{\pi^{n-1}(z^{n-1})} \tag{31}$$

which justifies the identity for the weight. Applying this at $(z^{n-1}, z^n) = (x^{n-1}, x_*^n) \in S$ gives

$$\frac{\mathrm{d}\mathbb{Q}^n}{\mathrm{d}\mathbb{P}^{n-1}}(x^{n-1}, x_*^n) = \frac{\pi^n(x_*^n)|\det(J_{T^n}(x^{n-1}))|}{\pi^{n-1}(x^{n-1})}. \tag{32}$$

Similarly, applying Equation (31) at $(z^{n-1}, z^n) = (x_*^{n-1}, x^n) \in S$ gives

$$\frac{\mathrm{d}\mathbb{Q}^n}{\mathrm{d}\mathbb{P}^{n-1}}(x_*^{n-1}, x^n) = \frac{\pi^n(x^n)|\det(J_{T^n}(x_*^{n-1}))|}{\pi^{n-1}(x_*^{n-1})}. \tag{33}$$

Together Equation (32) and Equation (33) establish the proposition. □

### D.1.2 TRAINING

Since APT provides approximate samples from both the target and reference densities for each flow, it enables a range of training objectives. Some choices include maximum likelihood estimation (MLE) (equivalent to forward KL), reverse KL, and symmetric KL (SKL), which averages the two. Each has trade-offs: forward KL promotes mode-covering, reverse KL is more mode-seeking, and SKL balances both. APT's parallel structure makes SKL particularly effective by providing access to samples at each intermediate annealing distribution, a feature many other methods lack. For example, sequential Monte Carlo (SMC)-based approaches such as FAB (Midgley et al., 2023) and CRAFT (Matthews et al., 2022) rely on samples from only one side, limiting their choice of loss functions.

Additionally, one can explore loss functions based on APT's rejection rates, such as the analytic round trip rate, see Proposition 2. Since higher round trip rates indicate more efficient mixing, optimizing for this metric improves sampling performance. Empirically, we found that using SKL yielded the most stable and robust results across different settings. This loss was therefore used in our final experiments. However, we leave it to future work to study other possible losses in more detail.

There are also possible variants to the training pipeline. In each case, we initialize normalizing flows to the identity transformation. One option is to run the APT algorithm with the current flows. After every (or several) steps of the APT algorithm, the current samples can be used to update the flow parameters. Since the flows are initialized at the identity transformation, the initial sampling of APT behaves similarly to PT. A second possible training pipeline is to instead use PT directly to generate a large batch of samples, and then use this batch of samples to update the flows. The latter approach is more stable and is thus what we employed in the final experiments. We provide more details in Appendix E. We note that the first training pipeline has the potential to allow for better exploration, and we therefore leave a more detailed exploration of it to future work.

## D.2 Further Details on Control Accelerated PT

At the limit $K \to \infty$, following (Berner et al., 2025, Lemma B.7), and by the controlled Crooks fluctuation theorem (Vaikuntanathan & Jarzynski, 2008; Vargas et al., 2024), we arrive at the generalised work functional

$$W_\infty^n(x) := \log w_\infty^n(x)$$
$$= \int_0^1 \nabla \cdot b_s^n(x_s) + \nabla U_s^n(x_s) \cdot b_s^n(x_s) + \partial_s U_s^n(x_s) \mathrm{d}s,$$

inducing the corresponding continuous processes $(X_s)_{s \in [0,1]}$, $(X'_s)_{s \in [0,1]}$ for $\mathbb{P}_\infty^{n-1}$, $\mathbb{Q}_\infty^n$ respectively, that is for the forward process

$$X_0 \sim \pi^{n-1}, \quad \mathrm{d}X_s = (\sigma_s^n)^2 \nabla U_s^n(X_s)\mathrm{d}s + b_s^n(X_s)\mathrm{d}s + \sigma_s^n \sqrt{2}\mathrm{d}\vec{B}_s,$$

and for the backwards process

$$X'_1 \sim \pi^n, \quad \mathrm{d}X'_s = -(\sigma_s^n)^2 \nabla U_s^n(X'_s)\mathrm{d}s + b_s^n(X'_s)\mathrm{d}s + \sigma_s^n \sqrt{2}\mathrm{d}\overleftarrow{B}_s.$$

The training objective is

$$\mathcal{L}(b_s, \phi_s, \sigma_s) = \sum_{n=1}^N \mathrm{SKL}(\mathbb{P}_K^{n-1}, \mathbb{Q}_K^n) \xrightarrow[K \to \infty]{} \sum_{n=1}^N \mathrm{SKL}(\mathbb{P}_\infty^{n-1}, \mathbb{Q}_\infty^n).$$

Note that the discrete version discussed in the main text is not the only discretisation choice. Other options (Albergo & Vanden-Eijnden, 2024; Máté & Fleuret, 2023) may also be applied.

## D.3 Further Details on Diff-APT

**Annealing path**  We use the following Variance-Preserving (VP) diffusion process

$$dY_s = -\gamma_s Y_s \mathrm{d}s + \sqrt{2\gamma_s}\mathrm{d}W_s, \quad Y_0 \sim \pi,$$

with the choice of schedule $\gamma_s = \frac{1}{2(1-s)}$ to define the path of distributions $(\pi_s^{\mathrm{VP}})_{s \in (0,1]}$ by $Y_s \sim \pi_{1-s}^{\mathrm{VP}}$. Due to the singularity at $s = 1$, $\pi_0^{\mathrm{VP}}$ is not defined by the path, but we can define this point to be a standard Gaussian as the path converges to this distribution in the limit as $s$ approaches 1 in order to define the full annealing path $(\pi_s^{\mathrm{VP}})_{s \in [0,1]}$.

**Accelerators**  For a given annealing schedule $0 = s_0 < \ldots < s_N = 1$, we construct $P_k^{n-1}, Q_{k-1}^n$ through the linear discretisation of the time-reversal SDE on the time interval $[s_{n-1}, s_n]$ with step size $\delta_n = (s_n - s_{n-1})/K$ and interpolating times $s_n^k = s_{n-1} + k\delta_n$.

In particular, we take $Q_{k-1}^n(x_k, dx_{k-1})$ to be $\mathcal{N}(\sqrt{1 - \alpha_{n,k-1}}x_k, \alpha_{n,k-1}\mathrm{I})$ where $\alpha_{n,k-1} = 1 - \exp(-2\int_{1-s_n^k}^{1-s_n^{k-1}} \gamma_s \mathrm{d}s)$ which is the closed-form kernel transporting $\pi_{s_n^k}^{\mathrm{VP}}$ to $\pi_{s_n^{k-1}}^{\mathrm{VP}}$. Furthermore, we take $P_k^{n-1}(x_{k-1}, dx_k)$ to be the exponential integrator given by $\mathcal{N}(\mu_{n,k-1}(x_{k-1}), \alpha_{n,k-1}\mathrm{I})$ where

$$\mu_{n,k-1}(x) = \sqrt{1 - \alpha_{n,k-1}}x + 2(1 - \sqrt{1 - \alpha_{n,k-1}})(x + \nabla \log \pi_{s_n^{k-1}}^{\mathrm{VP}}(x)).$$

**Network parametrisation**  We parametrise an energy-based model as outlined in Phillips et al. (2024) but modified to ensure that $\pi_0^\theta(x) \propto N(x; 0, \mathrm{I})$. For completeness, we specifically take $\log \pi_s^\theta(x) = \log g_s^\theta(x) - \frac{1}{2}||x||^2$ where

$$\log g_\theta(x, s) = [r_\theta(1) - r_\theta(s)][r_\theta(s) - r_\theta(0)]\langle N_\theta(x, s), x \rangle$$
$$+ [1 + r_\theta(1) - r_\theta(s)]\log g_1(\sqrt{s}x),$$

where $\pi(x) \propto g_1(x)N(x; 0, \mathrm{I})$. Here, $r$ is a scalar-valued neural network and $N$ is a vector-valued function in $\mathbb{R}^d$. We also note that $\pi_1^\theta(x) \propto \pi$, hence $\pi_s^\theta$ serves as a valid annealing path between $N(0, \mathrm{I})$ and $\pi$.

## E EXPERIMENTAL DETAILS

### E.1 TARGET DISTRIBUTIONS

**GMM-$d$**   We take the 40-mode Gaussian mixture model (GMM-2) in 2 dimensions from Midgley et al. (2023) where to extend this distribution to higher dimensions $d$, we extend the means with zero padding to a vector in $\mathbb{R}^d$ and keep the covariances as the identity matrix but now within $\mathbb{R}^d$. This helps to disentangle the effect of multi-modality from the effect of dimensionality on performance as we essentially fix the structure of the modes across different values of $d$. Following previous work (Akhound-Sadegh et al., 2024; Phillips et al., 2024), we also scale the distribution GMM-$d$ by a factor of 40 to ensure the modes are contained within the range $[-1, 1]^d$ for our experiments with APT, where we also use the same scaling for the PT baseline to ensure a fair comparison.

**DW-4**   We take the DW-4 target from Köhler et al. (2020) which describes the energy landscape for a toy system of 4 particles $\{x_1, x_2, x_3, x_4\}$ and $x_i \in \mathbb{R}^2$ given by

$$\pi(x) \propto \exp\left(-\frac{1}{2\tau}\sum_{i \neq j} a(d_{ij} - d_0) + b(d_{ij} - d_0)^2 + c(d_{ij} - d_0)^4\right),$$

where $d_{ij} = ||x_i - x_j||$ and we set $a = 0, b = -4, c = 0.9, \tau = 1$ in accordance with previous work.

**MW-32**   We take the ManyWell-32 target from Midgley et al. (2023) formed from concatenating 16 copies of the 2-dimensional distribution

$$\hat{\pi}(x_1, x_2) \propto \exp\left(-x_1^4 + 6x_1^2 + \frac{1}{2}x_1 - \frac{1}{2}x_2^2\right),$$

i.e. the distribution $\pi(x) = \prod_{i=1}^{16} \hat{\pi}(x_{2i-1}, x_{2i})$ where $x \in \mathbb{R}^{32}$. Each copy of $\hat{\pi}$ has 2 modes and hence $\pi$ contains in total $2^{16}$ modes.

**Alanine Dipeptide**   This is a small molecule with 22 atoms, each of which has 3 dimensions. The target energy is defined with the `amber14/protein.ff14SB` forcefield in vacuum using the DMFF library in JAX (Wang et al., 2023).

### E.2 NETWORK AND TRAINING DETAILS

**NF-APT**   For GMM-$d$, we use 20 RealNVP layers where we employ a 2-layer MLP with 128 hidden units for the scale and translation functions (Dinh et al., 2017). We initialize the flow at the identity transformation. We use the Adam optimizer with a learning rate of `1e-3`, perform gradient clipping with norm 1 and employ EMA with decay parameter 0.99.

We use the same training pipeline as for CMCD-APT. See further details in the CMCD-APT description below. Note that we also employ the SKL loss for training and the linear annealing path with a standard Gaussian as the reference distribution.

**CMCD-APT**   For both GMM-$d$ and MW-32, we use a 4-layer MLP with 512 hidden units, and for DW-4, we use a 4-layer EGNN (Satorras et al., 2021) with 64 hidden units. Recall that in CMCD-APT, we use the *geometric path*, $\pi^\beta(x) = \eta(x)^{1-\beta}\pi(x)^\beta$, which linearly interpolates between reference and target in log-space. Therefore, our network is conditional on the $\beta$. We optimise the MLP by Adam with a learning rate of `1e-3` and EGNN with a learning rate of `1e-4`. We additionally use gradient clipping with norm 1 for stability.

Furthermore, our training pipeline following the 3 stages outlined below:

- Tuning PT: We initialise $\{\beta_n\}_{n=1}^N$ uniformly from 0 to 1. Then, we run PT for 600 steps, remove the first 100 steps as burn-in, and take the last 500 steps to calculate the rejection rate between adjacent chains using this to apply Algorithm 2 to update the schedule. We repeat this process 10 times to ensure $\{\beta_n\}_{n=1}^N$ is stable.

- Collecting data and training: We use PT with the tuned schedule to collect data. For experiments with 5 chains, we run 200K steps to collect 200K samples for each chain. For experiments with 10 and 30 chains, we run 65536 steps to collect data. We then train CMCD for 100,000 iterations with a batch size of 512. In each batch, we randomly select the chain index and samples according to the chain to train CMCD using SKL. We repeat this step twice to ensure the network is well-trained until convergence.

- Testing: We follow Algorithm 1, running CMCD-APT for 100K steps, and calculate the round trip.

**Diff-APT** For GMM-$d$, we use a 3-layer MLP with 128 hidden units, for MW-32, we use a 3-layer MLP with 256 hidden units and for DW-4, we use a 3 layer EGNN with 128 hidden units. For all models, we use a learning rate of `5e-4`, gradient clipping with norm 1 for stability and EMA with decay parameter of 0.99. For training, we follow the same pipeline as CMCD-APT, but we only retain samples at the target chain in order to optimise the standard score matching objective. At sampling time, we tune the annealing schedule by running Diff-APT for 1,100 steps, discarding the first 100 samples as burn-in and using the last 1,000 steps to calculate rejection rates. We then use this to apply Algorithm 2 to update the schedule. We then repeat this 10 times where we initialise from the uniform schedule.

**PT** For all experiments with PT, we use the linear path with a standard Gaussian as our reference distribution. We tune the annealing schedule in the same manner as with Diff-APT to ensure comparisons with a strong baseline.

### E.3 FURTHER DETAILS ON COMPARISON OF ACCELERATION METHODS

For both training and testing for all methods, we take a single step of HMC with step size of $0.03$ and 5 leapfrog steps as our local exploration step across each annealing chain. We note that while we could have improved performance by tuning the step size for each annealing distribution, we keep this fixed to disentangle the effect of local exploration from our communication steps.

**Compute-normalised round trips** The compute-normalised round trips, as reported in Table 1, is computed by dividing the original round trips by the number of potential evaluations that a single "machine" is required to implement within a parallelised implementation of PT/APT - i.e. the number of potential evaluations required by the computation of $w_K^n(\vec{X}_{t,0:K}^{n-1})$ (or equivalently $w_K^n(\overleftarrow{X}_{t,0:K}^n)$) for a single $n$ and $t$. Similarly, we count the number of neural calls in the corresponding manner.

- NF-APT: We need 2 potential evaluations for $\pi^{n-1}(\vec{X}_{t,0}^{n-1})$ and $\pi^n(\vec{X}_{t,1}^{n-1})$ and single network evaluation.

- CMCD-APT: For $K > 0$, we need to calculate the potential and score[5] of $\pi^{n-1}$ at $\vec{X}_{t,0}^{n-1}$ and $\pi^n$ at $\vec{X}_{t,K}^{n-1}$. We also need to compute the score of $U_{s_k}^n$ at $\vec{X}_{t,k}^{n-1}$ for $k = 1, \ldots, K-1$. We note that we can reuse all of the above score evaluations for both the forward and reverse transition kernels of $P_k^{n-1}$ and $Q_{k-1}^n$. In total, this requires $K + 1$ potential and network evaluations.

- Diff-APT: For $K > 0$, we require $K + 1$ potential and network evaluations following the same logic as for CMCD. For $K = 0$, we require 2 potential and network evaluations for $\pi^{n-1}(X^{n-1})$ and $\pi^n(X^{n-1})$ as we parametrise our annealing path in terms of the target distribution $\pi$ and a neural network.

- PT: We need 2 potential evaluations for $\pi^{n-1}(X^{n-1})$ and $\pi^n(X^{n-1})$ and we do not require any network evaluation.

### E.4 FURTHER DETAILS ON SCALING WITH DIMENSION

For all methods, we take a single step of HMC with step size of $0.03$ and 5 leapfrog steps as our local exploration step across each annealing chain and take 100,000 samples. Additionally, we report the

---

[5]We count this as a single potential evaluation.

(compute-normalised) round trip rate which involves dividing the (compute-normalised) round trips by the number of sampling steps. We note that we use the same methodology as above for computing compute-normalised round trips.

### E.5 Further Details on Log-Normalising Constant (Free-Energy) Estimation

For CMCD-APT, we take a single step of HMC with step size 0.22 and 5 leapfrog steps for our local exploration step across each annealing chain for both DW-4 and MW-32. For Diff-APT, we take two steps of HMC with 5 leapfrog steps and step size of 0.2 and 0.22 respectively for DW-4 and MW-32 for our local exploration step across each annealing chain.

For all methods, we generate 100,000 samples at the target distribution. For each estimate of $\Delta F$, we subsample 1,000 samples uniformly without replacement from the 100,000 to compute the APT free energy estimator. This is then repeated 30 times for each method.

We take the ground truth free energy of DW-4 from estimating $\Delta F$ with the APT estimator using 100,000 samples from PT with 60 chains (after tuning). We take the ground truth free energy of MW-32 from Midgley et al. (2023) which calculates the normalising constant of a single copy of $\hat{\pi}$ numerically allowing for the trivial computation of the overall normalising constant.

### E.6 Further Details on Comparing APT with Neural Samplers

For CMCD, we take our CMCD-APT model on DW-4 with 30 chains and $K = 1, 2, 5$ from Section 6.3, and instead of sampling from these models using APT, we collect 5,000 independent samples from our reference distribution to which we apply our learned kernels $\prod_{n=1}^{N} \prod_{k=1}^{K} P_k^{n-1}$ which we recall are explicitly trained to transport samples from the reference to the target distribution. With our final samples, we then plot the histogram of $d_{ij}$ values for $i \neq j$—specifically, we note that DW-4 represents a distribution over 4 particles in 2D, therefore for each sample from DW-4, we compute the pairwise distances $d_{ij}$ $(i \neq j)$ over these particles in the sample; we then collect all of these values to plot their histogram.

For Diffusion, we follow the same procedure but we tune our annealing schedule with Algorithm 2 applied in the same way as in Appendix E.2 (this step is not required for CMCD as the annealing schedule is required to be fixed during training) before we collect samples by applying $P_k^{n-1}$.

For CMCD-APT and Diff-APT, we simply sample from each method to generate 5,000 samples at the target distribution before plotting the histogram of $d_{ij}$ values.

### E.7 Further Details on Alanine Dipeptide

We run CMCD-APT using 4 chains and $K = 1, 2, 5$. For the local move, we adopt HMC with a dynamic step size: when the acceptance rate is larger than 0.9, we increase the step size by 1.2; if the acceptance rate is smaller than 0.8, we divide the step size by 1.2. Other settings are the same as those of other targets.

### E.8 License

Our implementation is based on the following codebases:

- https://github.com/lollcat/fab-jax (Midgley et al., 2023) (MIT License)
- https://github.com/noegroup/bgflow (MIT License)
- https://github.com/angusphillips/particle_denoising_ diffusion_sampler (Phillips et al., 2024) (No License)
- https://github.com/gerkone/egnn-jax (MIT License)

### E.9 Computing Resources

The experiments conducted in this paper are not resource-intensive. We use a mixture of 24GB GTX 3090 and 80GB A100 GPU, but all experiments can be conducted on a single 80GB A100 GPU.

# F   ADDITIONAL EXPERIMENTAL RESULTS

## F.1   VISUALISATION OF MANYWELL-32

In Figures 5 and 6, we visualise ManyWell-32 samples generated by 1,000 consecutive CMCD-APT and Diff-APT steps with $N = 5, 10, 30$ and $K = 1, 5$ from ManyWell-32 via marginal projections over the first four dimensions. We also show 1,000 independent ground truth samples in Figure 7 for comparison. We choose to report only 1,000 steps to illustrate how fast our sampler mixes. As we can see, the mode weights become more accurate as we increase $N$ and $K$.

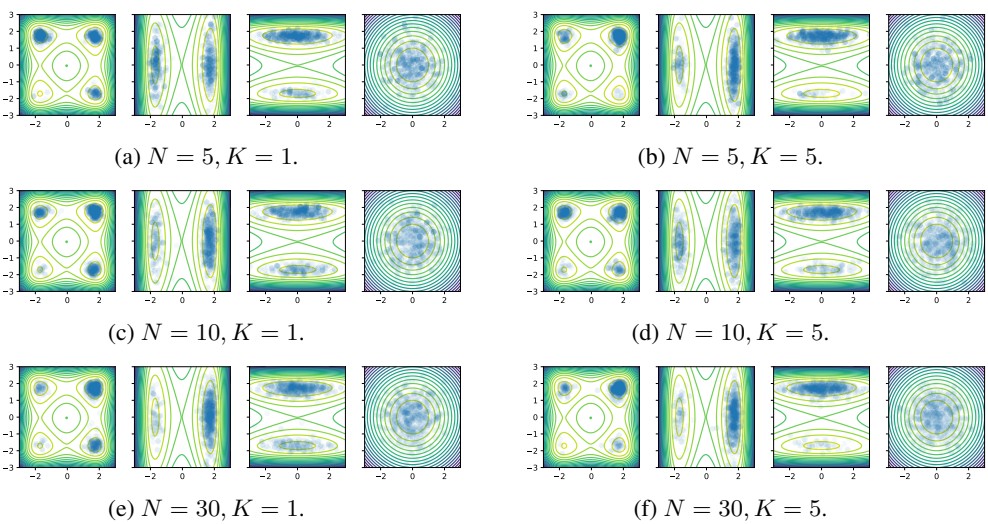

(a) $N = 5, K = 1.$              (b) $N = 5, K = 5.$

(c) $N = 10, K = 1.$              (d) $N = 10, K = 5.$

(e) $N = 30, K = 1.$              (f) $N = 30, K = 5.$

Figure 5: Visualisation of ManyWell-32 samples generated by 1,000 consecutive CMCD-APT steps.

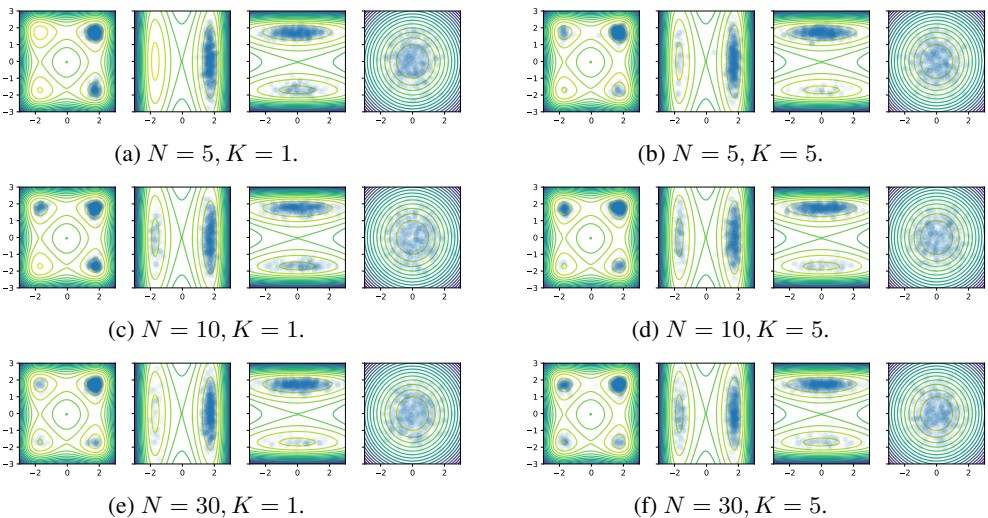

(a) $N = 5, K = 1$.  (b) $N = 5, K = 5$.

(c) $N = 10, K = 1$.  (d) $N = 10, K = 5$.

(e) $N = 30, K = 1$.  (f) $N = 30, K = 5$.

Figure 6: ManyWell-32 samples generated by 1,000 consecutive Diff-APT steps.

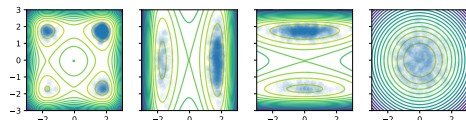

Figure 7: 1,000 independent ground truth samples from ManyWell-32

## F.2 COMPARISON OF ACCELERATION METHODS FOR MANYWELL-32 AND DW-4

We take our trained models from Section 6.3 and provide the same comparison with PT as in Table 1 for ManyWell-32 and DW-4 below.

Table 3: PT versus APT with different acceleration methods, targeting ManyWell-32 in 32 dimensions and standard Gaussian reference using $N = 5, 10, 30$ parallel chains for $T = 100,000$ iterations. For each method, we report the round trips (R), round trips per target evaluation, denoted as compute-normalised round trips (CN-R), the number of neural network evaluations per parallel chain every iteration (Neural Calls), and $\Lambda_K$ estimated using $N = 30$ chains ($\hat{\Lambda}_K$).

| # Chain | | | $N = 5$ | | $N = 10$ | | $N = 30$ | |
|---|---|---|---|---|---|---|---|---|
| Method | Neural Calls ($\downarrow$) | $\hat{\Lambda}_K$ ($\downarrow$) | R ($\uparrow$) | CN-R ($\uparrow$) | R ($\uparrow$) | CN-R ($\uparrow$) | R ($\uparrow$) | CN-R ($\uparrow$) |
| CMCD-APT ($K = 1$) | 2 | 4.384 | 1154 | 577.0 | 2802 | **1401.0** | 4729 | **2364.5** |
| CMCD-APT ($K = 2$) | 3 | 3.827 | 1587 | 529.0 | 3640 | 1213.3 | 5544 | 1848.0 |
| CMCD-APT ($K = 5$) | 6 | **3.148** | 2878 | 479.7 | 4790 | 798.3 | 6678 | 1113.0 |
| Diff-APT ($K = 1$) | 2 | 6.663 | 425 | 212.5 | 2402 | 1201 | 4398 | 2199 |
| Diff-APT ($K = 2$) | 3 | 5.225 | 1387 | 462.3 | 4022 | 1340.7 | 5894 | 1964.7 |
| Diff-APT ($K = 5$) | 6 | 3.94 | **3627** | **604.5** | **5704** | 950.7 | **7634** | 1272.3 |
| Diff-PT ($K = 0$) | 2 | 7.423 | 251 | 125.5 | 1561 | 780.5 | 3440 | 1720 |
| PT | **0** | 5.475 | 550 | 275 | 1879 | 939.5 | 3733 | 1866.5 |

Table 4: PT versus APT with different acceleration methods, targeting DW-4 in 8 dimensions and standard Gaussian reference using $N = 5, 10, 30$ parallel chains for $T = 100, 000$ iterations. For each method, we report the round trips (R), round trips per target evaluation, denoted as compute-normalised round trips (CN-R), the number of neural network evaluations per parallel chain every iteration (Neural Calls), and $\Lambda_K$ estimated using $N = 30$ chains ($\hat{\Lambda}_K$).

| # Chain | | | $N = 5$ | | $N = 10$ | | $N = 30$ | |
|---|---|---|---|---|---|---|---|---|
| Method | Neural Calls ($\downarrow$) | $\hat{\Lambda}_K$ ($\downarrow$) | R ($\uparrow$) | CN-R ($\uparrow$) | R ($\uparrow$) | CN-R ($\uparrow$) | R ($\uparrow$) | CN-R ($\uparrow$) |
| CMCD-APT ($K = 1$) | 2 | 3.173 | 3020 | 1510.0 | 6407 | 3203.5 | 9456 | **4728.0** |
| CMCD-APT ($K = 2$) | 3 | 2.671 | 4239 | 1413.0 | 7549 | 2516.3 | 10538 | 3512.7 |
| CMCD-APT ($K = 5$) | 6 | **2.107** | 6971 | 1161.8 | 9808 | 1634.7 | **12634** | 2105.7 |
| Diff-APT ($K = 1$) | 2 | 4.565 | 4331 | 2165.5 | 7397 | **3698.5** | 7729 | 3864.5 |
| Diff-APT ($K = 2$) | 3 | 3.810 | 7187 | **2395.7** | 10176 | 3392 | 9176 | 3058.7 |
| Diff-APT ($K = 5$) | 6 | 4.358 | **12456** | 2076 | **12740** | 2123.3 | 8104 | 1350.7 |
| Diff-PT ($K = 0$) | 2 | 4.739 | 2962 | 1481 | 5862 | 2921 | 7067 | 3533.5 |
| PT | **0** | 4.016 | 2329 | 1164.5 | 5128 | 2564 | 7610 | 3805 |

