# OpenReview forum: "Accelerated Parallel Tempering via Neural Transports"
_ICLR.cc/2026/Conference — ICLR 2026 Poster_

### Official Review · Reviewer_bR4T · 2025-10-23

**Soundness:** 4
**Presentation:** 3
**Contribution:** 2
**Rating:** 4
**Confidence:** 3

**Summary:**

This manuscript proposes a framework for the application of neural sampling methods to augment Parallel Tempering (PT) based MCMC. Neural samplers like normalizing flows, CMCD, and diffusion models are used to introduce time-inhomogeneous inner-loop Markov processes that accelerate the outer-loop parallel tempering by learning how to make adjacent chains overlap. This resulting procedure Accelerated Parallel Tempering (APT) increases the efficiency of PT in terms of the rate of round trips which is an indicator for the rate of mixing and it conserves the invariance with respect to the target distribution. Theoretical results underpin the conceptual soundness of this method and experimental results demonstrate that it yields practical benefits in several relevant sampling benchmarks.

**Strengths:**

- This method appears to be novel.

- The theoretical results are concise and clear and are related to the experimental results (e.g. connection between N and $\Lambda$)

- The mathematical notation is at an appropriate complexity level, resulting in good readability.

**Weaknesses:**

- Lack of comparison to neural samplers: the authors argue correctly that MCMC like PT and APT enjoy theoretical guarantees that are not provided in neural samplers.
This theoretical distinction is not a sound reason to exclude neural samplers entirely from the experimental comparison. The ultimate goal of these methods is exactly the same as for MCMC, namely to sample unnormalized distribution, and hence neural samplers should not be dismissed entirely based on consideration of asymptotic guarantees. The experiments shown in section 6.4 which compare to neural samplers (CMCD and diffusion samplers) are no fair comparisons since they utilize very different compute budgets (transport maps are concatenated here).

- As argued in the paper the number of round trips is indeed a well-established metric in the context of PT and therefore a well-motivated metric. Comparing to sampling methods more broadly is essential and hence a comparison to other methods should include more general sample quality metrics, like ELBO, Sinkhorn, ESS, log Z. The authors argue “As ESS measures the intertwined performance of the local exploration and swap kernels, we are instead interested in maximising communication between reference and target …”. But such these performance metrics are ultimately crucial to judge the performance of APT relative to other sampling methods.


- l147: the distributions $\pi$ are called potentials


- l206: “Theorem 1 shows we can quantify this  discrepancy in through the rejection rate”. The discrepancy in what?


- Reproducibility suffers from a lack of a code base and corresponding details of the implementation appear to be scattered across many prior works. However, the reviewer did not attempt to reproduce results.

**Questions:**

- What is the intuitive meaning of the global barrier $\Lambda$?

---

> ### Author Response · Authors · 2025-11-25
> **Reply to Reviewer bR4T from the authors**
>
> Thank you very much for the feedback and review score. We’re glad to be able to address any questions about our paper.
>
> ### Weaknesses
>
> > Lack of comparison to neural samplers...
>
> Our motivation behind comparing mainly against PT is that APT is essentially a MCMC algorithm, just like PT. This means APT is fundamentally distinct from neural samplers (or SMC methods), making an apples-to-apples comparison difficult. For instance, APT has mathematical guarantees of asymptotic consistency which means we care about evaluating performance in terms of its convergence rate (as measured by the round trip rate [10]), whereas, neural samplers have no such guarantees. This means people measure performance in terms of how well these methods avoid mode collapse (as measured by the 2-Wasserstein distance for instance). As the criteria for evaluating APT and neural samplers are fundamentally different, we decided it would make the most sense to compare against PT.
>
> Additionally, the different designs of PT, neural samplers, SMC-based methods also brings further difficulty in providing a fair comparison. For instance, APT/PT is constructed to fundamentally leverage parallel computation from the ability to parallelise different chains in a distributed setup. This also allows for implementations of APT to also parallelise training. For example, in NF-APT, we could have trained the normalising flows, that map between neighbouring annealing distributions, in parallel. This computational flexibility is distinct from neural samplers and SMC and makes meaningful direct comparison difficult.
>
> We also stress that PT should not be consider a weak baseline. PT is a state-of-the-art MCMC algorithm and we further strengthen our PT baseline by comprehensively tuning the annealing schedule using the algorithm from [1]. For instance, PT with schedule tuning has been used for extremely hard sampling problems such as generating images of black holes [2]. Moreover, we do provide a careful comparison with this SoTA method by reporting *compute-normalised* round trips that account for the additional sequential computation that a fully distributed implementation of APT performs over fully distributed PT and we still report improved performance for APT.
>
> As for the point that the CMCD and diffusion samplers are not a fair comparison, we would appreciate clarification as we are not sure what the reviewer means here due to the fact that neural samplers usually require sequential computation to sample from them. Additionally, from looking at Figure 4, even with more samples from CMCD and Diffusion, these models still will not be able to correct for the bias due to errors in the learned models (such neural samplers do not have any mechanisms to do so).

---

> > ### Author Response · Authors · 2025-11-25
> > **Reply to Reviewer bR4T from the authors**
> >
> > > As argued in the paper the number of round trips...
> >
> > See above. We also note that we do report log Z values in Figure 3.
> >
> > To provide further metrics and quantification of the performance of APT, we have the following analysis for the target: Alanine Dipeptide. Concretely, we fix the (wall-clock) sampling time (22.2s, this is the time for 10000 steps with standard PT) and we measure the round trip number, the samples collected and multivariate effective sample size (sample count adjusted for autocorrelation) within this time for different $K$:
> >
> > | Metric   |    PT ($N=6$)      |  APT ($N=6, K=1$) | APT ($N=6, K=2$)| APT ($N=6, K=5$) |
> > |----------|:-------------:|------:| ------: | ------: |
> > | Round trips |  17 | 113 | 213        | 385        |
> > | Samples Collected ($\times10^4$) |    10   |   6.2 | 3.9  | 2.3        |
> > | Effective Sample Size in Samples Collected | 1672 |   2274 |  2510       | 2294         |
> >
> >
> > We can see even APT collect less samples given the same wallclock time, it achieves more round trips and has more effective samples than PT. We will add these discussion with additional results in our camera-ready version.
> >
> > Additionally, we politely note that the main purpose of our paper is not necessarily to provide SoTA performance, but to provide the first unified and general framework for incorporating additional flexibility within the swap moves used in PT - i.e. we are introducing a "meta-algorithm" for sampling (from which better sampling algorithms can be constructed from).
> >
> > In particular, we provide the mathematical foundations of APT in how APT can be used for computing expectations and free energies, as well as proving APT has an analogous notion of the global communication barrier $\Lambda_K$ from PT (Theorem 2). This is important as it allows us to port over the analysis from [1] which allows us to characterise the different performance regimes of APT as we vary the number of chains $N$ in terms of $\Lambda_K$, in addition to the tuning algorithm which provides an effective way to tune the annealing schedule $\{\beta_n\}$ which is one of the most important hyper-parameters for APT/PT for performance. We appreciate that the significance of our theoretical work might not be entirely clear to those without familiarity with PT, so we will provide more details on this in the camera-ready version of our paper.
> >
> > > l147: the distributions are called potentials...
> >
> > Thanks you for catching this, we will correct this.
> >
> > > l206: “Theorem 1 shows we can quantify this discrepancy...
> >
> > Here the discrepancy refers to the statistical difference between the measures $\mathbb{P}_K^{n-1}$ and $\mathbb{Q}_K^n$ that represent the forward/backward processes that aim to bridge between neighbouring annealing distributions. Ideally, we want the measures to be the same to ensure the rejection rates between the distributions by APT are minimised. Indeed this informs the design of NF-APT, CMCD-APT, Diff-APT which aim to keep $\mathbb{P}_K^{n-1}$ and $\mathbb{Q}_K^n$ similar.
> >
> > > Reproducibility suffers from a lack of a code base...
> >
> > We will release our codebase upon the paper's acceptance.
> >
> > #### References
> >
> > [1] Non-reversible parallel tempering: a scalable highly parallel MCMC scheme
> >
> > [2] First Sagittarius A* Event Horizon Telescope Results. IV. Variability, Morphology, and Black Hole Mass

---

> > > ### Author Response · Authors · 2025-11-25
> > > **Reply to Reviewer bR4T from the authors**
> > >
> > > ## Conclusion
> > >
> > > Please let us know if any additional information or clarifications would be helpful. If we have adequately addressed your points, we hope you will consider increasing your score.

---

> > > > ### Comment · Reviewer_bR4T · 2025-11-26
> > > >
> > > > Thanks for addressing my concerns.
> > > >
> > > > In the context of the first weakness, the authors reply: “As for the point that the CMCD and diffusion samplers are not a fair comparison, we would appreciate clarification as we are not sure what the reviewer means here because neural samplers usually require sequential computation to sample from them.”
> > > >
> > > > This is of course correct, but my concerns are that comparing Diff-APT and CMCD-APT to Diffusion and CMCD is not computationally fair. The APT variants are computationally more expensive, and this should be mentioned. Also, it appears to be potentially misleading to call these ablated versions of Diff-APT and CMCD-APT simply Diffusion and CMCD, respectively. If I understand l446 ff correctly, these non-APT models originate from training of the corresponding APT-variants. Hence, they do not represent usual Diffusion or CMCD samplers as known in the literature.
> > > >
> > > > "Additionally, from looking at Figure 4, even with more samples from CMCD and Diffusion, these models still will not be able to correct for the bias due to errors in the learned models (such neural samplers do not have any mechanisms to do so)."
> > > >
> > > > Indeed, and I would suggest including this argument - this would make the discussion of the results in Fig. 4 more nuanced and precise than the current statement in l450: "As we can see, directly using the learned neural sampler dramatically drops performance...".

---

> > > > > ### Author Response · Authors · 2025-11-26
> > > > > **Reply to Reviewer bR4T from the authors**
> > > > >
> > > > > > my concerns are that comparing Diff-APT and CMCD-APT to Diffusion and CMCD is not computationally fair...
> > > > >
> > > > > Diff-APT, CMCD-APT indeed do require more NFEs than Diffusion and CMCD for sampling, however, we note that APT is designed to leverage parallel computation (something that Diffusion or CMCD cannot do) for both training and sampling. This can make sampling from Diff-APT, CMCD-APT faster than sampling from their neural sampler variant. For example, sampling from CMCD in Figure 4 requires $O(29K)$ sequential steps whereas if CMCD-APT is fully parallelised, sampling only requires $O(K)$ sequential steps (on each machine). Additionally, if we did draw more samples from Diffusion, CMCD to match the total number of NFEs used by Diff-APT, CMCD-APT, Figure 4 still suggests that this cannot overcome the bias within the neural samplers.
> > > > >
> > > > > We will make this trade-off between total NFEs and parallelisation clearer in the camera-ready version of our paper.
> > > > >
> > > > >
> > > > > > Hence, they do not represent usual Diffusion or CMCD samplers as known in the literature...
> > > > >
> > > > > We note that the diffusion model trained and used by Diff-APT is just a standard diffusion model trained with the standard score matching objective (albeit trained with data sampled iteratively). Therefore, we consider this underlying model as a standard neural sampler directly.
> > > > >
> > > > > As for CMCD-APT, we are essentially training individual CMCD models to transport between neighbouring annealing distributions $\pi^{\beta}, \pi^{\beta'}$, instead of directly from the reference $\pi^0$ to the target $\pi^1$, which is an easier learning problem (the fact that we can train with the symmetric KL should also improve things). This suggests that the concatenation of these individual CMCD models should perform on par or better than the original CMCD setup.
> > > > >
> > > > > However, we do agree that "CMCD" in Section 6.4 does not refer to the exact setup from [3], and we will make this clearer in the camera-ready version of our paper.
> > > > >
> > > > >
> > > > > > Indeed, and I would suggest including this argument...
> > > > >
> > > > > We thank the reviewer for this suggestion, we agree that this would be more nuanced and precise. We will update this in the camera-ready version of our paper.
> > > > >
> > > > >
> > > > > > What is the intuitive meaning of the global barrier...
> > > > >
> > > > > We're sorry for forgetting to give an answer to this initial question. See below for our response.
> > > > >
> > > > > From the theory of [1] (for standard PT), it can be shown that the rejection rates can be viewed as metric that measures the discrepancy between neighbouring annealing distributions (also see line 209 in the updated paper for the analogous case in APT). Moreover, it can be shown that these rejection rates can be integrated along the annealing path (in the limit as $N\to\infty$) and equals the *global barrier* $\Lambda$ for PT. This is a statistical invariant of the annealing path that captures the inherent statistical difficulty of swapping particles along the path. For example, it can be shown that we require $N>\Lambda$ in order for stable performance from PT.
> > > > >
> > > > > Moreover, this forms the theoretical basis for the schedule tuning algorithm in [1] (see Section 5) that automatically tunes the schedule $\{\beta_n\}$ from the estimated rejection rates when running PT to ensure that the global barrier of the annealing path between neighbouring distributions $\pi^{\beta_{n-1}}$ and $\pi^{\beta_n}$ is uniform along the path. This ensures that the rejection rates along the path are uniformly distributed and mitigates any bottlenecks to mixing that poor rejection rates/poorly chosen schedules might induce.
> > > > >
> > > > > The above discussion provides the motivation of why Theorem 2 is important (and more generally Section 3.2 and 4). It generalises the notion of rejection rates and $\Lambda$ to the APT setting which allows us to import the same theory over to APT. For example, this allows us to use the same schedule tuning algorithm in our paper for APT. We agree that this could be made clearer and we will provide further discussion of this point in the camera-ready version of our paper.
> > > > >
> > > > >
> > > > > ### Additional References
> > > > >
> > > > > [3] Transport meets Variational Inference: Controlled Monte Carlo Diffusions

---

> > > > > > ### Author Response · Authors · 2025-11-26
> > > > > > **Reply to Reviewer bR4T from the authors**
> > > > > >
> > > > > > We thank the reviewer again for their time and effort in the reviewing process and helping us to improve our paper. We hope this helps to clarify the points raised. If we have adequately addressed your concerns, we hope you will consider increasing your score.

---

> > > > > > > ### Comment · Reviewer_bR4T · 2025-11-27
> > > > > > >
> > > > > > > This settles my point about Fig. 4. Thanks for raising my initial point about the global barrier. In fact I have already read your explanations of this point in your response to Reviewer 93BL.
> > > > > > > I point out that the reference you provide above in this thread is wrong, it should be:
> > > > > > >
> > > > > > > [1] Non-reversible parallel tempering: A scalable highly parallel MCMC scheme.
> > > > > > >
> > > > > > > I will raise my score and I do recommend this paper to be accepted since it represents an overall technically solid and novel contribution.

---

> > > > > > > > ### Author Response · Authors · 2025-11-27
> > > > > > > > **Reply to Reviewer bR4T from the authors**
> > > > > > > >
> > > > > > > > We're glad to have been able to address your questions. We greatly thank the reviewer for updating their rating and for their support of our paper.

---

### Official Review · Reviewer_WMqg · 2025-10-24

**Soundness:** 4
**Presentation:** 4
**Contribution:** 4
**Rating:** 8
**Confidence:** 4

**Summary:**

The present paper proposes to accelerate the parallel tempering (PT) algorithm using learned transport processes between neighboring distributions in the annealing ladder. The approach draws inspiration from the statistical mechanics literature (works by Jarzynski and collaborators) and a single step version with normalizing flows (Invernizzi et al 2022). It bears similarities with previously proposed approaches mixing deep learning and Sequential Monte Carlo, namely the Annealed Flow Transport (Arbel et al 2021), NETs (Albergo et al 2024) and CMCD (Vargas et al 2024).

The proposed approach is presented in a general framework along with different concrete variants of the algorithm leveraging either normalizing flows, diffusion models or stochastic control. The algorithm is shown to yield consistent estimators of expectations and normalization constants. A theoretical analysis of performances is given under symplifying assumptions. A set of numerical experiments demonstrates a possible advantage of the method over vanilla parallel tempering and compares the different variant proposed. Finally, limitations are discussed in a short conclusion where, in particular, the need to take into account the computational cost of neural networks in future works is acknowledged by the authors.

**Strengths:**

- Although related to recent literature and not unexpected in this respect, the proposed algorithm is novel and the authors do a great job at presenting in general the method before showcasing different possible applications with different types of generative modeling ideas.
- The article also does a great job at connecting its method to the adjacent literature.
- The method is justified by proofs of consistency. I have not read in details the proofs, but the result appear reasonable and the adequate physics literature is cited.
- A theoretical analysis of performance for edge cases is also given.
- The numerical section spans different examples, notably in relatively high dimension, even investigating the impact of dimension increase in a synthetic experiment.

**Weaknesses:**

- It would be desirable to insist more on the question of the training of the neural networks in the main text and on the fact that it is not a trivial question in this sampling setting. For instance, for NF-APT, the authors state in Appendix C.1.2 that the retained strategy is to first run PT to be able to train the flows, which is arguably an important limitation.

- The introduction is not always fair to the adjacent literature
	- line 072 - “However, these methods usually incur a bias, foregoing theoretical guarantees of MCMC, and can be expensive to implement and train.” Is not true of the discrete flows approaches that are cited by the authors , nor of NETs (Albergo et al 2024).

	- line 095 - “By contrast, our framework leverages normalising flows to facilitate exchanges between all neighbouring temperature levels, thereby enhancing sampling efficiency across the entire annealing path and providing a more stable training objective” - the ambition of Invernizzi et al. is to drastically simplify the procedure by avoiding the necessity to have many replicas. As such, this last sentence is unclear, and the claim for the increased stability of the training objective needs to be substantiated. If I understand correctly this reference is implementing NF-APT with N = 1?


Minor:
- It would be worth mentioning [Noble2025] in section 5.3, as this reference already proposes to learn energy-based models along a noising process to ease sampling, although this reference was exploiting it in a different way.
- l103 the definition of $Z_n$ is lacking the exponent $n$ to $U$.
- l190 “we use using”
- In Eq 4, maybe restate that the brackets are notations for an expectation. Also this is a common notation in the statistics/Monte Carlo literature, it is not usually encountered in the machine learning literature.
- l215: “To also obtain free energy estimates, by averaging …” this sentence has no main proposition.
- The notation $\tau$ in proposition 250 is not introduced, I am guessing this the round trip rate.
- l262 “a good neural network approximation” it would be a good idea to state a good approximation of what if the authors want to make this point at this stage, otherwise it can wait for the next section.

[Noble2025] Noble, Maxence, Louis Grenioux, Marylou Gabrié, and Alain Oliviero Durmus. “Learned Reference-Based Diffusion Sampler for Multi-Modal Distributions.” Paper presented at The Thirteenth International Conference on Learning Representations. April 4, 2025. https://openreview.net/forum?id=fmJUYgmMbL.

**Questions:**

-

---

> ### Author Response · Authors · 2025-11-25
> **Reply to Reviewer WMqg from the authors**
>
> We sincerely thank the reviewer for their positive assessment and feedback. We are grateful for the opportunity to clarify the points raised.
>
> ## Weaknesses
>
> > It would be desirable to insist more on the question of the training of the neural networks...
>
> We agree with this point that an important direction of future research for APT is to improve the training of the neural samplers involved. However, we note that this is an issue with the area of neural samplers more broadly. For instance, [1] also uses an iterative approach for training, and the variational-based training of [2] requires differentiating through the SDE transporting from the reference to the target distribution which can suffer from mode collapse issues and expensive training.
>
> We would also like to point out the benefits for training that APT allows for which is distinct from other neural samplers. Due to the fact that APT leverages parallel computation, from the ability to parallelise over chains, we could implement NF-APT, CMCD-APT, Diff-APT to train each neural sampler transporting between neighbouring distributions $\pi^{\beta}, \pi^{\beta'}$ in parallel, potentially reducing wall-clock time, and reducing the difficulty of the learning problem. For example, in CMCD-APT, we only need to learn to transport between neighbouring distributions, instead of directly from the reference to the target, which should make training easier. Also, NF-APT has the benefit of using the symmetric KL for training, instead of the reverse KL usually required, since we have access to samples from both $\pi^{\beta}, \pi^{\beta'}$, which mitigates mode collapse issues usually seen.
>
>
> > line 072 - “However, these methods usually incur a bias...
>
> We thank the reviewer for catching this issue. We will correct this in the camera-ready version of our paper.
>
> > line 095 - “By contrast, our framework leverages normalising flows...
>
> This reference is somewhat similar to NF-APT with $N=1$, however, there are some important differences. Mainly, they assume that they already have a good number of samples at the target for training and try to train their normalising flow to directly map from the reference to target. We note that this is a much more difficult learning problem that what APT does from scaffolding the sampling problem with the annealing path - i.e. APT simplifies the learning problem from only needing to transport between neighbouring distributions.
>
>
> ### Minor
>
> We thank the reviewer for catching these typos and for the suggestions. We have corrected these in our updated version of our paper.
>
> > The notation in proposition 250 is not introduced...
>
> We note that we introduce $\tau$ (round trip rate) in line 242 (in the updated version of the paper), but we appreciate the point that $\tau(\mathbb{P}^{0:N-1}_K,\mathbb{Q}^{1:N}_K)$ is not clearly defined. We introduce the dependency on $\mathbb{P}^{0:N-1}_K,\mathbb{Q}^{1:N}_K$ to indicate we are considering the round trip rate of APT under the choice of $\mathbb{P}^{0:N-1}_K,\mathbb{Q}^{1:N}_K$.
>
>
> ## References
>
> [1] Particle Denoising Diffusion Sampler
>
> [2] Transport meets Variational Inference: Controlled Monte Carlo Diffusions

---

> > ### Comment · Reviewer_WMqg · 2025-11-27
> > **Thanks for your reply**
> >
> > I have no remaining concerns.

---

> > > ### Author Response · Authors · 2025-11-27
> > > **Reply to Reviewer WMqg from the authors**
> > >
> > > We greatly thank the reviewer again for their support of our paper. We're glad to have been able to address your questions.

---

### Official Review · Reviewer_93BL · 2025-10-26

**Soundness:** 3
**Presentation:** 1
**Contribution:** 1
**Rating:** 2
**Confidence:** 4

**Summary:**

The paper proposes an extension of Parallel Tempering (PT) called Accelerated Parallel Tempering (APT), in which the standard swap operation is enhanced with deterministic or stochastic transition kernels. These kernels are trained adaptively using samples generated by APT within a data-driven objective. The deterministic variant employs normalizing flows (NF-APT), while the stochastic variants derive from forward–backward discretizations of controlled annealed Langevin dynamics (CMCD-APT) or from diffusion models, where the forward process is exact and the reverse process is obtained by integrating the learned reverse SDE (Diff-APT). The sequence of intermediate potentials is user-defined for NF-APT and CMCD-APT (typically along a tempering path) and learned automatically in Diff-APT. Experimental results on synthetic benchmarks and particle systems demonstrate that APT accelerates sampling compared to standard parallel tempering.

**Strengths:**

* The paper addresses an important problem and demonstrates clear improvements in round-trip rates for parallel tempering.

**Weaknesses:**

* The paper is very poorly written. The presentation based on the Jarzynski framework makes it difficult to follow. A formulation directly grounded in the Metropolis–Hastings framework, with clearly defined target distribution and proposal kernel (see [1]), would greatly improve readability and conceptual clarity.
* The novelty is rather limited. The deterministic case has already been covered in [2] (as acknowledged by the authors), while the stochastic variant represents only a modest generalization of prior work on AIS [3,4] and SMC methods [5], against which no comparisons are provided.
* The proposed method does not address one of the core limitation of PT in multi-modal settings : mode switching. It is well known that along tempering paths, probability mass tends to shift abruptly between distinct high-probability regions [5,6], severely hindering mixing and performance in both PT and SMC. Even with accelerated swaps in NF-APT or CMCD-APT, the transition kernels remain local refinements (as shown in the left panel of Fig. 1) and cannot overcome this issue. Similarly, Diff-APT is affected because its learned marginal distributions rely on score matching, which is known to be mode-blind [7,8,9], causing the learned path to exhibit mode switching as well. Consequently, the method provides limited benefit in genuinely multi-modal settings. This issue becomes more pronounced in higher dimensions (likely explaining the use of a "perfect" path in Section 6.2). It would be informative to report results with a learned path. While increasing $K$ and reducing $N$ might partially mitigate this, it does not fundamentally resolve the scalability issue.
* The evaluation is quite narrow. The round-trip rate is the primary quantitative metric used (except for Fig. 3, which reports free-energy differences, and Figs. 4 and 6–7, which are purely qualitative). This metric alone may obscure important behaviors such as mode switching and limits meaningful comparison to standard PT. The justification for this metric relies solely on [10], which claims PT outperforms neural samplers. Moreover, the only non-PT baseline compared is CMCD, which is known to perform poorly [10,11,12].
* The target-informed parametrization (L1400) is known to substantially restrict the expressivity of the energy-based model [13] and to introduce significant computational overhead [10], further reducing the appeal of this approach.

[1] Syed, S., Bouchard-Côté, A., Deligiannidis, G., & Doucet, A. (2022). Non-reversible parallel tempering: A scalable highly parallel MCMC scheme. Journal of the Royal Statistical Society: Series B (Statistical Methodology), 84(2), 321–350.

[2] Invernizzi, M., Krämer, A., Clementi, C., & Noe, F. (2022). Skipping the Replica Exchange Ladder with Normalizing Flows. The Journal of Physical Chemistry Letters, 13(50), 11643–11649.

[3] Zhang, F., He, J., Midgley, L., Antorán, J., & Hernández-Lobato, J. (2024). Efficient and unbiased sampling of boltzmann distributions via consistency models. arXiv preprint arXiv:2409.07323.

[4] Fengzhe Zhang, Laurence I. Midgley, & José Miguel Hernández-Lobato. (2025). Efficient and Unbiased Sampling from Boltzmann Distributions via Variance-Tuned Diffusion Models.

[5] Phillips, A., Dau, H.D., Hutchinson, M., De Bortoli, V., Deligiannidis, G., & Doucet, A. (2024). Particle Denoising Diffusion Sampler. In Proceedings of the 41st International Conference on Machine Learning (pp. 40688–40724). PMLR.

[6] Bálint Máté, & François Fleuret (2023). Learning Interpolations between Boltzmann Densities. Transactions on Machine Learning Research.

[7] Wenliang, L., & Kanagawa, H.. (2021). Blindness of score-based methods to isolated components and mixing proportions.

[8] Zhang, M., Key, O., Hayes, P., Barber, D., Paige, B., & Briol, F.X. (2022). Towards Healing the Blindness of Score Matching. In NeurIPS 2022 Workshop on Score-Based Methods.

[9] Shi, Z., Yu, L., Xie, T., & Zhang, C.. (2024). Diffusion-PINN Sampler.

[10] Jiajun He, Yuanqi Du, Francisco Vargas, Dinghuai Zhang, Shreyas Padhy, RuiKang OuYang, Carla P Gomes, & José Miguel Hernández-Lobato (2025). No Trick, No Treat: Pursuits and Challenges Towards Simulation-free Training of Neural Samplers. In Frontiers in Probabilistic Inference: Learning meets Sampling.

[11] Junhua Chen, Lorenz Richter, Julius Berner, Denis Blessing, Gerhard Neumann, & Anima Anandkumar (2025). Sequential Controlled Langevin Diffusions. In The Thirteenth International Conference on Learning Representations.

[12] Noble, M., Grenioux, L., Gabrié, M., & Durmus, A. (2025). Learned Reference-based Diffusion Sampler for multi-modal distributions. In The Thirteenth International Conference on Learning Representations.

[13] Jiajun He, Yuanqi Du, Francisco Vargas, Yuanqing Wang, Carla P. Gomes, José Miguel Hernández-Laobato, & Eric Vanden-Eĳnden. (2025). FEAT: Free energy Estimators with Adaptive Transport.

**Questions:**

* In the introduction, the references to Noé et al. (2019), Midgley et al. (2023), and Gabrié et al. (2022) suggest that these methods "incur a bias" similar to the previously mentioned approaches. This interpretation is incorrect. The earlier works (except iDEM) perform variational inference to learn a generative model without data, where the bias arises from optimization and model misspecification. In contrast, the three cited works embedded the models within Monte Carlo schemes - such as IS, AIS, or MCMC - which, in the infinite-particle/chain length limit, correct the bias of the learned model. Their residual bias is purely statistical. Grouping these methods together is therefore misleading.
* The training procedures are described very vaguely. What are the exact loss functions used, and how are they implemented in practice? How many times is the adaptation loop repeated (i.e., how many optimizations of the data-based loss are performed) ?
* Could you visualize the learned diffusion potentials path over time for a simple mixture of two one-dimensional Gaussian distributions?
* Comparisons with standard AIS, SMC, PT (under an equivalent computational budget, including training), and other multimodal samplers (potentially ML-enhanced) on target-specific metrics would be highly informative.
* For Diff-APT, could you first train an unconstrained energy-based model and then learn an auxiliary function analogous to $r_{\theta}$ a posteriori? This might help avoid the expressivity constraints imposed by the current formulation.
* For Diff-APT, why not use the target score matching loss [A], similar to what PDDS employs [B] ?
* What mechanisms prevent the adaptive procedure from suffering from mode collapse, particularly given the mode-switching phenomenon? This concern is especially relevant since the loss functions include (at least partially) a reverse KL term.
* According to Appendix D, only a single HMC step is performed for the local exploration move, which seems unrealistically low. Could you provide an ablation study on the swap frequency, i.e., the number of interleaved local steps between swaps?
* The ManyWell-32 experiment in Appendix E.1 claims to demonstrate recovery of mode weights, yet no quantitative metrics are reported. Why not use the benchmark from [C], which is specifically designed to evaluate this capability?
* How are the chains initialized in PT/APT?
* The total number of MCMC steps (50k or 100k) is extremely large. Combined with the adaptive training loop, this implies a substantial computational cost. Could you (i) report actual wall-clock runtimes on a specific hardware setup and (ii) provide an ablation on the total number of MCMC steps?
* In Fig. 4, how do you explain the large performance gap between Diffusion and Diff-APT? If Diffusion is trained using samples from Diff-APT, this discrepancy seems counterintuitive.
* For ALDP, how are the chains initialized, and how do you sample from the $T = 1200K$ reference distribution?
* For ALDP, is the energy parametrization (L1400) rotation-invariant? Additionally, how do you ensure that both the transition kernels and the energy function are defined in the center-of-mass (CoM) space, i.e., the mass-centered coordinate system?

[A] Valentin De Bortoli, Michael Hutchinson, Peter Wirnsberger, & Arnaud Doucet. (2024). Target Score Matching.

[B] Phillips, A., Dau, H.D., Hutchinson, M., De Bortoli, V., Deligiannidis, G., & Doucet, A. (2024). Particle Denoising Diffusion Sampler. In Proceedings of the 41st International Conference on Machine Learning (pp. 40688–40724). PMLR.

[C] Grenioux, L., Noble, M., & Gabrie, M. (2025). Improving the evaluation of samplers on multi-modal targets. In Frontiers in Probabilistic Inference: Learning meets Sampling.

---

> ### Author Response · Authors · 2025-11-25
> **Reply to Reviewer 93BL from the authors**
>
> We thank the reviewer for their careful reading and many concrete suggestions. Below, we will clarify the points raised in the review.
>
> ## Weaknesses
> > The paper is very poorly written...
>
> We note that our intention from presenting our paper using the Jarzynski framework was to help people from other areas, such as computational physics and chemistry, that also heavily use sampling techniques to understand our methodology. We appreciate that this notation is less familiar to a machine learning audience and we have currently updated our paper to be more grounded in the Metropolis–Hastings notation. For example, see Equation 1, 2 in our updated version.
>
>
> > The novelty is rather limited...
>
> We respectfully strongly disagree with this characterisation. We believe it is quite dismissive to simply judge our contribution as only a modest generalisation of [1, 2, 3] or saying APT has already been covered by [4].
>
> This completely overlooks the contribution of the paper as providing the first unified and general framework for incorporating additional flexibility within the swap moves used in PT. In particular, we provide the mathematical foundations of APT in how APT can be used for computing expectations and free energies, as well as proving APT has an analogous notion of the global communication barrier $\Lambda_K$ from PT (Theorem 2). This is important as it allows us to port over the analysis from [5] which allows us to characterise the different performance regimes of APT as we vary the number of chains $N$ in terms of $\Lambda_K$, in addition to the tuning algorithm which provides an effective way to tune the annealing schedule $\{\beta_n\}$ which is one of the most important hyper-parameters for APT/PT for performance. We appreciate that the significance of our theoretical work might not be entirely clear to those without familiarity with PT, so we will provide more details on this in the camera-ready version of our paper.
>
> Finally, we explore the design space of APT, providing concrete implementations of APT using normalising flows, diffusions and neural samplers, and we provide careful benchmarking of APT against PT.
>
> We believe this is a novel and useful contribution to the literature that goes beyond [4] which doesn't provide any mathematical analysis, only considers a single PT chain and only uses normalising flows.
>
> Moreover, from looking at the papers of [1, 2, 3], it is unclear to us how exactly the reviewer is able to conclude that our work clearly follows. For example, [2] investigates improving importance sampling from diffusion models by tuning the variance when discretising the backward diffusion SDE. This is totally separate from the contributions of our paper. In addition, PT is a very different algorithm and is relatively unknown compared to importance sampling/SMC in the machine learning literature, hence, we believe it is unfair to claim that taking some inspiration from work in SMC and applying this to PT is only a modest contribution (especially with the aforementioned theory).
>
> As for the point on comparing against AIS, SMC etc., see the response to "The evaluation is quite narrow..." below.

---

> > ### Author Response · Authors · 2025-11-25
> > **Reply to Reviewer 93BL from the authors**
> >
> > > The proposed method does not address one of the core limitation of PT in multi-modal settings : mode switching...
> >
> > Firstly, we note that mode switching is a phenomenon associated with the use of the linear annealing path, not with PT. We politely stress that this indicates a misunderstanding of how PT and APT functions.
> >
> > Indeed, mode switching is much more of an issue with neural samplers and SMC compared with PT. This is due to the fact that PT is an MCMC method that uses Metropolis–Hastings corrections to ensure that modes are sampled according to the correct proportion and the fact that samples along different chains can move bidirectionally along the annealing path. Therefore, if mode switching occurs between $\pi^\beta$ and $\pi^{\beta'}$ ($\beta<\beta'$), PT will suffer from having a high rejection rate for this swap but it will not assign incorrect probability mass (instead the samples will be recycled in swaps in different directions). In contrast, neural samplers (such as flow-based methods) and SMC, due to the fact that they propagate samples in a single direction and do not have any Metropolis–Hastings correction mechanisms, will propagate samples that misassign probability mass when moving from $\pi^\beta$ to $\pi^{\beta'}$ via their transition kernels.
> >
> > Hence, we see that PT does not necessarily suffer from the mode collapse issue, induced by mode switching, that affects neural samplers and SMC. Instead, mode switching only affects the mixing rate of PT.
> >
> > Nevertheless, we can see that the presence of mode switching can be captured by the rejection rates computed along the annealing path by PT. From the theory of [5], it can be shown that the rejection rates can be integrated along the annealing path (in the limit as $N\to\infty$) and equals the *global barrier* $\Lambda$ for PT. This is a statistical invariant of the annealing path that captures the inherent statistical difficulty of swapping particles along the path. For example, it can be shown that we require $N>\Lambda$ in order for stable performance from PT.
> >
> > Moreover, this forms the theoretical basis for the schedule tuning algorithm in [5] (see Section 5) that automatically tunes the schedule $\{\beta_n\}$ from the estimated rejection rates when running PT to ensure that the global barrier of the annealing path between neighbouring distributions $\pi^{\beta_{n-1}}$ and $\pi^{\beta_n}$ is uniform along the path. This ensures that the rejection rates along the path are uniformly distributed and mitigates any bottlenecks to mixing that mode switching might induce.
> >
> > Therefore, we see that properly tuned PT is resilient to mode switching compared to neural samplers and SMC.
> >
> > The above discussion provides the motivation of why Theorem 2 is important. It generalises the notion of $\Lambda$ to the APT setting which allows us to import the same theory over to APT. This allows us to use the same schedule tuning algorithm in our paper to mitigate against mode switching.
> >
> > In addition, the comment "Even with accelerated swaps in NF-APT or CMCD-APT..." is incorrect when properly tuning APT (which we mention that we always use in the Appendix) and the comment about "transition kernels being local refinements" is also wrong as NF-APT can learn arbitrary bijections that do not need to be local.
> >
> > Finally, the discussion on the mode blindness of score matching is a valid criticism of Diff-APT but we do not claim in the paper that Diff-APT is the de facto way of using APT. The aim of our paper is to provide a general framework for constructing more flexible PT samplers. We agree that it is an important future direction of research to design better ways of learning energy-based score models. For instance, we could use [7, 8] to help regularise our score matching objective.

---

> > > ### Author Response · Authors · 2025-11-25
> > > **Reply to Reviewer 93BL from the authors**
> > >
> > > > The evaluation is quite narrow...
> > >
> > > Our motivation behind comparing mainly against PT is that APT is essentially a MCMC algorithm, just like PT. This means APT is fundamentally distinct from neural samplers (or SMC methods), making an apples-to-apples comparison difficult. For instance, APT has mathematical guarantees of asymptotic consistency which means we care about evaluating performance in terms of its convergence rate (as measured by the round trip rate [10]), whereas, neural samplers have no such guarantees. This means people measure performance in terms of how well these methods avoid mode collapse (as measured by the 2-Wasserstein distance for instance). As the criteria for evaluating APT and neural samplers are fundamentally different, we decided it would make the most sense to compare against PT.
> > >
> > > Additionally, the different designs of PT, neural samplers, SMC-based methods also brings further difficulty in providing a fair comparison. For instance, APT/PT is constructed to fundamentally leverage parallel computation from the ability to parallelise different chains in a distributed setup. This also allows for implementations of APT to also parallelise training. For example, in NF-APT, we could have trained the normalising flows, that map between neighbouring annealing distributions, in parallel. This computational flexibility is distinct from neural samplers and SMC and makes meaningful direct comparison difficult.
> > >
> > > We also stress that PT should not be consider a weak baseline. PT is a state-of-the-art MCMC algorithm and we further strengthen our PT baseline by comprehensively tuning the annealing schedule using the algorithm from [1]. For instance, PT with schedule tuning has been used for extremely hard sampling problems such as generating images of black holes [2]. Moreover, we do provide a careful comparison with this SoTA method by reporting *compute-normalised* round trips that account for the additional sequential computation that a fully distributed implementation of APT performs over fully distributed PT and we still report improved performance for APT.
> > >
> > > As for the point raised that the justification for using round trips is due to [9], we are not sure how the reviewer has concluded this. Also, we are not sure that we understand the reviewer's claim that focusing on round trips obscures comparisons with PT, since the round trip rate is a gold-standard metric for analysing PT-type algorithms in the literature.
> > >
> > > To provide more details on the performance of APT that could address the reviewer's concerns, we provide a more practical analysis of the sampling computational costs measured by wallclock time for the target: Alanine Dipeptide. Concretely, we fix the sampling time (22.2s, this is the time for 10,000 steps with standard PT on a RTX 3090) and we measure the round trip number, the samples collected and multivariate effective sample size (sample count adjusted for autocorrelation) within this time for different $K$:
> > >
> > > | Metric   |    PT ($N=6$)      |  APT ($N=6, K=1$) | APT ($N=6, K=2$)| APT ($N=6, K=5$) |
> > > |----------|:-------------:|------:| ------: | ------: |
> > > | Round trips |  17 | 113 | 213        | 385        |
> > > | Samples Collected ($\times10^4$) |    10   |   6.2 | 3.9  | 2.3        |
> > > | Effective Sample Size in Samples Collected | 1672 |   2274 |  2510       | 2294         |
> > >
> > >
> > > We can see even though APT collect less samples given the same wallclock time, it achieves more round trips and has more effective samples than PT. We will add these discussions with additional results in camera-ready version of our paper.

---

> ### Author Response · Authors · 2025-11-25
> **Reply to Reviewer 93BL from the authors**
>
> > The target-informed parametrization...
>
> We note that this is standard practice in the area and we agree that an important future direction of the field is to derive parametrisation that are not required to be target-informed.
>
> Also, we note that the primary focus of the paper is on providing a unified and general framework for incorporating additional flexibility in PT, instead of designing new parametrisations. In addition, we note that NF-APT does not suffer from this issue due to using normalising flows instead.
>
> ## Questions
>
> > In the introduction, the references...
>
> We thank the reviewer for catching this issue. We agree and will update our paper in the camera-ready version to reflect this.
>
> > The training procedures are described very vaguely...
>
> We provide details on our objectives in Section 5. These are all standard loss functions from the literature. We will update our paper in the camera-ready version to provide further explicit forms of the KL divergences and the standard score matching objective that we use.
>
> As for the number of iterations of our adaptation loop, we appreciate that this could be made clearer in the Appendix. For NF-APT, we repeat twice; for CMCD-APT, we repeat twice; for Diff-APT, we repeat twice for GMM-10, repeat three times for DW-4 and repeat four times for MW-32.
>
> > Could you visualize the learned diffusion potentials path...
>
> See https://ibb.co/zVhRn8sG. This is for the target: $0.3 * \mathcal{N}(-0.3, 0.2^2) + 0.7 * \mathcal{N}(0.5, 0.1^2)$.
>
>
> > Comparisons with standard AIS, SMC, PT...
>
> See our response to "The evaluation is quite narrow...".
>
> > For Diff-APT, could you first train an unconstrained energy-based model...
>
> See our response to "The target-informed parametrization...". Also, it is not clear what objective could be used to learn $r_\theta$.
>
>
> > For Diff-APT, why not use the target score matching loss...
>
> In our experiments, we didn't really find that the target score matching loss helped with training our diffusion model. Additionally, by using the standard score matching loss, we also reduce the number of target evaluations required.
>
> > What mechanisms prevent the adaptive procedure from suffering from mode collapse..
>
> See our response to "The proposed method does not address one of the core limitation of PT in multi-modal settings : mode switching..."
>
> Also, an advantage of NF-APT is that we are able to train using the symmetric KL objective, which mitigates mode collapse issues associated the reverse KL objective, due to the fact that APT provides samples from both $\pi^{\beta}, \pi^{\beta'}$. In contrast, SMC-based methods can only train with the reverse KL objective as they only have access to samples from a single direction which could further produce mode collapse issues.
>
>
> > According to Appendix D, only a single HMC step is performed for the local exploration move...
>
> We agree that only using a single HMC is unrealistically low, however this was chosen not to maximise performance but to try to isolate the effect of our swap moves in APT and PT in determining performance. With more swaps, we can expect to see improvements in the performance of our samplers from reducing the corrolation between samples. To see an ablation of this for PT, see Section 7.2 in [5].
>
> > The ManyWell-32 experiment in Appendix E.1...
>
> We note that the purpose of this section was to provide a visualisation of our sampling performance from showing different slices of our samples and density landscape. We also report normalising constant estimation performance for this target in Figure 3 showing that APT can recover the ground-truth normalising constant suggesting that we do recover mode weights.
>
> Also, we did not use the suggested paper, as we were not aware of it and it does not appear to have consensus in the community for benchmarking sampling methods.
>
> > How are the chains initialized in PT/APT?
>
> In general, we initialise all chains with samples from the prior distribution (see below for the case of ALDP).
>
> > The total number of MCMC steps (50k or 100k) is extremely large...
>
> As an example, training Diff-APT on DW-4 requires 30 minutes on a single A40. To gain a sense of how long other methods take, we note that [11] requires 1.4/4.3 hours for training on DW-4 (where they use a single A100 (40 GB) which is a much more powerful GPU). We also note that we do not fully exploit the parallelisation available to APT in this result as we only use a single device. Additionally, we are aiming to get the ablation done before the end of rebuttals.
>
> Additionally, from Figure 2 in [9], we see that other neural samplers require more than $10^8$ target evaluations for good performance on GMM-40 in 2D. This suggests they would require much more evaluations on harder targets such as DW-4, MW-32 and ALDP. Hence, this suggests that the number of steps that we require are still within a reasonable range.

---

> ### Author Response · Authors · 2025-11-25
> **Reply to Reviewer 93BL from the authors**
>
> > In Fig. 4, how do you explain the large performance gap...
>
> This difference in performance is due to the fact that Diffusion only simulates the reverse SDE to sample from the target and does not have any Metropolis–Hastings correction mechanisms in order to correct for the bias in the learned score model. In contrast, Diff-APT is able to correct for the bias as it is a MCMC algorithm.
>
> > For ALDP, how are the chains initialized...
>
> All chains are initialised at 0. We could instead initialise from a Gaussian or start with samples from lowest temperature chain. However, we found initialising to 0 works well in practice in this setting. As for sampling from the reference, we achieve this via MCMC sampling.
>
> > For ALDP, is the energy parametrization (L1400) rotation-invariant...
>
> Yes, both the target and the network (EGNN) are rotation-invariant. We project all samples to the CoM-free subspace by subtracting their mean. Implicitly, this is projecting the SDE used by CMCD to the CoM-free subspace. As all transition kernels are Gaussian, only the normalising constant for the Gaussians are changed but this does not affect the Metropolis–Hastings ratio used in APT. Finally, it is a trivial fact that the Boltzmann distribution of ALDP is invariant to the CoM of samples.
>
> ### References
>
> [1] Efficient and unbiased sampling of boltzmann distributions via consistency models
>
> [2] Efficient and Unbiased Sampling from Boltzmann Distributions via Variance-Tuned Diffusion Models
>
> [3] Particle Denoising Diffusion Sampler
>
> [4] Skipping the Replica Exchange Ladder with Normalizing Flows
>
> [5] Non-reversible parallel tempering: A scalable highly parallel MCMC scheme
>
> [6] Learning Interpolations between Boltzmann Densities
>
> [7] Learning normalized image densities via dual score matching
>
> [8] Consistent Sampling and Simulation: Molecular Dynamics with Energy-Based Diffusion Models
>
> [9] No Trick, No Treat: Pursuits and Challenges Towards Simulation-free Training of Neural Samplers
>
> [10] Uniform Ergodicity of Parallel Tempering With Efficient Local Exploration
>
> [11] Iterated Denoising Energy Matching for Sampling from Boltzmann Densities

---

> > ### Author Response · Authors · 2025-11-25
> > **Reply to Reviewer 93BL from the authors**
> >
> > ## Conclusion
> >
> > Please let us know if any additional information or clarifications would be helpful. If we have adequately addressed your points, we hope you will consider increasing your score.

---

> ### Comment · Reviewer_93BL · 2025-11-27
> **Answer to the rebuttal [1/3]**
>
> I would like to begin by emphasizing that my goal as a reviewer is to provide the AC with the most useful insights to support an informed decision, and to offer the authors constructive feedback on aspects of the work that could be clarified or improved. My review is not driven by ego; it is intended to be helpful and to foster productive discussion. I understand the frustrations often associated with the ML conference review process, and I apologize if any of my scientific comments came across as hurtful. With this in mind, I will now respond to the rebuttal.
>
> > [...] we have currently updated our paper to be more grounded in the Metropolis–Hastings notation
>
> Thank you for addressing this. What I had in mind was somewhat closer in spirit to the formulation in [1] (numbering of my answer), but since no other reviewers raised concerns about this aspect and the revised version already moves in that direction, I won’t elaborate further on this point.
>
> > We respectfully strongly disagree with this characterisation. We believe it is quite dismissive to simply judge our contribution as only a modest generalisation of [1, 2, 3] or saying APT has already been covered by [4]. [...]
>
> I also disagree with your characterization. I fully acknowledge the contribution of APT as both a practical and theoretical extension of [1], expanding the class of swap mechanisms through Markov kernels and transferring the associated theory. However, when viewed in the broader context of related work, the contribution appears incremental relative to [2, 3, 4, 5].
>
> In particular, both [1] and [2] develop diffusion-based importance-sampling proposals by composing denoising kernels (hence why, in my view, these works are conceptually closer to AIS than to standard IS). The work in [3] further generalizes this line of research (using connections between AIS and SMC) by introducing Markov kernels into the transition mechanism of SMC, which is precisely the type of modification that APT introduces in the PT setting. Moreover, deterministic kernels have also been incorporated into SMC in prior work [14,15]. From this perspective, the acceptance ratio in APT reduces to the ratio of importance weights in PDDS, underscoring the close parallel.
>
> My point is that the methodological shift you introduce for PT mirrors an already-established modification in the SMC literature, which makes the practical advancement of APT appear incremental. This is further reinforced by the fact that [2] already covers the deterministic-transition special case of APT.

---

> > ### Comment · Reviewer_93BL · 2025-11-27
> > **Answer to the rebuttal [2/3]**
> >
> > > [...] Therefore, we see that properly tuned PT is resilient to mode switching compared to neural samplers and SMC. [...]
> >
> > This is where our core disagreement lies.
> >
> > First, I would like to note that I find the remark “We politely stress that this indicates a misunderstanding of how PT and APT function” unhelpful. PT and APT necessarily operate along a chosen path; in your work, this path is either an annealing path or a noising path. While mode switching is indeed induced by the path, developing methodologies that robustly handle such issues is central to research in PT, APT, SMC, and related samplers.
> >
> > I strongly disagree with the statement “PT does not necessarily suffer from the mode collapse issue [...] Instead, mode switching only affects the mixing rate of PT.” In practice, PT is affected by mode switching in much the same way as SMC. (The reference to “neural samplers” is also not very meaningful here, as many recent ones do not rely on a sequence of intermediate densities at all.) To illustrate this concretely, I reproduced [a simple but representative experiment](https://ibb.co/jkTYbvD9). I considered a path between a Gaussian mixture and a Gaussian distribution using the noising path of a VP diffusion, for which the intermediate marginals are analytically available. I constructed two versions of the path: one with intentional mode switching (via changing mixture weights) and one without, both sharing identical endpoints. I then ran standard PT with AutoMALA transitions (8 steps), 50k MCMC iterations, and 128 uniform discretization levels. I show the chain trajectories over the 128 discretization points (with the true path in the background), followed by the resulting sample histogram, the swap-acceptance rates for all even and odd pairs, and finally the histogram of the samples at $t=0$.
> >
> > The results are unambiguous: PT is highly sensitive to mode switching. Even though swap acceptance rates remain extremely high (all above 95%), the final sample distribution exhibits significant weight misestimation in the multimodal case. This indicates that the schedule tuning algorithm of [1] is unlikely to alleviate such issues.
> > More generally, the claim that mode switching “only affects the mixing rate” is, in my view, not accurate. By that logic, multimodality “only” affects the mixing rate of the Langevin algorithm as well, yet this is precisely why substantial research in MCMC design continues.
> >
> > Given that mode switching is a genuine difficulty, the relevant question becomes: Can APT mitigate it? Based on the above observations, I doubt that tuning the schedule helps in this regime. Regarding the transition kernels: although NF-APT uses expressive bijective transformations, in practice the SKL objective (especially its reverse-KL component) could tend to favor mode-seeking, and thus often induces local refinements rather than global moves. Depending on the data feeding the forward-KL term, locality may again dominate. Therefore, it is unclear how APT’s kernels would address the underlying problem.
> >
> > I appreciate the authors’ acknowledgement of the limitations of Diff-APT. In practice, Diff-APT yields paths with substantial mode switching (typically far more severe than in my illustrative example) and its transition kernels are necessarily local. I believe these limitations should be made explicit in the manuscript, as they materially constrain the applicability of Diff-APT in genuinely multimodal settings.

---

> ### Comment · Reviewer_93BL · 2025-11-27
> **Answer to the rebuttal [3/3]**
>
> > [...] This means APT is fundamentally distinct from neural samplers (or SMC methods), making an apples-to-apples comparison difficult. [...]
>
> I appreciate the authors’ position and agree that comparing algorithms with fundamentally different structures is challenging, especially when parallel execution is involved. However, stating that such comparisons are effectively not possible overlooks a substantial body of work specifically aimed at developing fair and meaningful cross-paradigm evaluations [10,11,12,c,16]. It is also incorrect to assert that neural samplers lack theoretical guarantees; at minimum, the approach of [17] provides well-established theoretical guarantees.
>
> To be clear, I never claimed that PT is a weak baseline, PT is indeed a strong and widely used MCMC method. My concern is solely about evaluation breadth. While the round-trip rate is a standard diagnostic within the PT literature, several reviewers (including myself) believe that relying on it almost exclusively provides an incomplete picture. In my illustrative example, for instance, even when one mode disappears or the weights are significantly misestimated, the round-trip rate can remain high, offering no indication that the sampler is producing incorrect marginal distributions.
>
> I appreciate the additional experiments the authors provided. However, only the inclusion of complementary metrics (ones that directly assess distributional accuracy) would meaningfully address my concerns. Adding such metrics would considerably strengthen the empirical evaluation.
>
> **Conclusion**
>
> Most of my remaining questions are now either addressed or of secondary importance. Overall, I will keep my score for the moment. In my view, the main contribution of this work is the extension of the theoretical framework introduced in the journal paper [1]. My reasons for recommending rejection at a machine learning venue such as ICLR are the following: (i) the core idea of accelerating swap moves via Markov kernels has already been explored in AIS and especially SMC, but is only lightly cited here and not meaningfully compared to (either theoretically or empirically); (ii) the paper does not convincingly demonstrate that these improved swaps address key challenges in multimodal sampling (e.g., mode switching), where PT is known to struggle; (iii) the comparison to the broader literature on ML-enhanced sampling is insufficient, with diverse methods grouped under a vague category of “neural samplers”; and (iv) while the metrics used are standard within the PT community, they are opaque to ML practitioners and may obscure important limitations.
>
> I remain open to further discussion and am willing to raise my score if the authors substantively address any of these points.
>
> **Additional references**
>
> [14] Arbel, M., Matthews, A., & Doucet, A. (2021). Annealed Flow Transport Monte Carlo. In Proceedings of the 38th International Conference on Machine Learning (pp. 318–330). PMLR.
>
> [15] Matthews, A., Arbel, M., Rezende, D., & Doucet, A. (2022). Continual Repeated Annealed Flow Transport Monte Carlo. In Proceedings of the 39th International Conference on Machine Learning (pp. 15196–15219). PMLR.
>
> [16] Blessing, D., Jia, X., Esslinger, J., Vargas, F., & Neumann, G. (2024). Beyond ELBOs: A Large-Scale Evaluation of Variational Methods for Sampling. In Proceedings of the 41st International Conference on Machine Learning (pp. 4205–4229). PMLR.
>
> [17] Gabrié, M., Rotskoff, G., & Vanden-Eĳnden, E. (2022). Adaptive Monte Carlo augmented with normalizing flows. Proceedings of the National Academy of Sciences, 119(10), e2109420119.
>
> **Appendix**
>
> * The target distribution is $\pi(x) = 0.45 \mathcal{N}(-3.5 \mathbf{1}_2, 0.2^2 \mathrm{I}_2) + 0.5 \mathcal{N}(\mathbf{0}_2, 0.4^2 \mathrm{I}_2) + 0.25 \mathcal{N}(+3.5 \mathbf{1}_2, 0.2^2 \mathrm{I}_2)$
> * The noising is the VP with a factor $\sigma = 1/\sqrt{2}$ (such that $\eta$ is a centered Gaussian with variance $\sigma^2$) in front of the Brownian motion with a linear schedule with $\beta_{max} = 20.0$ and $\beta_{min} = 0.1$
> * The mode-switching version assigns a weight $w_1(t) = \min(\max(\cos^2(4 \pi t), 0.15), 0.45)$ to the first mode, the second mode stays at $w_2(t) = 0.5$ and the third mode is at $w_3(t) = 1 - (w_1(t) + w_2(t))$

---

> > ### Author Response · Authors · 2025-12-01
> > **Reply to Reviewer 93BL's response from the authors**
> >
> > > the core idea of accelerating swap moves via Markov kernels has already been explored in AIS and especially SMC...
> >
> > We maintain that we strongly disagree with this characterisation of our work. We stress that AIS/SMC and PT are two completely different algorithmic frameworks, with completely different motivations (from importance sampling/particle methods and MCMC), literatures, computational properties, diagnostics and guidelines. Therefore, it is reductive to simply state that our work is merely incremental, especially taking into account our formalisation of incorporating additional flexibility within the swap moves used in PT, mathematical analysis and empirical results.
> >
> > > I strongly disagree with the statement “PT does not necessarily suffer from the mode collapse issue...
> >
> > To address this point, we first note that:
> >
> > # the mode-switching experiment by the reviewer is completely incorrect making their conclusion deeply misleading.
> >
> > This is due to the fact that the two paths they consider (the standard diffusion path and mode-switching path) **do not have identical endpoints**.
> >
> > To see this, their target distribution is defined by $\pi\propto 0.45\mathcal{N}(-3.5, 0.2^2) + 0.5\mathcal{N}(0, 0.4^2) + 0.25\mathcal{N}(3.5, 0.2^2)$ and their diffusion path is induced by $dX_t = -\frac{1}{2}\beta_tX_t dt + \sigma \sqrt{\beta_t}dB_t$. Then, their mode-switching path is defined to be the same as diffusion path, but where we also change the weights of the modes by $w_1(t) = \text{min}(\text{max}(\cos^2(4\pi t), 0.15), 0.45)$ and $w_2(t)=0.5$ and $w_3(t) = 1 - (w_1(t) + w_2(t))$.
> >
> > However, we can see that at $t=0$, we have $w_1(0)=0.45, w_2(0)=0.5$ and $w_3(0) = 0.05$. **This defines a different target distribution $\pi'$ to $\pi$, therefore, their paths do not have identical endpoints**. To see the difference in the distributions, see https://ibb.co/x8q8GTzB.
> >
> > We also note that this is not just a typo by the reviewer. If we look at the reviewer's plots at https://ibb.co/jkTYbvD9, we see that for "Histogram at $t=0$", the target histograms (for both paths) represent $\pi$, as the last two modes have the same peak, whereas the left plot should represent $\pi'$. Indeed, it looks like if we plot the correct target $\pi'$ for the left-hand plots, it looks like their implementation of PT **does actually sample from the correct target despite mode switching**.
> >
> > We have verified this ourselves by implementing PT to sample using the mode-switching path (which targets $\pi'$) and we see that PT does actually sample from the correct target $\pi'$ correctly. See https://ibb.co/dJktrXgy for the plot of PT samples at the target distribution, and see https://ibb.co/vxdzXpTk for the plot of how the PT samples evolve during time.
> >
> > **Therefore, this invalidates the reviewer's conclusion from their experiment.** Indeed, the experiment actually supports that PT is robust to mode switching. Hence, we **strongly disagree** with the reviewer's viewpoint that round trips do not properly capture the performance of APT/PT and APT cannot function in multimodal settings. As the reviewer's criticism are mainly based on their experiment and their erroneous conclusion, we would like to highlight that this also raises questions about the points raised in their review. Overall, we maintain the position we set out from our original response on mode switching.
> >
> > Beyond the reviewer's mistake, we stress that dealing with mode switching is  **orthogonal** to the contributions in our paper. Our main contribution is showing how the convergence of PT can be accelerated by introducing flexibility in swaps, whereas dealing with mode switching requires designing new annealing paths (which is a different concern). Indeed, APT can be used with arbitrary annealing paths, so any methods that design new annealing paths to mitigate mode switching can be used with APT.
> >
> > Additionally, we note that Diff-APT can be seen as tackling mode switching from using an annealing path defined by a diffusion process which is known to not suffer from mode switching, instead of the linear annealing path. Additionally, issues with score blindness can be tackled with our previous suggestions of using [1, 2] to regularise the loss. Therefore, we strongly disagree with the reviewer's view that Diff-APT is not applicable in multimodal settings, especially since we do report good performance for ManyWell-32, which is a very multi-modal target, in Section 6.3 for Diff-APT.

---

> > > ### Author Response · Authors · 2025-12-01
> > > **Reply to Reviewer 93BL's response from the authors**
> > >
> > > > Additional metrics...
> > >
> > > As for the reviewer's concern about the lack of other metrics, we restate that APT is a MCMC algorithm that has mathematical gurentees of asymptotic convergence, therefore, round trips (which measures the mixing rate [3]) is the correct metric to assess the performance of APT.
> > >
> > > Nevertheless, we remind the reviewer that we do have additional metrics assessing the quality of our samples. For example, in Figure 3, we report normalising constant estimation for APT, showing that APT recovers the ground-truth normalising constants and with smaller variance than PT. Moreover, from looking at Appendix E.2 which reports round trip performance on DW-4 and MW-32, we see that round trips are positively correlated with normalising constants estimation performance. We also remind the reviewer that we compare against other neural samples in Section 6.4 where we show APT surpasses them in performance. Therefore, we would like to strongly push back against the reviewer's claim that we do not provide metrics other than round trips or compare against other neural samplers.
> > >
> > > Additionally, here we report W2 (2-Wasserstein) metrics for GMM-10 where we compare sampling with CMCD-APT vs the SMC algorithm in [4] (CMCD-SMC) - i.e. we take the same learned transport map from CMCD-APT and use it for APT sampling and SMC sampling with the algorithm from [4] (we resample when ESS < 30% and we also use systematic resampling which performs better than multinomial resampling). We draw 100,000 samples for both APT and SMC - i.e. we run APT for 100,000 iterations and we use a batch size of 100,000 for SMC. To compute our W2 metric, we subsample 5000 samples out of the final 100,000 samples and compute W2 with an independently sampled set of samples from the target. We then repeat this 50 times. We also use 10 annealing distributions for both methods.
> > >
> > >
> > > | Method   |   $K=1$      |  $K=2$ | $K=5$ |
> > > |----------|:-------------:|------:| ------:
> > > | CMCD-SMC |  $0.142\pm 0.015$ | $0.157\pm 0.008$ | $0.143\pm 0.009$ |
> > > | CMCD-APT | $0.113 \pm 0.016$ | $0.112\pm 0.011$ | $0.102 \pm 0.011$ |
> > >
> > > Also, for PT, we have $0.118\pm 0.011$.
> > >
> > > **Here, we see that CMCD-APT outperforms CMCD-SMC in terms of the W2 metric**.
> > >
> > > For another experiment to assess the distributional accuracy of APT compared to other neural samplers, here we plot the histogram of interatomic distance of samples from DW-4 (see Section 6.4) comparing CMCD-APT with CMCD-SMC (for $K=1$) with the same setup as above. See the plot here: https://ibb.co/pvM8K9h2. We see that CMCD-APT is able to correctly recover the marginal distribution of interatomic distances compared to CMCD-SMC which cannot.
> > >
> > > Also, for the ablation that the reviewer requested previously for the performance of APT where we use fewer samples during training, we have the following, where we test with CMCD-APT with $K=1$ and 10 chains on DW-4.
> > >
> > > | Number of samples collected per training iteration   |   Round trips      |
> > > |----------|:-------------:|
> > > | 2,000 | 4036 |
> > > | 5,000 | 6689 |
> > > | 10,000 | 6636 |
> > > | 50,000 | 6857  |
> > >
> > > As a baseline, we have PT with 10 chains has 5128 round trips. Therefore, we see that APT is flexible in terms of samples it needs.
> > >
> > >
> > > ## References
> > >
> > > [1] Learning normalized image densities via dual score matching
> > >
> > > [2] Consistent Sampling and Simulation: Molecular Dynamics with Energy-Based Diffusion Models
> > >
> > > [3] Uniform Ergodicity of Parallel Tempering With Efficient Local Exploration
> > >
> > > [4] Sequential Controlled Langevin Diffusions

---

> > > > ### Author Response · Authors · 2025-12-01
> > > > **Reply to Reviewer 93BL's response from the authors**
> > > >
> > > > ## Conclusion
> > > >
> > > > We hope that our response has been able to resolve the reviewer's concerns.

---

### Official Review · Reviewer_xVCo · 2025-11-01

**Soundness:** 3
**Presentation:** 3
**Contribution:** 1
**Rating:** 2
**Confidence:** 5

**Summary:**

This paper introduces Accelerated Parallel Tempering (APT), a new framework that integrates neural samplers into Parallel Tempering (PT) to improve sampling efficiency on complex, high-dimensional, and multi-modal distributions. Classical PT often struggles because adjacent temperature distributions share low overlap, limiting swap acceptance rates and requiring a large number of parallel chains. APT addresses this by using neural transports to “bridge” neighboring distributions before proposing swaps, dramatically increasing swap acceptance and reducing the number of chains needed—while preserving the exact asymptotic correctness of PT.

**Strengths:**

1. Math derivation is sound.

2. PT is known to be an effective sampler for multi-modal simulations. If diffusion-enhanced sampler really works, PT will surely boost the perfomance of multi-modal simulation.

**Weaknesses:**

1. I don't like the whole community that utilizes ideas like the diffusion model to do sampling, which is super expensive and doesn't make a lot of sense. I have extensive research experience in sampling and diffusion models but I don't this is a nice combination.

2. The motivation why do and when do we need diffusion models to do sampling is not well-supported. The intuition why a backward process is needed is not explained clearly.

3. for section 6.1, measuring the round trip is not a good idea, the eventual goal is to sample the 40-mode mixture distribution, you can simply measure the empirical TV/ KL/ W1 error.

4. No scalable real-world experiments. Molecular is too small.

**Questions:**

1. If gradient information is already cheap, who won't we just use vanilla PT sampler?

2. Low acceptance probability occurs when these distributions have minimal overlap. Addressing this requires increasing
the number of parallel chains N, which may not always be possible. Why? If a problem cannot be simulated using 4-chain PT, it must be a hard problem, I don't expect this algorithm to solve it as well.

3. Could you show the distribution error plot by comparing vanilla PT and CMCD-APT using different computational budgets?

---

> ### Author Response · Authors · 2025-11-25
> **Reply to Reviewer xVCo from the authors**
>
> We thank the reviewer for their time taken to review our paper. We hope to be able to both answer and clarify the points raised in the review.
>
> ## Weaknesses
>
> We would like to gently push back against the reviewer's view on this area of research as we believe that it is an interesting avenue to explore how advances in generative modelling and deep learning can accelerate the fundamental problem of sampling. Indeed, there has already been workshops for this area: https://sites.google.com/view/fpiworkshop/about?authuser=0 and https://fpineurips.framer.website/.
>
> > for section 6.1, measuring the round trip is not a good idea...
>
> We should like to note that using empirical TV/ KL/ W1 error is not suited to evaluating the performance of PT/MCMC-type algorithms due to the fact that such algorithms already have asymptotic guarantees of convergence (unlike neural samplers). Therefore, the more relevant metric for evaluating APT is the mixing rate which is captured by the round trip rate [1].
>
> > No scalable real-world experiments...
>
> We would like to gently push back on this point as we evaluate on Alanine Dipeptide which is a very challenging target that neural samplers struggle on. However, we do agree that an important future direction is to be able to scale up neural samplers and APT to more and more challenging distributions.
>
> ## Questions
>
> > Low acceptance probability occurs...
>
> In practice, many medium-sized and high-dimensional multi-modal problems cannot be adequately explored with a 4-chain vanilla PT scheme, and even when they can, the mixing times can still be long. Our work is explicitly motivated by this: APT is designed to accelerate convergence under the same number of energy/gradient evaluations by increasing swap acceptance and improving communication between annealing distributions.
>
> ### References
>
> [1] Uniform Ergodicity of Parallel Tempering With Efficient Local Exploration
>
> ## Conclusion
>
> Please let us know if any additional information or clarifications would be helpful. If we have adequately addressed your points, we hope you will consider increasing your score.

---

> ### Comment · Reviewer_xVCo · 2025-11-25
>
> Thanks for the reply.
>
> **asymptotic guarantees of convergence**
>
> An infinite time guarantee doesn't necessarily show it is fast in practice. Some speed comparison is helpful. Round trips are a good metric, but using that alone is not sufficient.
>
> **Alanine Dipeptide is a very challenging target for neural samplers**
>
> To my knowledge, Alanine Dipeptide is not particularly difficult for standard PT samplers. The fact that neural samplers struggle on such a baseline raises concerns about their practical effectiveness for real-world sampling tasks.
>
> **If a real-world high-dimensional/ multi-modal problem cannot be adequately explored with a 4-chain vanilla PT scheme**
>
> In that case, is exact sampling practically possible? Why don't we acknowledge the challenge and adopt variational inference?
>
> **Are there any tasks where generative/ neural samplers outperform standard samplers if compared fairly? This may be something I am missing.**

---

> > ### Author Response · Authors · 2025-11-26
> > **Reply to Reviewer xVCo from the authors**
> >
> > We would like to thank the reviewer for their helpful suggestions. Below we provide a response to the reviewer's concerns.
> >
> > We would like to note that neural samplers are an active and very recent area of research and comparing directly against mature sampling algorithms is not necessarily fair, as people are still exploring how to improve their performance. Moreover, we would like to gently state that the main purpose of our paper is on how neural samplers can be incorporated within PT (and providing the mathematical foundations for this and analysing APT) to allow for additional flexibility, in order to improve sampling performance. The training of these neural samplers is not the main point of the paper (however, we still note that APT does allow for some interesting training benefits for neural samplers from the ability to parallelise training across chains and employ the symmetric KL as an objective).
> >
> > > Round trips are a good metric, but using that alone is not sufficient...
> >
> > We would like to note that round trips are a gold-standard metric for evaluating PT-type algorithms. Moreover, they provide a hardware-agnostic metric for evaluating the speed at which PT-type algorithms convergence which is more reliable than hardware dependent metrics for comparing algorithms.
> >
> > > If a real-world high-dimensional/ multi-modal problem cannot be adequately explored...
> >
> > We would like to note that such situations, where a certain number of chains are not enough to achieve fast mixing sampling, is exactly what APT is designed to help rectify from employing neural samplers to accelerate mixing. Indeed, we can view NF-APT and CMCD-APT as employing variational inference to accelerate mixing while still keeping the mathematical guarantee of PT.
> >
> > ### Conclusion
> >
> > We hope to have addressed the reviewer's concerns. If we have, we hope you will consider increasing your rating.

---

> ### Comment · Reviewer_xVCo · 2025-11-27
>
> **The training of these neural samplers is not the main point of the paper**
>
> Why not? Diffusion models became popular only because we have abundant real-world image and video data. Neural samplers, in contrast, typically rely on simulated data to obtain their training samples. But if high-quality samples can already be generated through simulation, then the underlying sampling problem is essentially solved. This is my biggest concern.
>
> Nevertheless, I appreciate the authors' hard work and have improved the score to 4.
>
> =================================================================
>
> After discussing with colleagues working on neural samplers, I realized that some of my earlier views were incomplete. Amortizing the sampling cost and enabling efficient generation of new samples may indeed be an interesting direction for future work. I have updated my score to **6** and lowered my confidence accordingly.
>
> I also encourage the authors to articulate the current limitations of this area in the final version, so that junior researchers can gain a realistic understanding of its challenges and potential.
>
>
> [1] Deep unsupervised learning using nonequilibrium thermodynamics, 2015

---

> > ### Author Response · Authors · 2025-11-27
> > **Reply to Reviewer xVCo from the authors**
> >
> > We greatly thank the reviewer for reconsidering and updating their rating, and for supporting our paper.
> >
> > We fully agree with the reviewer that improving the training of neural samplers is an important direction of future work and that current approaches do still face many hurdles. We will ensure to add a discussion on the promise and current limitations of neural samplers in the camera-ready version of our paper. We hope that this will help readers in assessing this research area.

---

### Author Response · Authors · 2025-12-04
**Summary of Rebuttal**

To assist the AC with reviewing our submission, we would like to provide the following summary of the discussion during rebuttals.

High-level summary: We provide a unified and general framework (APT) for incorporating additional flexibility within the swap moves used in Parallel Tempering (PT) for accelerating the mixing of PT. We provide the mathematical foundations of APT by extending the standard theory of PT to the APT case. Furthermore, we explore the design space of APT from showing how different neural samplers can be incorporated to accelerate mixing and improve sample quality in a variety of experiments.

Rebuttal Status: We would like to note that the ratings changed in the following way during rebuttals:

- Reviewer xVCo: **2 -> 6**
- Reviewer 93BL: **2** (discussion cut short)
- Reviewer WMqg: **8 -> 8**
- Reviewer bR4T: **4 -> 6**

---

### Meta-Review · Area_Chair_KsYe · 2026-01-07

**Summary:**

This paper attempts to address the outstanding problem of sampling from high-dimensional and/or multimodal unnormalized density, by combining two powerful techniques, namely the classic idea of parallel tempering, and the recent progress of neural sampler. The benefits of the method are demonstrated both theoretically and empirically. Reviewers initially had significant concerns, but the majority of them were resolved during the rebuttal discussion. Authors stated that the promised rating changes were xVCo: 2 -> 6, 93BL: 2 (discussion cut short), WMqg: 8 -> 8, bR4T: 4 -> 6, and I checked carefully and confirm that was indeed the case. This brings the paper to a borderline case. Further based on the discussion between authors and reviewer 93BL, as well as my own reading of the submission, I recommend acceptance.

**Reviewer Concerns:**

Most are addressed.

**Reviewer Scores:**

already happened:
xVCo: 2 -> 6, WMqg: 8 -> 8, bR4T: 4 -> 6,

unclear:
93BL, but he might increase from 2.

---

### Decision · Program_Chairs · 2026-01-26

Accept (Poster)